# Tempora: Characterising the Time-Contingent Utility of Online Test-Time Adaptation

Sudarshan Sreeram [1]   Young D. Kwon [2]   Cecilia Mascolo [1]

## Abstract

Test-time adaptation (TTA) offers a compelling remedy for machine learning (ML) models that degrade under domain shifts, improving generalisation *on-the-fly* with only unlabelled samples. This flexibility suits real deployments, yet conventional evaluations unrealistically assume unbounded processing time, overlooking the accuracy-latency trade-off. As ML increasingly underpins latency-sensitive and user-facing use-cases, temporal pressure constrains the viability of adaptable inference; predictions arriving too late to act on are futile. We introduce *Tempora*, a framework for evaluating TTA under this pressure. It consists of temporal scenarios that model deployment constraints, evaluation protocols that operationalise measurement, and time-contingent utility metrics that quantify the accuracy-latency trade-off. We instantiate the framework with three such metrics: ❶ *discrete* utility for asynchronous streams with hard deadlines, ❷ *continuous* utility for interactive settings where value decays with latency, and ❸ *amortised* utility for budget-constrained deployments. By applying Tempora to 11 TTA methods, we find that *rank instability* persists across 750+ temporal evaluations spanning diverse datasets, models, and hardware platforms; *i.e.,* conventional rankings do not predict rankings under temporal pressure. The highest-utility method varies with the shift and temporal pressure, with no clear winner. By enabling systematic evaluation across diverse temporal constraints for the first time, Tempora reveals when and why rankings change, offering practitioners a lens for method selection and researchers a target for deployable adaptation. Code: https://github.com/sudotensor/tempora.

[1]Department of Computer Science and Technology, University of Cambridge, Cambridge, England, United Kingdom [2]Samsung AI Center, Cambridge, England, United Kingdom. Correspondence to: Sudarshan Sreeram <ss3122@cam.ac.uk>.

*Proceedings of the $43^{rd}$ International Conference on Machine Learning*, Seoul, South Korea. PMLR 306, 2026. Copyright 2026 by the author(s).

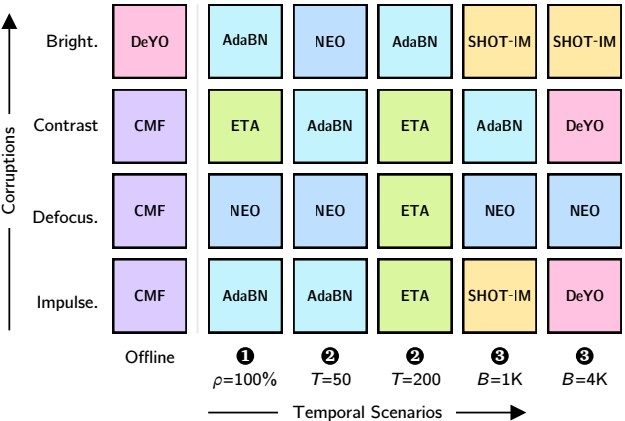

*Figure 1.* **Rank instability persists across temporal evaluations**. Cells show the highest-utility method among nine TTA methods tested on ImageNet-C (ResNet-50). *Offline*: No time constraints. *Temporal*: ❶ Discrete (hard deadlines), ❷ Continuous (latency penalties), and ❸ Amortised (budgeted overhead); formal definitions in §3. Rows reveal *rank instability*: the best method changes under temporal pressure. Similarly, columns show no consistent winner within any scenario. Aggregate benchmarks obscure these corruption-level dynamics. Extended results in Appendix C.

## 1. Introduction

Machine learning models increasingly serve as essential components of non-stationary real-world systems, from on-device scene analysis in computational photography (Apple, 2022) to tumour segmentation in clinical decision support (Dorfner et al., 2025). Standard inference treats these models as static artefacts and assumes that representations learned on fixed training distributions can inherently generalise to settings where the test distribution $\mathcal{T}$ drifts stochastically. However, this assumption is fragile: a ResNet-50 model (He et al., 2016) achieving 76.1% accuracy on ImageNet (Russakovsky et al., 2015) drops to 18% on ImageNet-C corruptions (Niu et al., 2022; Hendrycks & Dietterich, 2019); in practice, such corruptions arise unpredictably in systems like wildlife camera traps, where species classifiers face site-specific conditions unseen during training.

Test-time adaptation (TTA) addresses this fragility by treating inference as an adaptive process rather than a static endpoint. In the "*Fully TTA*" setting (Wang et al., 2021), in-

situ calibration uses only the incoming stream $x_i \sim \mathcal{T}$ and pre-trained parameters $\theta$ to bridge the generalisation gap, not requiring training data, labels, or modifications to pre-training. This flexibility becomes essential as models scale. Training-time robustness via augmentation cannot feasibly cover the combinatorial space of real-world shifts (Sun et al., 2020)[1], and model economics increasingly favour reuse over per-scenario re-training (Bommasani et al., 2021). While TTA enables this reuse, it is not without cost. Unlike standard inference, TTA introduces overhead, whether through closed-form solutions, gradient updates, or iterative optimisation. These costs lie on the critical path of each prediction, making a system's response time a function of design choices that current evaluation methodology ignores.

Crucially, these evaluations typically assume unbounded processing overhead for adaptation, releasing a new batch only after adaptation on the previous batch completes; we term this *offline*. However, in latency-sensitive deployments, a correct prediction arriving after a user has closed an app or a robot has committed to an action is futile; correctness and timeliness are jointly necessary. Latency, when reported, appears only to substantiate aggregate efficiency gains rather than to serve as a constraint that determines system viability.

Alfarra et al. (2024) surfaced this tension by using a method's processing overhead to limit adaptation opportunities; slower methods receive fewer batches. Their evaluation addresses a *discrete* temporal scenario where missing a deadline yields no value. However, real-world deployments span at least two more scenarios. First, in interactive systems, late predictions degrade user experience rather than causing outright failure; an on-device photo organiser that auto-tags images during bulk upload remains useful with a brief delay, but users grow impatient with longer waits. Second, some systems constrain total adaptation overhead while remaining agnostic to per-prediction latency. A drone performing visual crop inspection may budget onboard compute to preserve battery, requiring only that the mission completes before power runs out.

Without systematic coverage of these scenarios, practitioners cannot identify which methods fit their temporal constraints, and researchers lack a target for designing methods that will. Genuinely deployable methods justify their overhead through accuracy gains, integrating naturally into latency-sensitive systems; *computationally insolvent* methods consume budgets no accuracy benefit can repay. As methods grow increasingly sophisticated, temporal evaluation becomes essential; without it, the field risks chasing offline accuracy while producing methods that cannot serve real-world systems where adaptation is most needed.

In this paper, we introduce *Tempora*, a framework for evaluating TTA under temporal constraints. It comprises temporal scenarios (contextualising deployments), evaluation protocols (operationalising measurement), and time-contingent utility metrics (quantifying the accuracy-latency trade-off). Our contributions are as follows:

**Deployment realism and scope.** We ground Tempora in physical units (ms) that replace the model-relative proxies of prior work. This enables us to simulate exogenous constraints (*e.g.,* 50 ms response deadline) in an interpretable and consistent manner. In doing so, we revise Alfarra et al.'s discrete evaluation protocol and introduce the *continuous* and *amortised* archetypes. Specifically, we instantiate our framework with three metrics for distinct scenarios (§3): ❶ *Discrete utility* for hard deadlines, ❷ *Continuous utility* for interactive responses, and ❸ *Amortised utility* for budget-constrained deployments. Each couples accuracy with a scenario-appropriate temporal penalty; together, they map utility across the deployment space rather than a single point in a temporal vacuum.

**Utility decomposition and diagnosis.** We made Tempora's utility metrics decomposable to diagnose *when* and *why* rankings change, going beyond just *whether* they do. This decomposition allows us to isolate three failure modes, one from each scenario: availability collapse from missed batches, accuracy erosion due to delayed predictions, and harmful adaptation; all stem from adaptation overhead not justified by accuracy gains under temporal pressure.

**Persistence of rank instability and implications.** We apply Tempora to nine Fully TTA methods across 16 temporal scenarios on the 15 corruptions of ImageNet-C with ResNet-50[2]. We observe that *rank instability*, where offline rankings fail to predict rankings under temporal pressure, persists across a wider and more physically interpretable taxonomy of deployment constraints than previously reported by Alfarra et al. (2024). Of 240 temporal evaluations, the majority offline winner, CMF (Lee & Chang, 2024), loses in 211 cases (87.9%) and yields 15% utility on average to the winner. Rank instability varies across corruption types and increases with temporal pressure (§4.2). These trends and the observed failure modes inform three necessary properties for deployable adaptation that current methods lack (§4.3): compute allocation conditioned on the difficulty of the input shift, time-aware scaling that bounds response delay, and anytime performance so elapsed overhead yields gains over not adapting at all.

While conventional evaluation of TTA methods asks which

---

[1] While safety-critical systems may pre-train on anticipated shifts, exhaustive coverage of real-world conditions is infeasible. TTA targets the residual shifts that escape such anticipation.

[2] We use this conventional, episodic TTA benchmark to enable direct comparison with prior works. However, Tempora applies readily to other architectures, datasets, and hardware platforms. We show that our conclusions generalise to an extended evaluation suite, including ViT-B/16 on ImageNet-C/R/V2, in Appendix C.

is most accurate, Tempora reframes evaluation to ask which delivers acceptable accuracy at a viable cost. This revised framing informs both the selection and design of adaptation methods. For selection, rank instability under temporal pressure reveals winners that offline rankings obscure. For design, utility decompositions expose not just which methods win but why others fail. With this lens, adaptation can, for the first time, be systematically evaluated across the temporal constraints of real-world systems it aims to serve.

## 2. Background & Related Work

**Domain Shifts.** Models trained on a source distribution $\mathcal{S}$ often degrade when deployed on a target distribution $\mathcal{T}$ where $\mathcal{S} \neq \mathcal{T}$. The degradation depends on what changes between the distributions. This paper focuses on covariate shift, which occurs when the input marginals differ, *i.e.,* $\mathcal{P}_{\mathcal{S}}(\mathbf{X}) \neq \mathcal{P}_{\mathcal{T}}(\mathbf{X})$, whilst the conditional $\mathcal{P}(\mathbf{Y}|\mathbf{X})$ remains stable (Sugiyama et al., 2007). For example, wildfires cast an unnatural orange haze, causing the appearance of *known* objects to differ. While other shifts exist, TTA literature predominantly benchmarks against covariate shifts induced by *corruptions* (Hendrycks & Dietterich, 2019).

**Test-Time Adaptation.** Approaches to handling these shifts vary by data availability and temporal scope. We focus on *Fully Test-Time Adaptation* (Fully TTA) (Wang et al., 2021), which constrains adaptation to the test environment using only pre-trained parameters $\theta$ and incoming unlabelled samples $x_i \sim \mathcal{T}$. This precludes access to source data or target labels, ensuring broad applicability as a drop-in enhancement for standard inference pipelines. We further focus on the *online* setting, which imposes a sequential commitment: the system must predict on the current input before observing the next input, necessitating *in-situ* processing.

**Method Taxonomy.** Existing Fully TTA methods can be broadly categorised by the component of the inference pipeline they modify. *Input adaptation* methods (Gao et al., 2023) project samples back to the source domain. *Output adaptation* (Boudiaf et al., 2022) and *activation adaptation* (Liu et al., 2021; Murphy et al., 2025) adjust logits or features directly, often keeping the backbone frozen. The most common category, *parameter adaptation* (Wang et al., 2021; Niu et al., 2022; Liang et al., 2020), updates network weights, typically affine parameters in normalisation layers, via backpropagation on self-supervised objectives such as entropy minimisation. A method's adaptation overhead may occur before a prediction is made, after it, or both. The split of operations on either side of the prediction boundary determines how the accuracy-latency trade-off manifests under different temporal constraints.

**Evaluation Contexts.** Conventional, offline evaluation ignores how adaptation overhead interacts with accuracy, un-

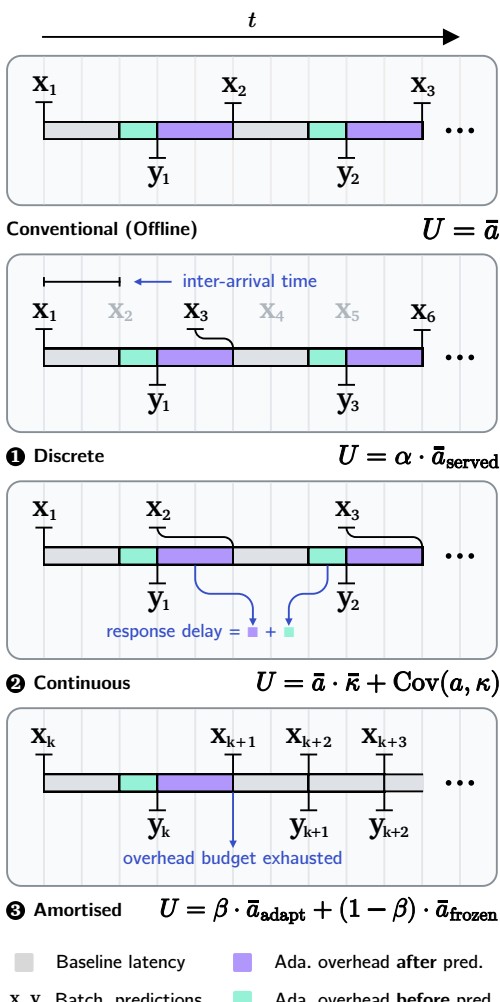

*Figure 2.* **Evaluation protocols for measuring time-contingent utility.** Each panel depicts the interaction between an input stream, a single model pipeline, and predictions over time $t$. Processing batch $\mathbf{x}_i$ to emit prediction $\mathbf{y}_i$ incurs baseline latency (grey), intrinsic overhead (green) that delays emission, and extrinsic overhead (purple) that stalls the pipeline before the next batch can begin; methods vary in their overhead profile. *Offline*: The stream waits for adaptation to complete before releasing the next batch; utility is the mean accuracy. ❶ *Discrete*: Batches arrive asynchronously at fixed intervals; those arriving while the pipeline is occupied are skipped, so utility scales with availability $\alpha$. ❷ *Continuous*: The stream releases $\mathbf{x}_{i+1}$ upon receiving $\mathbf{y}_i$ (greedy user); response delay incurs hyperbolic penalty $\kappa$. ❸ *Amortised*: Follows offline until overhead budget is exhausted, then frozen inference only; utility is a weighted combination over both phases.

realistically assuming unbounded time to serve predictions; see Figure 2. Recent progress on improving evaluation has focused on *distributional* realism and *epistemic* practices. BoTTA (Danilowski et al., 2025) models scenarios with limited data and non-stationary streams; UniTTA (Du et al., 2025) models 36 such streams via a Markov state-transition matrix. TTAB (Zhao et al., 2023b) evaluates beyond co-

variate shifts and also critiques hyperparameter sensitivity and pre-trained model selection. However, the *operational* context, specifically the temporal pressure imposed by the wider system, remains under-explored.

**Temporal Pressure.** Alfarra et al. (2024) address this gap by penalising slow adaptation methods with fewer batches from an asynchronous data stream, where batches arrive at intervals $\gamma$ set to the standard inference latency. This models a discrete, "use-it-or-lose-it" scenario where predictions are deadline-driven. When the wall-clock time under adaptation $\delta$ exceeds $\gamma$, $\lceil \delta/\gamma \rceil - 1$ subsequent batches bypass adaptation and receive predictions from a fallback model.

They find that, under temporal constraints, faster and simpler methods that adapt on more data outperform slower, sophisticated methods. This mirrors an earlier observation from online continual learning by Ghunaim et al. (2023), who used a similar sample starvation penalty. Both works reveal what we term *rank instability*: the failure of offline rankings to predict rankings under temporal constraints.

However, both arrive at this observation through a narrow construction: they express the speed of a data stream relative to a model-specific proxy; Alfarra et al. (2024) do so relative to the source model's inference latency, and Ghunaim et al. (2023) do so relative to a baseline's FLOPs. As such, the temporal constraint is implicitly defined and stretches to accommodate the algorithm (*e.g.,* tuning a stream's speed to match a method's FLOPs) rather than reflecting a fixed deployment reality (*e.g.,* a camera's frame rate).

This proxy makes it hard to model diverse operational conditions: if the "unit" of evaluation is the model's own speed, one cannot easily simulate exogenous factors consistently across models and methods (*e.g.,* a 500 ms user patience threshold). An independent definition is essential to interpret *when* rank instability emerges, not just *whether* it does.

Beyond this, Alfarra et al.'s discrete evaluation protocol, most relevant to our work, has three additional limitations:

- *Risk-free evaluation*: The fallback model guarantees valid predictions, masking cases where adaptation is unavailable. Also, it assumes zero-cost parallel model execution, which is often prohibitive in resource-scarce settings typical of edge deployments. We build on the more realistic single-model variant noted in Alfarra et al.'s appendix to compute discrete utility.

- *Disproportionate penalties*: The ceiling operation creates stepped penalties that obscure incremental efficiency gains. For $\gamma = 100$ ms, any $\delta \in (100, 200]$ ms incurs a 50% miss rate despite nearly $2\times$ variation in overhead. In general, any $\delta \in (k\gamma, (k + 1)\gamma]$ maps to the same miss rate $1 - 1/(k + 1)$, and optimisations within a tier go unrewarded.

- *Forced idling*: The protocol enforces rigid synchronisation, where a batch arriving at $t$ is treated as missed if adaptation is not ready exactly at $t$. Consequently, a method completing at $t + \epsilon$ idles for nearly $\gamma$ despite being available, wasting the window to serve batch $t$.

Finally, prior works do not expose *why* rankings change and only report accuracy, *i.e.,* the net effect; system errors, where batches are skipped, and model errors, where served batches are misclassified, remain conflated in this measure. Our framework, *Tempora*, defined the following section, addresses these limitations. We ground evaluation in physical units (ms) and accordingly revise the discrete protocol, while also extending the scope to a wider taxonomy of constraints that model the accuracy-latency trade-off beyond just discrete temporal semantics. Further details on Alfarra et al.'s protocol appear in Appendix B.

## 3. Tempora

Conventional evaluations decouple theoretical accuracy from operational utility; they model inference as a static mapping $f_\theta : \mathcal{X} \to \mathcal{Y}$, reporting mean accuracy $\bar{a} = \frac{1}{N} \sum \mathbb{1}(f_{\theta_i}(x_i) = \hat{y}_i)$ and implicitly assuming unbounded processing time. Although works report latency to supplement efficiency claims (Niu et al., 2022), it only captures static, aggregate speedups. This decoupling obscures the temporal regimes in which methods remain viable; diffusion-based input adaptation (Gao et al., 2023), for instance, is $810\times$ slower than standard inference (Alfarra et al., 2024).

In this paper, we formalise the data stream as a sequence of tuples $\{(\mathbf{x}_i, t_i)\}_{i=1}^{N}$, where $N$ is the total number of batches and $t_i$ denotes the arrival time of batch $\mathbf{x}_i$. We then address the evaluation gap identified in §2 by introducing three time-contingent metrics that define operational utility $U(\{(\mathbf{y}_i, \delta_i)\}_{i=1}^{N})$ as a functional that maps the joint trajectory of predictions $\mathbf{y}_i$ and latency $\delta_i$ to a scalar value; this conditions utility on both correctness and timeliness. We structure these metrics around three archetypes: *discrete* (environment-led), *continuous* (user-led), and *amortised* (resource-led). The following sections formalise each archetype alongside its temporal scenario, evaluation protocol, and utility metric; together, they form *Tempora*. Concrete examples and parameter choices appear in §4.1.

### 3.1. Discrete Utility

**Scenario.** Following Alfarra et al. (2024), we begin with the most constrained setting: asynchronous streams where batches arrive at fixed intervals despite the readiness of a single-model prediction pipeline. This models autonomous systems, sensor networks, and monitoring applications where data arrives at rates dictated by external processes. If the pipeline cannot keep pace, batches are lost irrecoverably.

**Protocol.** Batches arrive at times $t_i = (i-1)\gamma$ for inter-arrival time $\gamma > 0$. When processing completes, the pipeline serves the most recent arrival; intervening batches are forfeit. We address the disproportionate penalties and forced idling limitations by introducing a batch-sized buffer that holds the incoming batch in memory from $t_i$ to $t_i + \gamma$. In our protocol, batch-skipping is not predetermined and emerges as a natural consequence of the wall-time simulation. Appendix B elaborates on our revision of Alfarra et al.'s protocol.

Let $(s_j, f_j)$ denote the start and finish times of the $j$-th processing event, with $s_1 = 0$ and $f_1 = \delta_1$. The recurrence $s_{j+1} = max(f_j, t_{p_{j+1}})$ and $f_{j+1} = s_{j+1} + \delta_{p_{j+1}}$ governs timing, where $p_j$ denotes the batch served at event $j$: $p_1 = 1$ and $p_{j+1} = max(p_j + 1, \lfloor f_j/\gamma \rfloor + 1)$. This captures two regimes: when $\delta_i < \gamma$, processing outpaces arrivals and $p_j = j$; when $\delta_i > \gamma$, the pipeline falls behind and batches are skipped. Replacing $+1$ with $+2$ in the floor term recovers Alfarra et al.'s stricter synchronisation.

**Metric.** Let $\mathcal{Q}$ denote the set of served batch indices and $a_i$ the accuracy on batch $i$. Discrete utility is formally:

$$U_{\text{discrete}} = \frac{1}{N} \sum_{i \in \mathcal{Q}} a_i = \underbrace{\frac{|\mathcal{Q}|}{N}}_{\alpha} \cdot \underbrace{\frac{1}{|\mathcal{Q}|} \sum_{i \in \mathcal{Q}} a_i}_{\bar{a}_{\text{served}}} \quad (1)$$

Availability $\alpha = |\mathcal{Q}|/N$, the fraction of batches served, and served accuracy $\bar{a}_{\text{served}}$ separate two failure modes: system errors and model errors. Given standard inference accuracy $a_0$, a method must achieve $\bar{a}_{\text{served}} > a_0/\alpha$ to outperform standard inference; for $\alpha = 0.5$, accuracy must double, a requirement that most methods fail to meet.

### 3.2. Continuous Utility

**Scenario.** Discrete utility models environment-led arrivals, but not all systems impose rigid schedules. Timeliness is often governed by interactivity or downstream service-level constraints, where excess latency degrades user experience. We model this via a greedy user who submits batch $i+1$ upon receiving $\mathbf{y}_i$; the stream synchronises with prediction emission, and no batches are skipped. This represents maximal demand on pipeline responsiveness.

**Protocol.** We penalise latency via hyperbolic decay (Mazur, 1987), consistent with evidence linking subjective time perception to value discounting (Brocas et al., 2018). This formulation captures non-linear user patience: a lag of 100 ms is perceptually sharper than an equivalent slip at 1 s. Hyperbolic discounting is desirable as it is monotone decreasing, bounded in $[0, 1]$, equals 1 at zero overhead, and continuous. It preserves meaningful ranking gradation across thresholds.

Let $\lambda$ denote standard inference latency. We decompose $\delta_i = e_i + \ell_i$ at the prediction boundary, where $e_i$ spans pickup to prediction and $\ell_i$ spans prediction to availability.

This reveals two types of adaptation overhead: *intrinsic* overhead $e_i - \lambda$ delays the immediate response, while *extrinsic* overhead $\ell_i$ stalls the pipeline before the next batch is picked up; see Figure 2. The user experiences wait time $w_i = \ell_{i-1} + e_i$, with $\ell_0 = 0$, resulting in an effective response delay $d_i = \max(0, w_i - \lambda)$.

**Metric.** Continuous utility is formally:

$$U_{\text{continuous}} = \frac{1}{N} \sum_{i=1}^{N} a_i \kappa_i = \bar{a} \cdot \bar{\kappa} + \text{Cov}(a, \kappa) \quad (2)$$

where $\kappa_i = (1 + d_i/(T - \lambda))^{-1}$ is a decay factor that discounts accuracy using a human-computer interaction (HCI) threshold $T > \lambda$; value halves when $w_i = T$, and a lower $T$ imposes more temporal pressure. It accommodates service-level constraints where deadlines are targets, not cliffs. The decomposition of utility separates accuracy $\bar{a}$ from responsiveness $\bar{\kappa}$. Alignment $\text{Cov}(a, \kappa)$ is negative when accurate predictions are slow. Matching the baseline requires $\bar{a} \geq a_0/\bar{\kappa}$ at zero alignment.

### 3.3. Amortised Utility

**Scenario.** Both preceding metrics impose per-batch temporal structure, tying deadlines or penalties to individual processing events. Some deployments instead constrain total overhead while remaining agnostic to its distribution. Edge devices, for instance, may budget computation across an overnight session. Varying this budget traces a curve characterising the accuracy-budget trade-off, addressing how much accuracy a method purchases per unit overhead.

**Protocol.** Let $c_i = \delta_i - \lambda$ denote adaptation overhead for batch $i$. Given budget $B \geq 0$, we define the cutoff $m = \max\{j : \sum_{i=1}^{j} c_i \leq B\}$. For $i \leq m$, adaptation proceeds; for $i > m$, the model freezes at state $\theta_m$ and inference continues without updates. This models deployments where users require results within a time envelope but don't penalise intermediate latency.

**Metric.** Let $\beta = m/N$ denote the adapted fraction. Amortised utility is formally:

$$U_{\text{amortised}} = \beta \cdot \bar{a}_{\text{adapt}} + (1 - \beta) \cdot \bar{a}_{\text{frozen}} \quad (3)$$

where $\bar{a}_{\text{adapt}}$ and $\bar{a}_{\text{frozen}}$ are mean accuracies over adapted and frozen phases. The decomposition reveals whether adaptation gains persist after budget exhaustion: $\bar{a}_{\text{frozen}} < a_0$ indicates adaptation-induced drift that harms post-freeze performance; $\bar{a}_{\text{frozen}} > a_0$ indicates adaptation contributed durable improvements by amortising the initial overhead. Not all methods are stateful or exhaust the budget; comparison across methods occurs via Pareto efficiency over varying $B$, identifying which method delivers maximum value at each budget and where returns diminish.

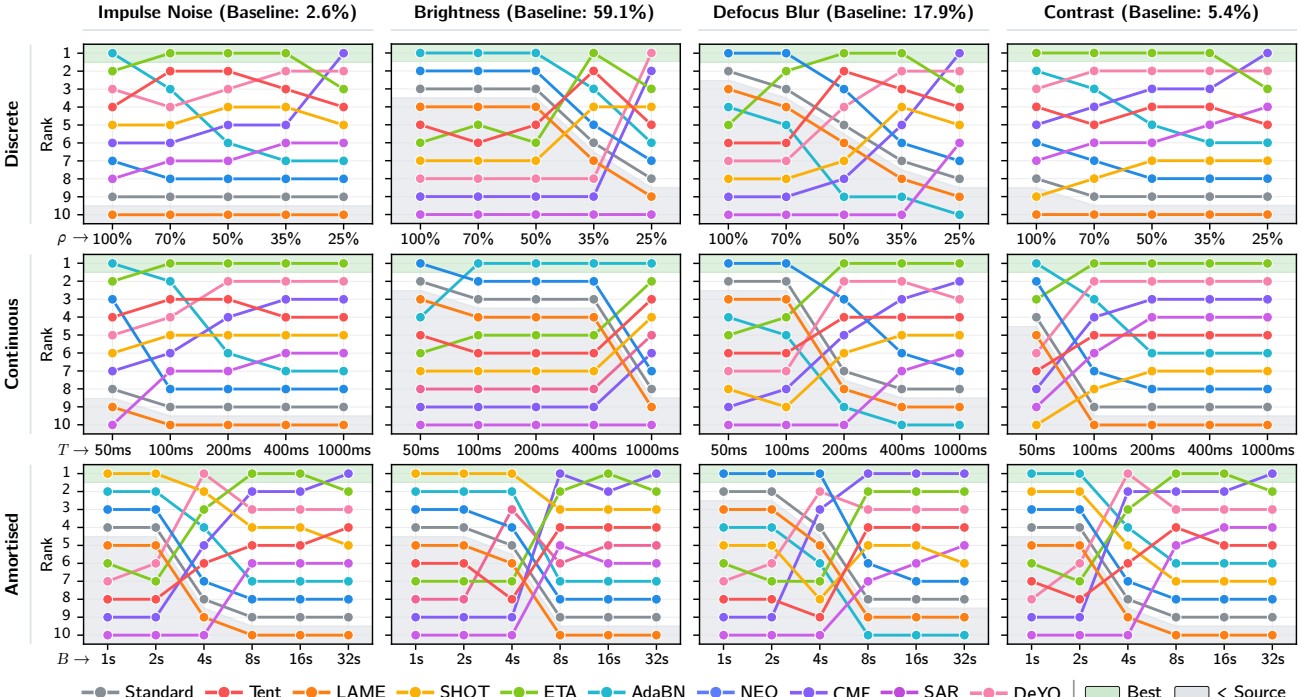

*Figure 3.* **Rank instability persists across temporal scenarios and corruption types.** Rows show discrete (utilisation $\rho$), continuous (threshold $T$), and amortised (budget $B$) evaluations; columns show one corruption from each category, spanning baseline accuracies from 2.6% to 59.1%. Green regions mark the best method; grey regions mark methods worse than standard inference. At relaxed thresholds, rankings converge towards offline; under pressure, instability increases. No method dominates. Best viewed in colour.

*Table 1.* Offline accuracy (%) with ranks for the four corruptions in Figure 5. CMF ranks first across 10 of 15; DeYO ranks first across the remaining five. We only show four for brevity.

| | Corruptions | | | |
|---|---|---|---|---|
| **Method** | Impulse | Brightness | Defocus | Contrast |
| Standard | 2.64 (9) | 59.13 (9) | 17.91 (8) | 5.38 (9) |
| NEO | 5.18 (8) | 60.27 (8) | 21.11 (7) | 8.07 (8) |
| LAME | 2.24 (10) | 58.74 (10) | 17.63 (9) | 5.15 (10) |
| AdaBN | 16.67 (7) | 65.33 (7) | 15.10 (10) | 16.87 (6) |
| Tent | 31.27 (5) | 67.47 (5) | 27.72 (5) | 26.34 (5) |
| ETA | 38.18 (3) | 67.85 (3) | 33.19 (3) | 45.63 (3) |
| SHOT-IM | 30.81 (6) | 67.62 (4) | 27.31 (6) | 13.15 (7) |
| DeYO | 38.36 (2) | **67.98** (1) | 33.79 (2) | 46.21 (2) |
| CMF | **39.31** (1) | 67.97 (2) | **34.91** (1) | **47.17** (1) |
| SAR | 32.58 (4) | 67.41 (6) | 29.10 (4) | 38.26 (4) |

## 4. Time-Contingent Evaluation

We evaluate nine[3] Fully TTA methods: AdaBN (Li et al., 2017), LAME (Boudiaf et al., 2022), NEO (Murphy et al., 2025), Tent (Wang et al., 2021), ETA (Niu et al., 2022), SHOT-IM (Liang et al., 2020), CMF (Lee & Chang, 2024), DeYO (Lee et al., 2024b), and SAR (Niu et al., 2023); the first three are gradient-free, and the last six are gradient-

---

[3]We include SPA (Niu et al., 2025) and ZeroSIAM (Chen et al., 2026) for our evaluation on ViT-B/16 in Appendix C.

based. Standard inference is the non-adaptive baseline.

Following Niu et al. (2022) and Alfarra et al. (2024), we evaluate on ImageNet-C (Hendrycks & Dietterich, 2019) at severity level 5, which comprises 15 corruption types across four categories (noise, blur, weather, digital) applied to the ImageNet validation set (Russakovsky et al., 2015). Evaluation is *episodic*, with methods reset between corruptions and batches sampled i.i.d. within each; all methods use their supplied default hyperparameters.

We use an ImageNet-pretrained ResNet-50 (He et al., 2016) backbone with batch size 64 and evaluate on an Nvidia RTX 4080 GPU, where standard inference processes batches in $\approx 38.67$ ms. We set $\lambda = 39.9$ ms using a $6\sigma$ provision, targeting five-nines availability common in production systems. We report results on a single seed (2025).

### 4.1. Evaluation Setup

We evaluate across 16 temporal scenarios spanning the three protocols of §3, each grounded in deployment realities. Relaxing temporal pressure recovers offline rankings (Table 1), establishing a continuum to unconstrained evaluation. For brevity, both this table and Figure 5 focus on four representative corruptions. Across all 15 types, this yields 240 evaluations per method; extended results appear in Appendix C.

*Table 2.* Per-batch overhead decomposition ($\lambda = 39.9$ ms; notation per §3), averaged across 11,715 batches. The last six methods are gradient-based. Timings vary by chosen hardware platform.

| Method | Latency Breakdown (ms) | | | | Slowdown |
| | $\bar{\delta}$ | $\bar{e}$ | $\bar{e} - \lambda$ | $\bar{\ell}$ | $\bar{\delta}/\lambda$ |
| --- | --- | --- | --- | --- | --- |
| Standard | 38.7 | 38.7 | 0.0 | 0.0 | 0.97× |
| NEO | 38.8 | 38.7 | 0.0 | 0.0 | 0.97× |
| LAME | 40.3 | 40.2 | 0.3 | 0.0 | 1.01× |
| AdaBN | 41.1 | 41.0 | 1.1 | 0.0 | 1.03× |
| Tent | 97.1 | 41.1 | 1.2 | 56.1 | 2.43× |
| ETA | 97.7 | 41.1 | 1.2 | 56.6 | 2.45× |
| SHOT-IM | 121.0 | 41.1 | 1.2 | 79.8 | 3.03× |
| DeYO | 129.7 | 41.1 | 1.2 | 88.6 | 3.25× |
| CMF | 160.1 | 41.1 | 1.2 | 119.0 | 4.01× |
| SAR | 195.2 | 41.1 | 1.2 | 154.1 | 4.89× |

*Table 3.* Discrete utility decomposition at $\rho = 100\%$, aggregated across 15 corruptions (11,715 batches). Availability $\alpha$, the fraction of batches served, determines ranking despite gradient-based methods achieving higher served accuracy.

| Method | $\alpha$ (%) | $|\mathcal{Q}|$ | $\bar{a}_{\text{served}}$ (%) | $U_{\text{discrete}}$ (%) |
| --- | --- | --- | --- | --- |
| Standard | 100.0 | 11,715 | 18.2 | 18.2 |
| NEO | 100.0 | 11,715 | 22.1 | 22.1 |
| LAME | 98.8 | 11,579 | 17.5 | 17.3 |
| AdaBN | 97.2 | 11,388 | 31.7 | **30.8** |
| Tent | 41.2 | 4,830 | 40.3 | 16.6 |
| ETA | 41.0 | 4,800 | 45.6 | 18.7 |
| SHOT-IM | 33.2 | 3,885 | 40.6 | 13.5 |
| DeYO | 31.8 | 3,722 | 45.0 | 14.3 |
| CMF | 25.1 | 2,943 | **46.0** | 11.5 |
| SAR | 20.6 | 2,415 | 37.6 | 7.8 |

**Discrete Utility.** Utilisation $\rho = \lambda/\gamma$ controls how tightly the inter-arrival time $\gamma$ is coupled to the baseline latency $\lambda$. We evaluate at $\rho \in \{100, 70, 50, 35, 25\}\%$, corresponding to $\gamma \in \{39.9, 56.4, 79.8, 112.8, 159.6\}$ ms. This follows $\sqrt{2}$ geometric spacing for uniform logarithmic coverage. These values map to practical deployment capacities. Consider, for instance, an in-store video surveillance system processing 30 fps streams with a batch size of 64. At $\rho = 100\%$, the saturated pipeline processes 1,604 samples per second, supporting approximately 53 cameras with no slack for adaptation; at $\rho = 25\%$, capacity drops to 13 cameras but provides 4× headroom. The spacing captures this trade-off between system capacity and adaptation tolerance.

**Continuous Utility**. Threshold $T$, which we ground in HCI research on perceived responsiveness (Nielsen, 1994; Card et al., 1983), sets the half-life for prediction value: utility halves when total response time reaches $T$; adaptation overhead up to $T - \lambda$ retains at least half the initial accuracy. We evaluate at $T \in \{50, 100, 200, 400, 1000\}$ ms, corresponding to overhead tolerances of $\approx \{10, 60, 160, 360, 960\}$ ms. Researchers with different $\lambda$ can anchor $T$ to their system's responsiveness constraints. Consider an on-device photo organiser that auto-tags images during bulk upload: at $T = 50$ ms, users expect instant tagging and tolerate negligible overhead; at $T = 1000$ ms, users accept a brief pause for improved accuracy. Varying $T$ models user or downstream expectations of system responsiveness, from latency-critical interfaces to tolerant background tasks.

**Amortised Utility.** Budget $B$ caps cumulative adaptation overhead; once exhausted, the model freezes for the remaining batches. With $N = 781$ batches per corruption, processing with standard inference takes $\lambda N \approx 31$ s. We evaluate $B \in \{1, 2, 4, 8, 16, 32\}$ s, spanning 3% to 105% of this baseline in power-of-two increments; researchers can scale $B$ to their deployment's overhead tolerance. Consider a drone performing visual crop inspection that continuously captures and batches images for onboard classification; its

flight time and onboard compute are battery-constrained. At $B = 1$ s, minimal adaptation preserves resources to maximise field coverage. At $B = 32$ s, more adaptation possibly restricts this coverage. The budget reflects resource commitment, and our evaluation reveals how much was consumed and whether it yielded gains over standard inference.

### 4.2. Rank Instability

Figure 5 reveals that rankings are unstable across temporal scenarios and corruption types. CMF, the majority offline winner (10/15), achieves the first rank in only 29 of 240 evaluations (12.1%); upon losing, it yields 15% utility on average to the winner, and underperforms standard inference in 79 cases (32.9%). In contrast, ETA, which wins none of the offline corruptions, is the majority temporal winner with 103 of 240 evaluations (42.9%) in its favour; it mainly wins in mildly relaxed settings (35% $\leq \rho \leq$ 70%, $T \geq 100$ ms, and $8 \leq B \leq 16$ s). This instability reveals how varying the resource constraints dictates *when* specific methods succeed.

Table 2 decomposes per-batch latency to reveal the root cause: intrinsic overhead $\bar{e} - \lambda$ from modified forward passes is negligible across all methods (0–1.2 ms), while extrinsic overhead $\bar{\ell}$ from backpropagation dominates for gradient-based methods at 56–154 ms, yielding 2.4–4.9× slowdowns. This cost, invisible to offline evaluation, determines whether a method can keep pace with temporal pressure.

**Method Selection.** Instability varies across corruptions. Under $T = 50$ ms, Spearman correlation with the offline ranking ranges from $r_s = -0.75$ (brightness) to $r_s = 0.12$ (impulse noise); under $\rho = 100\%$, from $r_s = -0.71$ (frost) to $r_s = 0.66$ (contrast). Per-corruption rankings do not follow a fixed pattern across scenarios; temporal pressure triggers the instability while corruption characteristics modulate its severity. As pressure relaxes, offline rankings become more predictive: under discrete evaluation, mean correlation rises from $r_s = -0.19$ at $\rho = 100\%$ to $r_s = 0.97$ at

*Table 4.* Continuous utility decomposition at $T = 50$ ms. Responsiveness $\bar{\kappa}$ determines ranking; covariance near zero indicates accuracy and responsiveness are approximately independent.

| Method | $\bar{a}$ (%) | $\bar{\kappa}$ (%) | Cov($a, \kappa$) | $U_{\text{continuous}}$ (%) |
|---|---|---|---|---|
| Standard | 18.16 | 100.0 | 0.0000 | 18.16 |
| NEO | 22.14 | 100.0 | 0.0000 | 22.14 |
| LAME | 17.40 | 95.7 | 0.0001 | 16.96 |
| AdaBN | 31.72 | 89.6 | 0.0000 | **28.42** |
| Tent | 42.88 | 15.1 | $-0.0001$ | 6.46 |
| ETA | 48.35 | 15.0 | $-0.0002$ | 7.22 |
| SHOT-IM | 42.43 | 11.2 | $-0.0001$ | 4.73 |
| DeYO | 48.76 | 10.2 | $-0.0003$ | 4.92 |
| CMF | **49.13** | 7.9 | $-0.0002$ | 3.84 |
| SAR | 44.14 | 6.2 | $-0.0001$ | 2.73 |

*Table 5.* Amortised utility decomposition at $B = 1$ s. For gradient-based methods, frozen accuracy determines rankings. For gradient-free methods, adapt accuracy determines ranking.

| Method | $\beta$ (%) | $\bar{a}_{\text{adapt}}$ (%) | $\bar{a}_{\text{frozen}}$ (%) | $U_{\text{amortised}}$ (%) |
|---|---|---|---|---|
| Standard | 100.0 | 18.16 | - | 18.16 |
| NEO | 100.0 | 22.14 | - | 22.14 |
| LAME | 100.0 | 17.40 | - | 17.40 |
| AdaBN | 100.0 | 31.72 | - | 31.72 |
| Tent | 2.3 | 32.11 | 0.10 | 0.84 |
| ETA | 2.3 | 33.37 | 0.10 | 0.87 |
| SHOT-IM | 1.7 | 32.23 | 32.22 | **32.22** |
| DeYO | 6.7 | 8.66 | 0.10 | 0.67 |
| CMF | 1.7 | 21.91 | 0.10 | 0.48 |
| SAR | 0.9 | 33.54 | 0.10 | 0.40 |

$\rho = 25\%$; under amortised evaluation, from $r_s = -0.51$ at $B = 2$ s to $r_s = 0.44$ at $B = 4$ s. While specific thresholds vary with model and dataset, practitioners should validate method selection under their deployment's temporal constraints rather than relying on offline rankings alone.

### 4.3. Trade-Off Analysis

Utility decompositions reveal protocol-specific bottlenecks that explain *why* rankings are unstable. We analyse the strictest temporal scenarios for each protocol ($\rho = 100\%$, $T = 50$ ms, $B = 1$ s), where instability is most pronounced. Decompositions shown aggregate across all 15 corruptions.

Table 3 shows discrete utility following Equation (1) at $\rho = 100\%$. ETA achieves the second-highest served accuracy (45.6%) but serves only 41% of batches; the product yields 18.7% utility. Meanwhile, AdaBN's served accuracy is lower (31.7%) but its 97.2% availability yields 30.8% utility, a 12.1% utility gain over ETA. Availability places a ceiling on a method's achievable utility; even at 100% served accuracy, serving $\alpha$ fraction of batches achieves at most $\alpha$ utility. When this ceiling falls below a competitor's utility, no accuracy gain can compensate; we term this state *computational insolvency*. To match AdaBN, ETA would need 75.1% served accuracy, far exceeding any observed TTA performance on ImageNet-C; to do the same, SAR would need 149.5%, a mathematical impossibility.

Table 4 shows continuous utility following Equation (2) at $T = 50$ ms. Unlike availability, responsiveness imposes a multiplicative penalty rather than a ceiling; every prediction is served, but late predictions lose value. ETA's 48.4% accuracy is discounted by 85% to yield just 7.2% utility. AdaBN's 90% responsiveness preserves most of its 31.7% accuracy, yielding 28.4% utility. To close this gap, ETA would require 189% accuracy, signalling insolvency by multiplication rather than exclusion; the insolvency threshold varies by corruption. On brightness at $T = 50$ ms, where baseline accuracy is already 59.1%, standard inference outperforms all gradient-based methods; the overhead cannot

be justified when return on compute is limited.

Table 5 shows amortised utility following Equation (3) at $B = 1$ s. Per corruption, gradient-based methods exhaust their budget within the first $\sim$20 batches, freezing for the remaining 97%; what happens after freezing determines utility. All but one gradient-based method collapses to 0.1% frozen accuracy, dragging overall utility below 1%, $20\times$ worse than standard inference. Insolvency here takes the form of harmful adaptation: overhead spent with net negative gains.

SHOT-IM is an outlier to this trend: despite its higher per-batch overhead than ETA (79.8 ms vs 56.6 ms), it maintains 32.2% frozen accuracy, matching its adapted accuracy and yielding the highest utility among gradient-based methods. This resilience is invisible to offline evaluation and stems from its retention of source running statistics, providing a robust distribution estimate at the point of freezing. Conversely, other methods degenerate as their accumulated running statistics are not sufficiently mature. We expand on this with an ablation in Appendix C.

**Method design.** These decompositions reveal three failure modes: availability collapse, accuracy erosion, and harmful adaptation; all stem from overhead not justified by accuracy gains under temporal pressure. Addressing these failures requires three properties absent from current methods, which we view as logically necessary for deployable adaptation.

First, *corruption-conditioned compute allocation*: corruptions perturb features differently, yet current methods apply fixed mechanisms with uniform overhead; brightness shifts all pixels uniformly and corrects cheaply, while Gaussian noise perturbs pixels independently, requiring more compute to recover structure. A method that scales the overhead accordingly would balance per-corruption trade-offs such that the return on compute remains effective across the board. However, estimating corruption difficulty without supervision remains an open question.

Second, *time-aware scaling*: methods must bound response delay to promptly conclude adaptation before deadlines or

discounting erases gains. Doing so protects availability and responsiveness from collapsing under strict temporal pressure; methods with negligible overhead satisfy it vacuously.

Third, *anytime performance*: elapsed overhead should yield gains over standard inference regardless of when adaptation halts. Guaranteeing this ensures model quality under adaptive inference stays above that of a non-adaptive baseline. Methods that degrade before improving, as most gradient-based ones do under amortised evaluation, risk leaving predictions worse than if adaptation had never begun.

While different evaluation setups may alter the severity of the failure modes for specific methods, the underlying trade-off remains. Thus, these properties are universally applicable wherever a model must adapt under time constraints.

## 5. Discussion

Our evaluation is scoped to the canonical TTA context of image classification on ImageNet-C with ResNet-50 to enable direct comparison with prior work. Even in this well-studied benchmark, Tempora reveals the persistence of rank instability across a range of realistic deployment scenarios. While distinct modalities (text, audio), tasks (segmentation, translation), domains (wildlife monitoring, medical imaging), architectures (vision transformers, mobile-optimised networks), and hardware (single-board computers, microcontrollers) entail unique trade-offs, the underlying need to reconcile adaptation overhead with temporal constraints remains universal, as do the properties we identify in §4.3.

This generality is only actionable if the evaluation is parameterised around physically-interpretable constraints that practitioners control. Each scenario in Tempora is anchored to such a constraint: (1) for *discrete*, system utilisation $\rho = \lambda/\gamma$, which controls the per-batch headroom for adaptation, is derived using a model's baseline latency $\lambda$ and a stream's inter-arrival time $\gamma$; (2) for *continuous*, threshold $T$ is anchored to an application's user responsiveness expectations or downstream service-level objectives; (3) for *amortised*, budget $B$ defines the cumulative wall-clock overhead that a deployment can absorb. Practitioners can then use the decomposed metrics as diagnostic tools to identify failure modes and acceptable trade-offs in their use case.

To show the generality of our observations, we expand on our primary setup in Appendix C to cover more models (ViT-B/16, ResNet-18), datasets (ImageNet-V2/R, CIFAR-10-C), methods (SPA, ZeroSIAM) and hardware (Raspberry Pi 5 16GB). Crucially, rank instability persists across all these choices. For instance, on ImageNet-C with ViT-B/16, NEO wins a majority of temporal evaluations (81 of 225) despite never winning offline. Also, across hardware, the winners distribution shifts because gradient-based methods are less optimised on CPUs. This variation is acceptable, as rankings

are context-dependent and should not be generalised beyond the hardware and setting in which they were obtained.

**Limitations.** We evaluate nine Fully TTA methods, balancing coverage with tractable overhead. This rules out methods such as DDA (Gao et al., 2023) and MEMO (Zhang et al., 2022), whose incremental gains are unlikely to offset their notable overhead under temporal pressure. We also exclude wrapper-like methods (Zhao et al., 2023a; Zhang et al., 2025; Kim et al., 2025), which would combinatorially expand the evaluation space. Beyond these exclusions, our episodic, i.i.d. evaluation affects method fit: LAME, designed for non-i.i.d. streams, underperforms standard inference in all 240 cases, illustrating that method-evaluation mismatch can render adaptation counterproductive.

This mismatch is also governed by hyperparameters. We report results on a single seed and use the default hyperparameters for each method. While we share Alfarra et al.'s observation that seed variation is negligible, hyperparameter influence on rankings remains a broader epistemic concern in TTA design and evaluation. Nevertheless, for methods approaching computational insolvency, the temporal component determines utility; this places a ceiling that current hyperparameters cannot lift. As such, there may be method ranking pairs that tuning cannot simply flip.

**Future Work.** Such mismatches motivate evaluating methods in contexts suited to their design. Integrating Tempora with efforts in distributional realism (§2) provides fairer settings for methods targeting continual, non-stationary streams (Wang et al., 2022; Yuan et al., 2023; Song et al., 2023; Hong et al., 2023); this could model complex deployments and inform new application-specific methods. Moreover, while we present each utility metric for a distinct scenario, they can be combined to model richer temporal dynamics; a battery-operated person detector, where visitors arrive asynchronously and notifications must be prompt, would require all three utility metrics.

## 6. Conclusion

We introduced *Tempora*, a framework for evaluating TTA methods under temporal pressure, comprising three metrics that characterise time-contingent utility under distinct deployment constraints. Each decomposes into interpretable factors revealing why methods succeed or fail. Across 750+ temporal evaluations, we find rank instability is pervasive: offline rankings do not predict rankings under temporal pressure. Utility decompositions expose three failure modes stemming from overhead that accuracy gains cannot justify. Addressing these to realise deployable adaptation requires corruption-conditioned overhead allocation, time-aware scaling, and anytime performance. Tempora provides the lens to evaluate progress toward this goal.

## Acknowledgements

This work is supported by Nokia Bell Labs through a donation and by EPSRC through grant EP/Z53447X/1.

## Impact Statement

This paper promotes more realistic evaluation of test-time adaptation by accounting for a range of latency constraints that arise in real-world deployments. By exposing how conventional rankings can fail under temporal pressure, *Tempora* helps practitioners select methods suited to application-specific time budgets and provides researchers with design targets for deployable adaptation. The proposed evaluation framework does not introduce new adaptive capabilities, but rather improves transparency around existing methods, supporting more responsible and deployment-aware use of adaptable machine learning systems.

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

# A. Notation & Terminology

Table 6 summarises the notation used throughout the paper. Table 7 defines key terms.

The terms "*online*" and "*offline*" are overloaded in the TTA literature. Alfarra et al. (2024) use online to denote evaluation under temporal constraints, *i.e.,* live, real-time processing, and offline to denote relaxed evaluation without such constraints. We adopt this interpretation of offline in our work. Our use of online, however, refers to sequential evaluation where the model must commit to a prediction on the current batch before observing subsequent batches; this interpretation appears in Wang et al. (2021). A third setting, which one might call "*fully offline*," provides access to the entire target domain before any predictions are required. SHOT-IM (Liang et al., 2020) and AdaBN (Li et al., 2017) were initially developed for this setting; the generality of their adaptation mechanisms, detailed in §C.1, affords transferability to the online setting we evaluate in.

*Table 6.* Mathematical notation grouped by context.

| Symbol | Description | Symbol | Description |
|---|---|---|---|
| *General* | | *Continuous Utility* | |
| $N$ | Total number of batches | $e_i$ | Intrinsic time (pickup to emission) |
| $\mathbf{x}_i$ | Input batch $i$ | $\ell_i$ | Extrinsic time (emission to next pickup) |
| $\mathbf{y}_i$ | Predictions for batch $i$ | $w_i$ | User wait time ($\ell_{i-1} + e_i$) |
| $t_i$ | Arrival time of batch $i$ | $d_i$ | Effective response delay |
| $\delta_i$ | Processing latency for batch $i$ | $\kappa_i$ | Decay factor (responsiveness) |
| $a_i$ | Accuracy on batch $i$ | $T$ | HCI threshold |
| $a_0$ | Standard inference accuracy | $\bar{\kappa}$ | Mean responsiveness |
| $\lambda$ | Standard inference latency | $\mathrm{Cov}(a, \kappa)$ | Accuracy-responsiveness alignment |
| $\theta$ | Model parameters | | |
| $r_s$ | Spearman rank correlation | | |
| *Discrete Utility* | | *Amortised Utility* | |
| $\gamma$ | Inter-arrival time | $c_i$ | Adaptation overhead ($\delta_i - \lambda$) |
| $\rho$ | Pipeline utilisation ($\lambda/\gamma$) | $B$ | Overhead budget |
| $\mathcal{Q}$ | Set of served batch indices | $m$ | Cutoff batch index (budget exhaustion) |
| $\alpha$ | Availability ($|\mathcal{Q}|/N$) | $\beta$ | Adapted fraction ($m/N$) |
| $\bar{a}_{\mathrm{served}}$ ($\bar{a}_{\mathrm{s}}$) | Mean accuracy (served batches) | $\bar{a}_{\mathrm{adapt}}$ ($\bar{a}_{\mathrm{a}}$) | Mean accuracy (adaptation phase) |
| $(s_j, f_j)$ | Start/finish times of event $j$ | $\bar{a}_{\mathrm{frozen}}$ ($\bar{a}_{\mathrm{f}}$) | Mean accuracy (frozen phase) |
| $p_j$ | Batch index served at event $j$ | $\theta_m$ | Frozen model state after budget exhaustion |

*Table 7.* Key terminology introduced in this paper.

| Term | Definition |
|---|---|
| Fully TTA | Test-time adaptation using only pre-trained parameters $\theta$ and unlabelled target samples $x_i \sim \mathcal{T}$, without access to source training data or supervision. |
| Episodic evaluation | Each corruption is treated as an independent episode, with the model and adaptation mechanism reset to its original state between corruptions. |
| Continual evaluation | Adaptation is cumulative across all corruptions; any updates made persists across the continuous stream. Ordering of corruptions may vary. |
| i.i.d. (samples) | Samples within a batch are drawn independently and identically from the target domain $\mathcal{T}$; no structure in class ordering or temporal dependencies. |
| Availability | Fraction of batches served under discrete evaluation ($\alpha$); more generally, whether a model pipeline or system is ready to serve requests. |
| Intrinsic overhead | Added latency from a modified forward pass ($e_i - \lambda$); delays the immediate response. Arises from operations such as computing batch statistics. |
| Extrinsic overhead | Post-prediction computation ($\ell_i$); stalls the pipeline before next batch. |
| Responsiveness | Decay factor penalising late predictions under continuous evaluation ($\kappa$). |
| Computational insolvency | When a method's overhead cannot be justified by accuracy gains under temporal pressure, relative to standard inference or competing methods. |
| Rank instability | Phenomenon where method rankings change under temporal pressure relative to that observed in offline evaluation. |
| Anytime performance | Property where elapsed overhead yields gains over standard inference regardless of when adaptation halts. |

## B. Discrete Protocol Design

This section expands on Tempora's discrete protocol (§3.1) and its relationship with prior work. We compare our buffered approach with Alfarra et al.'s protocol variants (Figure 4), justify design choices, and note implementation details. Although focused on the discrete temporal setting, some observations generalise to Tempora's broader framework.

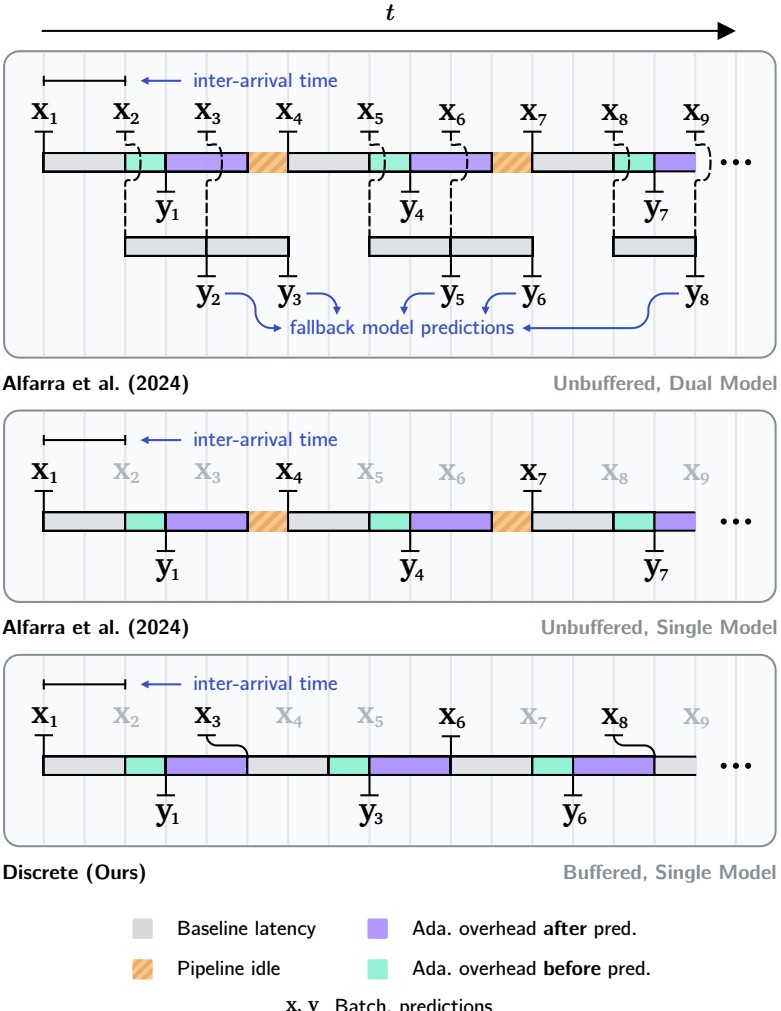

*Figure 4.* **Comparison of discrete evaluation protocols.** *Top:* Alfarra et al.'s dual-model variant uses a fallback model pipeline that serves predictions for batches arriving while the primary model adapts; the dashed lines indicate these batches redirected to the second pipeline. This model may be periodically updated with the adapted state. The setup assumes zero-cost parallel model execution. *Middle:* Their single-model variant skips missed batches and forces the pipeline to idle until the next arrival. *Bottom:* Our buffered protocol retains the most recent batch, serving it when the pipeline is free. This avoids fallback assumptions and forced idling while exposing availability as the cost of slow adaptation. The legend largely remains the same as in Figure 2, with the inclusion of pipeline idling (yellow).

### B.1. Design Decisions & Relationship to Prior Work

Our discrete *scenario* is conceptually similar to that proposed by Alfarra et al. (2024). Both evaluate adaptation methods under a response deadline imposed by an asynchronous data stream and penalise slow adaptation with fewer batches (§2). Unadapted batches are deemed skipped. In their *dual-model* protocol, the centrepiece of their work, these "skipped" batches receive predictions from a fallback inference model, which may be periodically updated with the adapted state; given arrival interval $\gamma = 40\,\text{ms}$ and adaptation time $\delta_i = 75\,\text{ms}$, every other batch receives a prediction from the fallback. This approach guarantees near-100% system availability, barring resource contention from parallel model pipelines. However, this setting imposes no constraint on the timeliness of adaptation, which operates as a secondary, passive process; adaptation affects task performance (*accuracy*), not system performance (*availability*), operating under a safety net provided by the fallback model.

The primary evaluative objective thus reduces to maximising accuracy as in offline evaluation, since system availability is guaranteed by construction. However, system availability is distinct from method availability, *i.e.* the proportion of batches adapted on over the stream; under the *single-model* protocol, both are the same. Both Ghunaim et al. (2023) and Alfarra et al. (2024) predetermine method availability. In the former, stream-model relative complexity ($C_s$) is an integer ratio of an online continual learning (OCL) method's computational cost (in FLOPs) to that of a baseline OCL method. They compute this ratio once upfront, as each considered method has a fixed cost per update step. Alfarra et al. introduce an analogous concept for online TTA termed "relative adaptation speed of g", which is also an integer ratio, this time of the method speed ($\delta$, in ms) to that of the stream ($\gamma$); they fix the latter to the latency of a non-adaptive baseline model. Here, the ratio is recomputed after every adaptation step to account for the variable per-batch timing of some adaptation methods (*e.g.,* ETA, SAR, etc.). Nevertheless, these integer ratios are used as *inputs* to predetermine, at each update step, how many subsequent batches to skip. Neither work tracks skipped batches as an aggregate *outcome* (method availability) over the stream.

Our protocol is based on the *single-model* variant briefly noted in Alfarra et al.'s appendix. We do so for two reasons. First, system performance is no longer shielded from adaptation overhead: skipped batches are discarded[4], which caps peak achievable utility (§4.3). Second, single-model pipelines are common in edge deployments, which host many real-world ML applications but are resource-constrained. While removing the safety net surfaces method availability as an observable quantity, rigid synchronisation (§2) persists, forcing the adaptation pipeline to idle on incoming batches (Figure 4).

Our view is that an evaluation protocol should remain agnostic to adaptation details where possible, presenting batches or streams with delivery constraints and letting methods decide their response. This is not to say protocols should impose no constraints; deployment realities demand some. However, these constraints should be well-motivated. Forced idling appears to be an unintended side effect of the ceiling operation in the penalty mechanism; as Alfarra et al.'s analysis does not report availability, this subtlety may not have been apparent. While response delay or stale adaptation could justify idling in specific deployments, these are not general evaluation concerns.

*Table 8.* Effect of buffering on availability and utility for ResNet-50 on ImageNet-C ($\lambda = 39.9$ ms). Unbuffered (Ub) is our recontextualised variant of Alfarra et al.'s protocol. Buffered (Buf) is our single-slot queue protocol at $\rho = 100\%$. $\alpha$ is the fraction of arriving samples served. $U_d = (\alpha/100) \cdot \bar{a}_s$ is system-level utility. $\Delta$ is buffered minus unbuffered (percentage points). Methods are ranked by $U_d$; **bold** rank indicates improvement.

| Method | Availability $\alpha$ (%) | | | Utility $U_d$ (%) | | | Rank |
| | Ub | Buf | $\Delta$ | Ub | Buf | $\Delta$ | Ub→Buf |
|---|---|---|---|---|---|---|---|
| Standard | 100.0 | 100.0 | +0.0 | 18.2 | 18.2 | +0.0 | 2→4 |
| AdaBN | 50.1 | 97.2 | +47.1 | 15.9 | **30.8** | +14.9 | **3→1** |
| LAME | 66.8 | 98.8 | +32.0 | 13.0 | 17.3 | +4.3 | **6→5** |
| NEO | 100.0 | 100.0 | +0.0 | **22.1** | 22.1 | +0.0 | 1→2 |
| Tent | 33.4 | 41.2 | +7.8 | 13.1 | 16.6 | +3.6 | 5→6 |
| ETA | 33.4 | 41.0 | +7.6 | 15.0 | 18.7 | +3.7 | **4→3** |
| SHOT-IM | 25.1 | 33.2 | +8.1 | 9.9 | 13.5 | +3.6 | **9→8** |
| DeYO | 26.3 | 31.8 | +5.5 | 11.7 | 14.3 | +2.6 | 7→7 |
| CMF | 22.2 | 25.1 | +2.9 | 10.1 | 11.5 | +1.4 | 8→9 |
| SAR | 20.1 | 20.6 | +0.5 | 7.5 | 7.8 | +0.2 | 10→10 |

*Table 9.* Timing for ResNet-50 on ImageNet-C ($\lambda = 39.9$ ms). Waitspan is mean forced idling time (s) for unbuffered (Ub); always zero for buffered (Buf). Response time is sum of mean prediction time and queue wait per batch (ms). Queue wait is zero for Ub and bounded by $\lambda$ for Buf. All metrics averaged across corruptions.

| Method | Waitspan (s) | Response Time (ms) | |
| | | Ub | Buf |
|---|---|---|---|
| Standard | 0.0 | 38.7 ± 0.0 | 38.7 ± 0.0 |
| AdaBN | 15.1 | 41.0 ± 0.1 | 60.8 ± 11.5 |
| LAME | 10.1 | 40.1 ± 0.9 | 53.6 ± 13.2 |
| NEO | 0.0 | 38.7 ± 0.0 | 38.7 ± 0.0 |
| Tent | 5.9 | 41.0 ± 0.0 | 61.1 ± 11.6 |
| ETA | 5.7 | 41.0 ± 0.0 | 61.0 ± 11.6 |
| SHOT-IM | 7.6 | 41.1 ± 0.0 | 60.5 ± 11.8 |
| DeYO | 5.4 | 41.0 ± 0.0 | 60.9 ± 12.1 |
| CMF | 3.5 | 41.0 ± 0.0 | 62.2 ± 13.1 |
| SAR | 0.7 | 41.0 ± 0.0 | 61.3 ± 12.2 |

**Buffer mechanics.** We introduce a single batch-sized buffer ($\approx 37$ MB for ImageNet) to keep the pipeline saturated. Table 8 highlights the impact of this change, particularly in its fair treatment of efficient methods. Notably, under Alfarra et al.'s protocol[5], AdaBN and ETA rank below the non-adaptive baseline mainly due to forced idling; AdaBN spends nearly half the evaluation time idling, as detailed in Table 9. The buffer corrects this; it improves AdaBN's availability by 47.1 pp, pushing it to rank highest in utility. It does prolong the mean response time as batches wait in the buffer, but this is acceptable as the discrete protocol imposes no timeliness constraint, which concerns the continuous protocol.

---

[4] Alfarra et al. (2024) assign random predictions to these skipped batches. This lower bounds accuracy to that of a blind classifier, which is 0.1% for ImageNet-1K and 10% for CIFAR-10. For many tasks, this approach may introduce an unacceptable level of operational risk due to potential false positives. In our protocol, we submit null predictions in such cases.

[5] To be clear, *unbuffered* is our revision of Alfarra et al.'s protocol that avoids model-relative proxies; we recontextualise their evaluation under our wall-time simulation where batch-skipping is not pre-determined. The revision includes some methodological corrections but is otherwise faithful to their intent.

We limit the buffer to a single batch because larger buffers offer diminishing returns under fixed arrival rates; they delay the onset of evictions but do not change the steady-state eviction rate. Availability may only improve due to the drain phase, where arrivals cease with the data stream. Larger buffers are meaningful in absorbing transient surges (bursty arrivals). Modelling variable-rate streams is a derivative temporal scenario that we leave to future work.

As mentioned, buffers introduce a queuing delay. Without one, batches are served immediately upon arrival. With a buffer of $b$ batches, a batch waits at most $b \cdot \gamma$ before being evicted or picked up for processing; for Tables 8 and 9, $b = 1$ and $\gamma = \lambda$, as we model utility under $\rho = 100\%$. The buffer size thus governs a trade-off: larger buffers absorb backlog and reduce evictions but increase response latency and data staleness. For our single batch-sized buffer, the maximum queuing delay is $\gamma$. Sizing beyond the utilisation factor ($b > \rho$) is wasteful as the buffer never saturates.

**Design targets.** Alfarra et al.'s focus lies in *whether* methods yield tangible returns when starved of batches under a shared, formalised penalty; ours lies in *why*, revealed through the accuracy-availability decomposition absent from their framing. When a method's wall-clock time $\delta_i$ fits within $\gamma$, the fallback only serves to add resilience. Under this framing, methods that keep pace with the stream are not distinctly rewarded beyond those that adapt periodically while skipping most batches. We view adaptation not as an auxiliary process but as a necessary evolution of inference. Alfarra et al.'s framing, while valuable for benchmarking sample starvation, does not surface design targets for this transition. Tempora's decompositions aim to provide them, facilitating deployable adaptation methods that might underlie adaptable intelligence at scale.

### B.2. Implementation Details

Although both ours and Alfarra et al.'s protocols describe asynchronous batch arrivals, neither operates on a real-time testbed. Instead, both evaluations simulate such data streams retrospectively. Alfarra et al. check the batch processing time $\delta_i$ and predetermine a fixed number of subsequent batches to skip. We instead formalise batch skipping implicitly through a forward-stepping discrete-event simulation that tracks a live pipeline completion timeline. Arriving batches are routed to a fixed-capacity queue that handles buffering and evictions. This approach generalises the streaming setup: by setting our queue capacity to zero, we recover Alfarra et al.'s unbuffered protocol within a unified event-loop framework.

**Timing methodology.** In our protocol, batch arrivals are exogenously determined (*e.g.,* a camera's frame rate). Our timing model is stable and independent from model-relative proxies. In contrast, Alfarra et al. (2024) measures the arrival interval dynamically by running the non-adaptive baseline immediately before *each* adaptation step. While intended to capture hardware variability, this approach unrealistically couples the arrival rate to per-batch measurement noise and introduces potential priming effects from back-to-back model execution.

While $\gamma$ is a free variable in our formulation, we initially bind it to a base interval $\lambda$, *i.e.,* the standard inference latency. We determine $\lambda$ via a rigorous offline sweep under a full evaluation setup, including five batches of warmup and uniform batch sizes throughout[6]. To conservatively estimate the per-batch completion time of a non-adaptive model, we use a $6\sigma$ provision over this sweep, setting $\lambda = \text{mean} + 6\sigma$ (§4). This differs from the raw per-batch latency that Alfarra et al. use.

Nevertheless, setting $\gamma = \lambda$ allows us to model a pre-existing inference system (anchor) where every batch is served by a non-adaptive baseline, representing 100% system utilisation ($\rho$). Systematic variation of the arrival interval via the relationship $\rho = \lambda/\gamma$ enables us to sweep operating points with varying headroom for adaptation relative to this anchor (§4.1). Since only the adaptive model runs on served batches, our protocol reduces total evaluation time while enabling practitioners to control $\gamma$ based on their setup.

**Batch normalisation states.** A subtle implementation detail affects frozen model behaviour. In PyTorch, batch normalisation (BN) layers operate in four modes; only one uses the running buffers to normalise the incoming batch, requiring the layer to be in evaluation mode and running buffers to exist. This mode canonically applies to frozen model inference. However, many TTA implementations clear the running buffers at setup, forcing the layers to recompute normalisation statistics on each batch even after the model is nominally frozen. This introduces overhead that unintentionally persists beyond adaptation. We provide a toggle to restore the intended frozen behaviour under amortised evaluation. For methods that modify BN layers, we track running statistics over the target domain to ensure valid buffers exist for frozen inference.

---

[6]We drop the last odd-sized batch, *i.e.,* 16 samples for ImageNet-C corruptions ($50,000/64 = 781.25$)

# C. Extended Time-Contingent Evaluation

This section expands upon our findings in §4. First, we provide detailed method descriptions and offline baselines, followed by corruption-aggregated sweeps across all temporal scenarios. Second, we consider per-corruption analysis, introducing a winners table spanning all 240 evaluations alongside utility decompositions under strict pressure. To demonstrate the generality of our observations, we report additional results on an extended suite of datasets, models, and hardware platforms. All evaluations use a single seed, with complete per-batch logs made available in our code repository.

## C.1. Method Overview

We evaluate 11 Fully TTA methods[7] spanning two families (gradient-free and gradient-based), with standard inference serving as a non-adaptive baseline. This section summarises each method's mechanism. Their hyperparameters follow original defaults and are specified in the supplementary code. Table 10 reports offline accuracy across all corruptions.

AdaBN (Li et al., 2017) recomputes normalisation statistics on the incoming batch and accumulates an exponential moving average of these statistics across batches. LAME (Boudiaf et al., 2022) only refines output probabilities via Laplacian-regularised optimisation; it constructs a $k$-nearest-neighbour affinity matrix in the feature space and iteratively adjusts class probabilities to balance closeness to the model's initial beliefs with consistency among neighbours. NEO (Murphy et al., 2025) maintains a running mean of penultimate-layer features and centres each batch by subtracting this mean before classification, correcting for feature-space drift.

*Table 10.* Offline accuracy (%) across all 15 ImageNet-C corruptions with ResNet-50.

| Corruption | Gradient-Free | | | | Gradient-Based | | | | | |
|---|---|---|---|---|---|---|---|---|---|---|
| | Standard | AdaBN | LAME | NEO | Tent | ETA | SHOT-IM | SAR | DeYO | CMF |
| Gaussian noise | 3.00 | 16.15 | 2.58 | 5.23 | 29.98 | 36.01 | 29.39 | 31.46 | 36.73 | **37.58** |
| Shot noise | 3.70 | 16.76 | 3.20 | 6.13 | 31.68 | 38.69 | 32.00 | 31.40 | 38.70 | **39.77** |
| Impulse noise | 2.64 | 16.67 | 2.24 | 5.18 | 31.27 | 38.18 | 30.81 | 32.58 | 38.36 | **39.31** |
| Defocus blur | 17.91 | 15.10 | 17.63 | 21.11 | 27.72 | 33.19 | 27.31 | 29.10 | 33.79 | **34.91** |
| Glass blur | 9.73 | 15.44 | 8.94 | 12.60 | 26.86 | 33.19 | 26.77 | 28.21 | 33.53 | **34.80** |
| Motion blur | 14.71 | 26.22 | 13.89 | 17.76 | 41.14 | 47.78 | 43.20 | 41.70 | **48.31** | 47.85 |
| Zoom blur | 22.46 | 38.90 | 21.87 | 26.57 | 49.26 | 52.74 | 50.44 | 49.23 | 52.81 | **53.08** |
| Snow | 16.60 | 34.18 | 15.23 | 21.72 | 47.21 | 52.09 | 49.11 | 47.23 | 52.53 | **52.63** |
| Frost | 23.06 | 33.11 | 22.30 | 27.67 | 41.15 | 45.99 | 41.49 | 42.47 | 46.24 | **46.79** |
| Fog | 24.01 | 47.83 | 22.23 | 30.57 | 57.56 | 60.03 | 57.84 | 57.64 | **60.53** | 60.32 |
| Brightness | 59.13 | 65.33 | 58.74 | 60.27 | 67.47 | 67.85 | 67.62 | 67.41 | **67.98** | 67.97 |
| Contrast | 5.38 | 16.87 | 5.15 | 8.07 | 26.34 | 45.63 | 13.15 | 38.26 | 46.21 | **47.17** |
| Elastic transform | 16.51 | 44.18 | 14.60 | 24.93 | 54.63 | 57.74 | 55.39 | 54.65 | **58.42** | 57.90 |
| Pixelate | 20.87 | 49.10 | 20.30 | 26.05 | 58.46 | 60.93 | 59.08 | 58.37 | **61.49** | 60.85 |
| JPEG compression | 32.64 | 39.99 | 32.12 | 38.26 | 52.47 | 55.22 | 52.90 | 52.46 | 55.77 | **55.96** |
| **Mean** | 18.16 | 31.72 | 17.40 | 22.14 | 42.88 | 48.35 | 42.43 | 44.14 | 48.76 | **49.13** |

Tent (Wang et al., 2021), which introduced the Fully TTA paradigm, minimises the entropy of predictions by updating affine BN parameters. ETA (Niu et al., 2022) extends Tent with two-stage sample filtering: a *reliability* filter excludes high-entropy samples that produce noisy gradients and a *redundancy* filter excludes samples whose softmax probabilities are similar to a moving mean of previously adapted samples' probabilities, avoiding repeated gradient signals. While only filtered samples inform updates, PyTorch backpropagates through the index mask and computes gradients for all samples regardless; this explains why ETA's latency does not improve despite filtering. SHOT-IM (Liang et al., 2020) minimises entropy while maximising prediction diversity to prevent class collapse; it updates the entire feature extractor.

SAR (Niu et al., 2023) uses sharpness-aware minimisation on entropy for stability; it first perturbs weights in the direction of the entropy gradient and computes an update at the perturbed point, encouraging convergence to flatter minima. It resets the model when the moving mean of batch entropy dips below a threshold and also filters high-entropy samples like ETA.

DeYO (Lee et al., 2024a) swaps out ETA's second-stage redundancy filter for a spatial grounding criterion. This new stage performs a patch-shuffle augmentation on each retained sample and runs a second no-gradient forward pass to measure PLPD (Prediction Loss under Patch Disruption), *i.e.,* the drop in predicted class probability when spatial layout is randomly

---

[7]SPA and ZeroSIAM are exclusive to our evaluation on ImageNet-C/R/V2 with ViT-B/16.

permuted. Samples insensitive to this disruption (low drop) are presumed texture-reliant and discarded; the survivors participate in the gradient update where confident and spatially grounded predictions are upweighted.

CMF (Lee & Chang, 2024) performs updates through a diversity- and certainty-weighted objective. Each batch is filtered to retain low entropy and diverse (*i.e.,* probability vector dissimilar to a running class-distribution estimate) samples. The filtered samples contribute a sigmoid-based entropy surrogate (SLR) and a cross-entropy consistency term between the original and a randomly augmented view of the input, requiring a second forward pass similar to DeYO. After each gradient step, a Kalman-style filter regularises the weight trajectory by maintaining a hidden model state that is first pulled towards the frozen source model (mean-reversion prior) and then updated towards the gradient-adapted model with a gain proportional to accumulated prediction variance; this prevents catastrophic drift without the explicit resets as in SAR.

SPA (Niu et al., 2025) enforces consistency between a clean view and two structure-preserving augmented views of each batch. One augmentation randomly masks low-frequency amplitude components in the Fourier domain; the other injects patch-wise Gaussian noise into the high-frequency signal, with per-patch noise magnitudes updates adversarially to maximise perturbation. Predictions on each augmented view are pushed towards those of the clean view via KL divergence; this is restricted to samples where the clean view is more confident, providing an implicit reliability filter.

ZeroSIAM (Chen et al., 2026) adapts both the affine normalisation parameters and a lightweight projector head by enforcing consistency between two views of each batch via entropy minimisation and symmetric KL divergence. The first forward pass produces clean backbone predictions; a second pass routes the same inputs through the projector, introducing a learned perturbation in the feature space. Entropy is minimised on the projector view to drive adaptation, while the symmetric KL between the clean and projector views prevents the projector from trivially collapsing. The latter serves a role analogous to the consistency terms used in SPA and CMF, which similarly use a second forward pass to stabilise their objectives.

## C.2. Corruption-Aggregated Parameter Sweeps

Tables 11–13 report utility across all parameter values for each protocol, aggregated across 15 corruptions. We omit standard deviation because it would reflect corruption difficulty rather than method consistency; the former varies widely, and corruption-level breakdowns appear in the following section. Latency variance is also minimal: across 11,715 batches, standard deviations are $< 1$ ms for gradient-free methods and $< 4$ ms for gradient-based methods, negligible relative to the 50–150 ms mean differences between method families.

*Table 11.* Discrete utility (%) across utilisation levels ($\rho$), aggregated across 15 corruptions.

| Method | $\rho = 100\%$ | $\rho = 70\%$ | $\rho = 50\%$ | $\rho = 35\%$ | $\rho = 25\%$ |
|---|---|---|---|---|---|
| Standard | 18.16 | 18.16 | 18.16 | 18.16 | 18.16 |
| NEO | 22.14 | 22.14 | 22.14 | 22.14 | 22.14 |
| LAME | 17.29 | 17.40 | 17.40 | 17.40 | 17.40 |
| AdaBN | **30.83** | **31.72** | 31.72 | 31.72 | 31.72 |
| Tent | 16.63 | 23.98 | 34.92 | 42.88 | 42.88 |
| ETA | 18.70 | 27.25 | **39.19** | **48.35** | 48.35 |
| SHOT-IM | 13.47 | 19.24 | 27.71 | 39.72 | 42.43 |
| DeYO | 14.29 | 20.53 | 29.38 | 42.01 | 48.76 |
| CMF | 11.55 | 16.65 | 23.94 | 34.28 | **48.90** |
| SAR | 7.75 | 11.32 | 16.62 | 24.23 | 35.48 |

*Table 12.* Continuous utility (%) across HCI thresholds ($T$), aggregated across 15 corruptions.

| Method | $T = 50$ms | $T = 100$ms | $T = 200$ms | $T = 400$ms | $T = 1000$ms |
|---|---|---|---|---|---|
| Standard | 18.16 | 18.16 | 18.16 | 18.16 | 18.16 |
| NEO | 22.14 | 22.14 | 22.14 | 22.14 | 22.14 |
| LAME | 16.96 | 17.32 | 17.37 | 17.39 | 17.40 |
| AdaBN | **28.42** | **31.11** | 31.49 | 31.62 | 31.68 |
| Tent | 6.46 | 21.98 | 31.60 | 37.00 | 40.47 |
| ETA | 7.22 | 24.66 | **35.53** | **41.67** | **45.61** |
| SHOT-IM | 4.73 | 18.09 | 28.18 | 34.64 | 39.13 |
| DeYO | 4.92 | 19.46 | 31.14 | 38.95 | 44.55 |
| CMF | 3.84 | 16.40 | 28.08 | 36.84 | 43.66 |
| SAR | 2.73 | 12.35 | 22.43 | 30.86 | 38.00 |

*Table 13.* Amortised utility (%) across overhead budgets ($B$), aggregated across 15 corruptions.

| Method | $B = 1$s | $B = 2$s | $B = 4$s | $B = 8$s | $B = 16$s | $B = 32$s |
|---|---|---|---|---|---|---|
| Standard | 18.16 | 18.16 | 18.16 | 18.16 | 18.16 | 18.16 |
| NEO | 22.14 | 22.14 | 22.14 | 22.14 | 22.14 | 22.14 |
| LAME | 17.40 | 17.40 | 17.40 | 17.40 | 17.40 | 17.40 |
| AdaBN | 31.72 | 31.72 | 31.72 | 31.72 | 31.72 | 31.72 |
| Tent | 0.84 | 1.56 | 24.05 | 40.60 | 42.55 | 43.12 |
| ETA | 0.87 | 1.68 | 29.23 | **46.85** | **48.30** | 48.56 |
| SHOT-IM | **32.22** | **35.26** | 37.24 | 40.52 | 42.07 | 42.75 |
| DeYO | 0.67 | 1.75 | **39.47** | 42.72 | 44.79 | 46.55 |
| CMF | 0.48 | 0.89 | 23.30 | 46.59 | 48.05 | **49.13** |
| SAR | 0.40 | 0.63 | 1.31 | 36.56 | 39.47 | 42.33 |

Table 14 summarises temporal decomposition terms under strict pressure; they are timing-derived and nearly constant across corruptions, with LAME being an exception due to its iterative optimisation converging at variable rates ($\alpha$: 96.9–100.0%, $\bar{\kappa}$: 89.6–99.8%). Under amortised evaluation ($B = 1$ s), gradient-based methods exhaust their budget within the first 7–18 batches of 781 total (<3% of the stream); this explains why frozen accuracy dominates overall utility.

*Table 14.* Temporal decomposition terms under strict pressure ($\rho$=100%, $T$=50 ms, $B$=1 s), aggregated across 15 corruptions.

| | Gradient-Free | | | Gradient-Based | | | | | | |
|---|---|---|---|---|---|---|---|---|---|---|
| Factor | Standard | NEO | LAME | AdaBN | Tent | ETA | SHOT-IM | DeYO | CMF | SAR |
| $\alpha$ (%) at $\rho$=100% | 100.0 | 100.0 | 98.8 | 97.2 | 41.2 | 41.0 | 33.2 | 31.8 | 25.1 | 20.6 |
| $\bar{\kappa}$ (%) at $T$=50 ms | 100.0 | 100.0 | 95.7 | 89.6 | 15.1 | 15.0 | 11.2 | 10.2 | 7.9 | 6.2 |
| $m$ at $B$=1 s | – | – | – | – | 18 | 18 | 13 | 53 | 14 | 7 |

## C.3. Per-Corruption Analysis

Table 36 reports the winning method for each corruption under 17 temporal scenarios, including offline. ETA wins in 103 of 255 cases (40.4%), but its dominance is confined to mildly relaxed settings; it mainly wins at $35\% \le \rho \le 70\%$, $T \ge 100$ ms, and $8 \le B \le 16$ s. AdaBN and NEO capture most high-pressure scenarios, with SHOT-IM winning under tight amortised budgets due to its frozen accuracy robustness. The phase transition from gradient-free to gradient-based winners is visible across all three protocols.

*Table 15.* Winning method per corruption across 17 scenarios; win counts in legend.

| Corruption | Offline | Discrete ($\rho$ %) | | | | | Continuous ($T$ ms) | | | | | Amortised ($B$ s) | | | | | |
|---|---|---|---|---|---|---|---|---|---|---|---|---|---|---|---|---|---|
| | | 100 | 70 | 50 | 35 | 25 | 50 | 100 | 200 | 400 | 1k | 1 | 2 | 4 | 8 | 16 | 32 |
| Gaussian noise | C | A | E | E | E | C | A | E | E | E | | S | S | D | E | C | C |
| Shot noise | C | A | E | E | E | C | A | E | E | E | E | S | S | D | E | E | C |
| Impulse noise | C | A | E | E | E | C | A | E | E | E | E | S | S | D | E | E | C |
| Defocus blur | C | N | N | E | E | C | N | N | E | E | | N | N | N | C | C | C |
| Glass blur | C | A | E | E | E | C | A | E | E | E | E | A | S | D | C | C | C |
| Motion blur | D | A | E | E | E | D | A | A | E | E | E | S | S | D | E | E | C |
| Zoom blur | C | A | A | E | E | C | A | A | E | E | E | A | S | S | E | E | C |
| Snow | C | A | A | E | E | D | A | A | E | E | E | S | S | D | E | E | C |
| Frost | C | A | A | E | E | D | A | A | E | E | E | S | S | D | C | E | C |
| Fog | D | A | A | E | E | D | A | A | A | E | E | A | S | S | E | E | C |
| Brightness | D | A | A | A | E | D | N | A | A | A | A | S | S | S | C | E | C |
| Contrast | C | E | E | E | E | C | A | E | E | E | E | A | A | D | E | E | C |
| Elastic transform | D | A | A | E | E | D | A | A | A | E | E | A | S | S | E | E | E |
| Pixelate | D | A | A | E | E | D | A | A | A | E | E | S | S | D | E | E | C |
| JPEG compression | C | A | A | E | E | D | N | A | E | E | E | S | S | D | E | C | C |

**E** TA (103)   **A** daBN (55)   **C** MF (39)   **S** HOT-IM (26)   **D** eYO (23)   **N** EO (9)

Tables 17–22 decompose utility under strict temporal pressure; relaxed scenarios move towards offline performance and are less informative. We only report accuracy and utility as the decomposition terms (Table 14) are nearly constant across corruptions. For gradient-free methods, the decomposition is near-trivial: minimal overhead yields $\bar{a} \approx U$. For gradient-based methods, overhead dominates: ETA achieves 36% accuracy but only 5.4% utility at $T{=}50$ ms. Under amortised evaluation, SHOT-IM maintains $\bar{a}_{\text{frozen}} \approx 18$–$66\%$. It does not discard source running statistics unlike other methods (§4.3), which only capture a premature distribution estimate from 7–18 batches of data. Table 16 shows that if the other methods preserved source statistics, contrary to their original design, their utility would improve significantly.

*Table 16.* Amortised utility $U_a$ (%) per corruption ($B = 1\,\text{s}$) for ResNet-50 on ImageNet-C, ablating the source of SHOT-IM's resilience. SHOT-IM$^*$ denotes a variant where the SHOT-IM loss restricted to normalisation layers only and the source running statistics are reset; conversely, ETA$^*$ denotes a variant of standard ETA but without resetting the source statistics. We show four representative corruptions for brevity, but the trend persists across all 15 corruptions.

| Corruption | SHOT-IM | SHOT-IM$^*$ | ETA | ETA$^*$ |
|---|---|---|---|---|
| Impulse Noise | 19.8 | 0.5 | 0.5 | **22.7** |
| Defocus Blur | 11.4 | 0.4 | 0.5 | **16.3** |
| Brightness | 65.7 | 1.6 | 1.6 | **66.5** |
| Contrast | 13.7 | 0.5 | 0.5 | **17.6** |

*Table 17.* Discrete utility (%) decomposition at $\rho{=}100\%$ (gradient-free).

| | Standard | | AdaBN | | LAME | | NEO | |
|---|---|---|---|---|---|---|---|---|
| **Corruption** | $\bar{a}_s$ | $U_d$ | $\bar{a}_s$ | $U_d$ | $\bar{a}_s$ | $U_d$ | $\bar{a}_s$ | $U_d$ |
| Gaussian noise | 3.0 | 3.0 | 16.1 | 15.7 | 2.6 | 2.5 | 5.2 | 5.2 |
| Shot noise | 3.7 | 3.7 | 16.7 | 16.3 | 3.2 | 3.1 | 6.1 | 6.1 |
| Impulse noise | 2.6 | 2.6 | 16.7 | 16.2 | 2.2 | 2.2 | 5.2 | 5.2 |
| Defocus blur | 17.9 | 17.9 | 15.1 | 14.7 | 17.6 | 17.6 | 21.1 | 21.1 |
| Glass blur | 9.7 | 9.7 | 15.4 | 15.0 | 9.0 | 8.9 | 12.6 | 12.6 |
| Motion blur | 14.7 | 14.7 | 26.2 | 25.5 | 13.9 | 13.5 | 17.8 | 17.8 |
| Zoom blur | 22.5 | 22.5 | 38.9 | 37.8 | 21.9 | 21.9 | 26.6 | 26.6 |
| Snow | 16.6 | 16.6 | 34.2 | 33.2 | 15.2 | 15.1 | 21.7 | 21.7 |
| Frost | 23.1 | 23.1 | 33.1 | 32.2 | 22.3 | 22.0 | 27.7 | 27.7 |
| Fog | 24.0 | 24.0 | 47.9 | 46.5 | 22.2 | 22.0 | 30.6 | 30.6 |
| Brightness | 59.1 | 59.1 | 65.3 | 63.5 | 58.7 | 58.7 | 60.3 | 60.3 |
| Contrast | 5.4 | 5.4 | 16.9 | 16.4 | 5.1 | 5.1 | 8.1 | 8.1 |
| Elastic transform | 16.5 | 16.5 | 44.1 | 42.9 | 14.6 | 14.5 | 24.9 | 24.9 |
| Pixelate | 20.9 | 20.9 | 49.1 | 47.7 | 20.3 | 20.1 | 26.0 | 26.0 |
| JPEG compression | 32.6 | 32.6 | 40.0 | 38.9 | 32.1 | 32.1 | 38.3 | 38.3 |

*Table 18.* Discrete utility (%) decomposition at $\rho{=}100\%$ (gradient-based).

| | Tent | | ETA | | SHOT-IM | | DeYO | | CMF | | SAR | |
|---|---|---|---|---|---|---|---|---|---|---|---|---|
| **Corruption** | $\bar{a}_s$ | $U_d$ | $\bar{a}_s$ | $U_d$ | $\bar{a}_s$ | $U_d$ | $\bar{a}_s$ | $U_d$ | $\bar{a}_s$ | $U_d$ | $\bar{a}_s$ | $U_d$ |
| Gaussian noise | 25.9 | 10.7 | 33.1 | 13.5 | 26.9 | 8.9 | 32.8 | 10.9 | 33.1 | 8.3 | 22.5 | 4.6 |
| Shot noise | 27.7 | 11.4 | 35.2 | 14.4 | 29.1 | 9.6 | 34.7 | 11.5 | 35.8 | 9.0 | 24.9 | 5.1 |
| Impulse noise | 27.0 | 11.1 | 33.6 | 13.8 | 27.6 | 9.2 | 34.5 | 11.4 | 35.0 | 8.8 | 23.1 | 4.8 |
| Defocus blur | 24.0 | 9.9 | 29.8 | 12.2 | 24.7 | 8.2 | 29.7 | 9.9 | 29.5 | 7.5 | 21.5 | 4.4 |
| Glass blur | 23.3 | 9.6 | 29.4 | 12.1 | 23.8 | 7.9 | 29.7 | 9.9 | 30.0 | 7.6 | 21.1 | 4.3 |
| Motion blur | 37.0 | 15.3 | 43.9 | 18.0 | 39.2 | 13.0 | 42.3 | 13.6 | 44.6 | 11.2 | 32.7 | 6.7 |
| Zoom blur | 47.4 | 19.6 | 50.8 | 20.8 | 48.9 | 16.2 | 49.6 | 15.4 | 50.7 | 12.7 | 44.4 | 9.1 |
| Snow | 44.0 | 18.1 | 48.6 | 19.9 | 46.7 | 15.5 | 49.4 | 15.5 | 49.3 | 12.3 | 40.2 | 8.3 |
| Frost | 39.4 | 16.3 | 44.2 | 18.1 | 40.1 | 13.3 | 44.0 | 14.1 | 44.0 | 10.9 | 36.8 | 7.6 |
| Fog | 55.7 | 23.0 | 58.5 | 24.0 | 56.8 | 18.9 | 58.4 | 17.9 | 58.8 | 14.5 | 52.6 | 10.8 |
| Brightness | 67.5 | 27.8 | 67.7 | 27.8 | 67.4 | 22.3 | 68.4 | 20.2 | 67.4 | 17.1 | 66.8 | 13.8 |
| Contrast | 26.5 | 10.9 | 41.4 | 17.0 | 16.2 | 5.4 | 40.3 | 12.8 | 42.4 | 10.7 | 27.8 | 5.7 |
| Elastic transform | 52.6 | 21.7 | 55.8 | 22.9 | 53.4 | 17.7 | 54.8 | 16.8 | 56.6 | 14.2 | 48.3 | 10.0 |
| Pixelate | 57.1 | 23.6 | 59.4 | 24.3 | 57.0 | 18.9 | 59.2 | 17.9 | 59.4 | 14.9 | 54.3 | 11.2 |
| JPEG compression | 50.2 | 20.7 | 53.2 | 21.8 | 51.3 | 17.0 | 53.3 | 16.5 | 53.5 | 13.3 | 47.1 | 9.7 |

*Table 19.* Continuous utility (%) decomposition at $T=50$ ms (gradient-free).

| Corruption | Standard $\bar{a}$ | Standard $U_c$ | AdaBN $\bar{a}$ | AdaBN $U_c$ | LAME $\bar{a}$ | LAME $U_c$ | NEO $\bar{a}$ | NEO $U_c$ |
|---|---|---|---|---|---|---|---|---|
| Gaussian noise | 3.0 | 3.0 | 16.2 | 14.5 | 2.6 | 2.3 | 5.2 | 5.2 |
| Shot noise | 3.7 | 3.7 | 16.8 | 15.0 | 3.2 | 2.9 | 6.1 | 6.1 |
| Impulse noise | 2.6 | 2.6 | 16.7 | 15.0 | 2.2 | 2.0 | 5.2 | 5.2 |
| Defocus blur | 17.9 | 17.9 | 15.1 | 13.5 | 17.6 | 17.6 | 21.1 | 21.1 |
| Glass blur | 9.7 | 9.7 | 15.4 | 13.8 | 8.9 | 8.6 | 12.6 | 12.6 |
| Motion blur | 14.7 | 14.7 | 26.2 | 23.5 | 13.9 | 12.7 | 17.8 | 17.8 |
| Zoom blur | 22.5 | 22.5 | 38.9 | 34.8 | 21.9 | 21.7 | 26.6 | 26.6 |
| Snow | 16.6 | 16.6 | 34.2 | 30.6 | 15.2 | 14.7 | 21.7 | 21.7 |
| Frost | 23.1 | 23.1 | 33.1 | 29.7 | 22.3 | 21.2 | 27.7 | 27.7 |
| Fog | 24.0 | 24.0 | 47.8 | 42.9 | 22.2 | 21.4 | 30.6 | 30.6 |
| Brightness | 59.1 | 59.1 | 65.3 | 58.5 | 58.7 | 58.6 | 60.3 | 60.3 |
| Contrast | 5.4 | 5.4 | 16.9 | 15.1 | 5.1 | 5.1 | 8.1 | 8.1 |
| Elastic transform | 16.5 | 16.5 | 44.2 | 39.5 | 14.6 | 14.2 | 24.9 | 24.9 |
| Pixelate | 20.9 | 20.9 | 49.1 | 44.0 | 20.3 | 19.5 | 26.0 | 26.0 |
| JPEG compression | 32.6 | 32.6 | 40.0 | 35.8 | 32.1 | 32.0 | 38.3 | 38.3 |

*Table 20.* Continuous utility (%) decomposition at $T=50$ ms (gradient-based).

| Corruption | Tent $\bar{a}$ | Tent $U_c$ | ETA $\bar{a}$ | ETA $U_c$ | SHOT-IM $\bar{a}$ | SHOT-IM $U_c$ | DeYO $\bar{a}$ | DeYO $U_c$ | CMF $\bar{a}$ | CMF $U_c$ | SAR $\bar{a}$ | SAR $U_c$ |
|---|---|---|---|---|---|---|---|---|---|---|---|---|
| Gaussian noise | 30.0 | 4.5 | 36.0 | 5.4 | 29.4 | 3.3 | 36.7 | 3.9 | 37.6 | 2.9 | 31.5 | 1.9 |
| Shot noise | 31.7 | 4.8 | 38.7 | 5.8 | 32.0 | 3.6 | 38.7 | 4.1 | 39.8 | 3.1 | 31.4 | 1.9 |
| Impulse noise | 31.3 | 4.7 | 38.2 | 5.7 | 30.8 | 3.4 | 38.4 | 4.1 | 39.3 | 3.1 | 32.6 | 2.0 |
| Defocus blur | 27.7 | 4.2 | 33.2 | 5.0 | 27.3 | 3.0 | 33.8 | 3.6 | 34.9 | 2.8 | 29.1 | 1.8 |
| Glass blur | 26.9 | 4.0 | 33.2 | 4.9 | 26.8 | 3.0 | 33.5 | 3.5 | 34.8 | 2.7 | 28.2 | 1.7 |
| Motion blur | 41.1 | 6.2 | 47.8 | 7.1 | 43.2 | 4.8 | 48.3 | 4.9 | 47.9 | 3.7 | 41.7 | 2.6 |
| Zoom blur | 49.3 | 7.4 | 52.7 | 7.9 | 50.4 | 5.6 | 52.8 | 5.3 | 53.1 | 4.2 | 49.2 | 3.1 |
| Snow | 47.2 | 7.1 | 52.1 | 7.8 | 49.1 | 5.5 | 52.5 | 5.3 | 52.6 | 4.1 | 47.2 | 2.9 |
| Frost | 41.2 | 6.2 | 46.0 | 6.9 | 41.5 | 4.6 | 46.2 | 4.8 | 46.8 | 3.6 | 42.5 | 2.6 |
| Fog | 57.6 | 8.7 | 60.0 | 9.0 | 57.8 | 6.5 | 60.5 | 5.9 | 60.3 | 4.7 | 57.6 | 3.6 |
| Brightness | 67.5 | 10.2 | 67.8 | 10.2 | 67.6 | 7.6 | 68.0 | 6.5 | 68.0 | 5.4 | 67.4 | 4.2 |
| Contrast | 26.3 | 4.0 | 45.6 | 6.8 | 13.2 | 1.5 | 46.2 | 4.7 | 47.2 | 3.7 | 38.3 | 2.4 |
| Elastic transform | 54.6 | 8.2 | 57.7 | 8.6 | 55.4 | 6.2 | 58.4 | 5.8 | 57.9 | 4.5 | 54.6 | 3.4 |
| Pixelate | 58.5 | 8.8 | 60.9 | 9.1 | 59.1 | 6.6 | 61.5 | 6.0 | 60.9 | 4.8 | 58.4 | 3.6 |
| JPEG compression | 52.5 | 7.9 | 55.2 | 8.3 | 52.9 | 5.9 | 55.8 | 5.6 | 56.0 | 4.4 | 52.5 | 3.3 |

*Table 21.* Amortised utility (%) decomposition at $B=1$ s (gradient-free).

| Corruption | Standard $\bar{a}_a$ | Standard $\bar{a}_f$ | Standard $U_a$ | AdaBN $\bar{a}_a$ | AdaBN $\bar{a}_f$ | AdaBN $U_a$ | LAME $\bar{a}_a$ | LAME $\bar{a}_f$ | LAME $U_a$ | NEO $\bar{a}_a$ | NEO $\bar{a}_f$ | NEO $U_a$ |
|---|---|---|---|---|---|---|---|---|---|---|---|---|
| Gaussian noise | 3.0 | – | 3.0 | 16.2 | – | 16.2 | 2.6 | – | 2.6 | 5.2 | – | 5.2 |
| Shot noise | 3.7 | – | 3.7 | 16.8 | – | 16.8 | 3.2 | – | 3.2 | 6.1 | – | 6.1 |
| Impulse noise | 2.6 | – | 2.6 | 16.7 | – | 16.7 | 2.2 | – | 2.2 | 5.2 | – | 5.2 |
| Defocus blur | 17.9 | – | 17.9 | 15.1 | – | 15.1 | 17.6 | – | 17.6 | 21.1 | – | 21.1 |
| Glass blur | 9.7 | – | 9.7 | 15.4 | – | 15.4 | 8.9 | – | 8.9 | 12.6 | – | 12.6 |
| Motion blur | 14.7 | – | 14.7 | 26.2 | – | 26.2 | 13.9 | – | 13.9 | 17.8 | – | 17.8 |
| Zoom blur | 22.5 | – | 22.5 | 38.9 | – | 38.9 | 21.9 | – | 21.9 | 26.6 | – | 26.6 |
| Snow | 16.6 | – | 16.6 | 34.2 | – | 34.2 | 15.2 | – | 15.2 | 21.7 | – | 21.7 |
| Frost | 23.1 | – | 23.1 | 33.1 | – | 33.1 | 22.3 | – | 22.3 | 27.7 | – | 27.7 |
| Fog | 24.0 | – | 24.0 | 47.8 | – | 47.8 | 22.2 | – | 22.2 | 30.6 | – | 30.6 |
| Brightness | 59.1 | – | 59.1 | 65.3 | – | 65.3 | 58.7 | – | 58.7 | 60.3 | – | 60.3 |
| Contrast | 5.4 | – | 5.4 | 16.9 | – | 16.9 | 5.1 | – | 5.1 | 8.1 | – | 8.1 |
| Elastic transform | 16.5 | – | 16.5 | 44.2 | – | 44.2 | 14.6 | – | 14.6 | 24.9 | – | 24.9 |
| Pixelate | 20.9 | – | 20.9 | 49.1 | – | 49.1 | 20.3 | – | 20.3 | 26.0 | – | 26.0 |
| JPEG compression | 32.6 | – | 32.6 | 40.0 | – | 40.0 | 32.1 | – | 32.1 | 38.3 | – | 38.3 |

Gradient-free methods never exhaust the budget; $\bar{a}_f$ is undefined and utility follows offline rankings.

*Table 22.* Amortised utility (%) decomposition at $B=1$ s (gradient-based).

| Corruption | Tent | | ETA | | SHOT-IM | | | DeYO | | CMF | | SAR | |
|---|---|---|---|---|---|---|---|---|---|---|---|---|---|
| | $\bar{a}_a$ | $U_a$ | $\bar{a}_a$ | $U_a$ | $\bar{a}_a$ | $\bar{a}_f$ | $U_a$ | $\bar{a}_a$ | $U_a$ | $\bar{a}_a$ | $U_a$ | $\bar{a}_a$ | $U_a$ |
| Gaussian noise | 18.6 | 0.5 | 19.7 | 0.6 | 19.4 | 18.3 | 18.3 | 20.0 | 0.5 | 19.8 | 0.3 | 18.3 | 0.3 |
| Shot noise | 18.3 | 0.5 | 20.3 | 0.6 | 19.0 | 18.6 | 18.6 | 20.6 | 0.5 | 20.3 | 0.3 | 18.3 | 0.3 |
| Impulse noise | 17.0 | 0.5 | 18.9 | 0.5 | 16.7 | 19.9 | 19.8 | 19.5 | 0.5 | 8.2 | 0.3 | 18.1 | 0.3 |
| Defocus blur | 15.5 | 0.5 | 16.8 | 0.5 | 13.1 | 11.4 | 11.4 | 18.3 | 0.5 | 7.7 | 0.3 | 16.1 | 0.2 |
| Glass blur | 15.4 | 0.5 | 16.5 | 0.5 | 16.1 | 15.1 | 15.2 | 16.9 | 0.4 | 15.3 | 0.3 | 16.5 | 0.2 |
| Motion blur | 26.6 | 0.7 | 27.3 | 0.7 | 26.3 | 27.9 | 27.9 | 1.3 | 0.6 | 27.6 | 0.4 | 27.7 | 0.3 |
| Zoom blur | 38.3 | 1.0 | 39.8 | 1.0 | 39.4 | 38.0 | 38.0 | 2.4 | 0.7 | 41.7 | 0.6 | 40.4 | 0.5 |
| Snow | 34.9 | 0.9 | 36.5 | 0.9 | 36.8 | 34.9 | 34.9 | 37.2 | 0.8 | 39.4 | 0.6 | 39.5 | 0.5 |
| Frost | 34.6 | 0.9 | 35.7 | 0.9 | 34.7 | 35.0 | 35.0 | 35.8 | 0.7 | 36.5 | 0.5 | 34.6 | 0.4 |
| Fog | 46.9 | 1.2 | 48.3 | 1.2 | 46.6 | 44.9 | 44.9 | 47.2 | 0.9 | 49.8 | 0.6 | 46.7 | 0.5 |
| Brightness | 64.9 | 1.6 | 64.8 | 1.6 | 64.5 | 65.7 | 65.7 | 65.3 | 1.0 | 9.2 | 0.8 | 67.4 | 0.7 |
| Contrast | 16.8 | 0.5 | 18.8 | 0.5 | 17.4 | 13.6 | 13.7 | 19.9 | 0.5 | 18.9 | 0.3 | 18.3 | 0.3 |
| Elastic transform | 41.9 | 1.1 | 43.3 | 1.1 | 42.4 | 43.4 | 43.4 | 43.1 | 0.8 | 44.4 | 0.6 | 44.0 | 0.5 |
| Pixelate | 49.4 | 1.2 | 50.0 | 1.2 | 49.5 | 50.7 | 50.7 | 49.7 | 0.9 | 52.1 | 0.7 | 53.6 | 0.6 |
| JPEG compression | 42.5 | 1.1 | 43.8 | 1.1 | 41.3 | 45.8 | 45.7 | 8.3 | 0.8 | 43.9 | 0.6 | 43.8 | 0.5 |

For all methods except SHOT-IM, the frozen accuracy ($\bar{a}_f$) is uniformly 0.1% across all corruptions. Gradient-based methods exhaust their budget within 7–18 batches of 781; >97% of the stream runs on frozen inference.

Table 23 aggregates losses and the average utility deficit relative to the cell winner across all 240 evaluation cells, revealing a sharp trade-off between win frequency and worst-case risk. This profile is contextualised by Table 24, which tracks the frequency and scenario distribution of absolute utility drops below standard inference. Together, these failure dynamics explain the shifting rank correlations in Table 25, which measures the Spearman correlation (r) between offline and temporal rankings; for each cell, all 10 methods are ranked by their offline accuracy and by their utility metric, then $r$ is computed between the two rank vectors. Under severe temporal pressure, the mean correlation is consistently negative, indicating that offline-optimal methods perform poorly; this correlation moves toward near-perfect alignment as pressure relaxes.

*Table 23.* Performance deficit profile across 240 evaluation cells for ImageNet-C with ResNet-50. We report the number of such cells where a method fails to rank first (losses) and its average utility deficit to the cell winner (Mean $\Delta$, pp). Lower is better for both.

| Method | Overall | | Discrete | | Continuous | | Amortised | |
|---|---|---|---|---|---|---|---|---|
| | Losses | Mean $\Delta$ | Losses | Mean $\Delta$ | Losses | Mean $\Delta$ | Losses | Mean $\Delta$ |
| Standard | 240 | 22.06 | 75 | 22.30 | 75 | 19.21 | 90 | 24.24 |
| NEO | 231 | 18.78 | 73 | 18.82 | 71 | 16.08 | 87 | 20.96 |
| LAME | 240 | 22.86 | 75 | 23.08 | 75 | 20.08 | 90 | 25.00 |
| AdaBN | 185 | 11.44 | 53 | 12.62 | 48 | 10.15 | 84 | 11.44 |
| Tent | 240 | 12.00 | 75 | 8.20 | 75 | 9.86 | 90 | 16.95 |
| ETA | 137 | 14.40 | 39 | 7.87 | 31 | 15.54 | 67 | 17.67 |
| SHOT-IM | 214 | 10.24 | 75 | 11.95 | 75 | 12.41 | 64 | 5.71 |
| DeYO | 222 | 11.73 | 67 | 10.60 | 75 | 9.56 | 80 | 14.71 |
| CMF | 211 | 15.00 | 68 | 14.77 | 75 | 11.60 | 68 | 18.96 |
| SAR | 240 | 20.07 | 75 | 21.38 | 75 | 16.09 | 90 | 22.28 |

*Table 24.* Number of temporal evaluations in which each method's utility falls below standard inference, and the share of those misses attributable to each evaluation category. ImageNet-C with ResNet-50.

| Method | Total (240) | Discrete (75) | Continuous (75) | Amortised (90) |
|---|---|---|---|---|
| NEO | 0 | – | – | – |
| LAME | 240 | 31% | 31% | 38% |
| AdaBN | 17 | 29% | 35% | 35% |
| Tent | 63 | 17% | 29% | 54% |
| ETA | 57 | 16% | 28% | 56% |
| SHOT-IM | 40 | 42% | 50% | 8% |
| DeYO | 65 | 23% | 31% | 46% |
| CMF | 79 | 28% | 30% | 42% |
| SAR | 109 | 29% | 29% | 41% |

*Table 25.* Spearman rank correlation ($r$) between offline accuracy and temporal utility rankings for ResNet-50 across 15 ImageNet-C corruptions. Strong negative correlations under tight budgets ($\rho = 100\%$, $T = 50$ ms, $B \leq 2$ s) show that offline-optimal methods perform worst under real-time pressure, only aligning ($r \geq 0.89$) as constraints relax.

| Corruption | Discrete ($\rho$, %) | | | | | Continuous ($T$, ms) | | | | | Amortised ($B$, s) | | | | | |
|---|---|---|---|---|---|---|---|---|---|---|---|---|---|---|---|---|
| | 100 | 70 | 50 | 35 | 25 | 50 | 100 | 200 | 400 | 1000 | 1 | 2 | 4 | 8 | 16 | 32 |
| *Noise* | | | | | | | | | | | | | | | | |
| Gaussian Noise | 0.50 | 0.60 | 0.73 | 0.81 | 0.96 | 0.02 | 0.56 | 0.83 | 0.92 | 0.92 | -0.66 | -0.61 | 0.50 | 0.92 | 0.94 | 0.95 |
| Shot Noise | 0.58 | 0.73 | 0.81 | 0.84 | 1.00 | -0.01 | 0.61 | 0.88 | 0.94 | 0.94 | -0.41 | -0.39 | 0.77 | 0.96 | 0.96 | 0.99 |
| Impulse Noise | 0.50 | 0.59 | 0.73 | 0.81 | 0.96 | 0.12 | 0.56 | 0.83 | 0.92 | 0.92 | -0.62 | -0.61 | 0.50 | 0.92 | 0.92 | 0.95 |
| *Blur* | | | | | | | | | | | | | | | | |
| Defocus Blur | -0.68 | -0.47 | 0.16 | 0.59 | 0.96 | -0.68 | -0.54 | 0.54 | 0.88 | 0.93 | -0.75 | -0.73 | 0.03 | 0.92 | 0.95 | 0.98 |
| Glass Blur | -0.25 | 0.27 | 0.68 | 0.77 | 0.96 | -0.68 | 0.37 | 0.88 | 0.92 | 0.93 | -0.67 | -0.61 | 0.62 | 0.95 | 0.94 | 0.94 |
| Motion Blur | -0.25 | 0.37 | 0.67 | 0.83 | 0.98 | -0.66 | 0.32 | 0.83 | 0.93 | 0.95 | -0.47 | -0.38 | 0.65 | 0.94 | 0.90 | 0.95 |
| Zoom Blur | -0.66 | 0.02 | 0.42 | 0.71 | 1.00 | -0.66 | -0.33 | 0.59 | 0.81 | 0.95 | -0.48 | -0.44 | 0.58 | 0.90 | 0.90 | 0.93 |
| *Weather* | | | | | | | | | | | | | | | | |
| Snow | -0.27 | 0.28 | 0.54 | 0.72 | 0.95 | -0.71 | 0.16 | 0.58 | 0.79 | 0.94 | -0.48 | -0.48 | 0.45 | 0.95 | 0.95 | 0.98 |
| Frost | -0.71 | -0.47 | 0.15 | 0.59 | 0.95 | -0.71 | -0.55 | 0.24 | 0.81 | 0.90 | -0.59 | -0.59 | 0.03 | 0.95 | 0.93 | 0.95 |
| Fog | -0.56 | -0.03 | 0.39 | 0.67 | 0.95 | -0.70 | -0.15 | 0.47 | 0.67 | 0.88 | -0.50 | -0.47 | 0.49 | 0.84 | 0.84 | 0.85 |
| Brightness | -0.67 | -0.65 | -0.67 | -0.03 | 0.88 | -0.75 | -0.65 | -0.65 | -0.65 | 0.44 | -0.45 | -0.45 | -0.07 | 0.82 | 0.87 | 0.88 |
| *Digital* | | | | | | | | | | | | | | | | |
| Contrast | 0.66 | 0.83 | 0.92 | 0.94 | 1.00 | -0.30 | 0.83 | 0.95 | 0.95 | 0.95 | -0.66 | -0.61 | 0.70 | 0.95 | 0.96 | 0.99 |
| Elastic transform | -0.12 | 0.20 | 0.53 | 0.67 | 0.96 | -0.70 | 0.21 | 0.54 | 0.71 | 0.88 | -0.50 | -0.47 | 0.49 | 0.84 | 0.84 | 0.90 |
| Pixelate | -0.31 | 0.35 | 0.64 | 0.79 | 0.99 | -0.60 | 0.22 | 0.60 | 0.72 | 0.94 | -0.39 | -0.39 | 0.68 | 0.93 | 0.93 | 0.95 |
| JPEG compression | -0.66 | -0.66 | 0.09 | 0.60 | 0.99 | -0.67 | -0.66 | 0.09 | 0.78 | 0.94 | -0.44 | -0.44 | 0.10 | 0.96 | 0.99 | 0.99 |
| **Mean** | -0.19 | 0.13 | 0.45 | 0.69 | 0.97 | -0.51 | 0.07 | 0.55 | 0.74 | 0.89 | -0.54 | -0.51 | 0.44 | 0.92 | 0.92 | 0.95 |
| **Std** | 0.51 | 0.49 | 0.40 | 0.22 | 0.03 | 0.31 | 0.51 | 0.41 | 0.39 | 0.13 | 0.11 | 0.11 | 0.27 | 0.05 | 0.04 | 0.04 |

## C.4. Vision Transformer (ViT-B/16) Main Results

This section provides the baseline performance, computational overhead, and utility decompositions for ViT-B/16 (Dosovitskiy et al., 2021) under temporal constraints on ImageNet-C. Table 26 establishes the offline baseline across four corruptions, and Table 28 expands on this to cover all 15 corruptions. Table 27 details the empirical latency decomposition ($\bar{\delta}, \bar{e}, \bar{\ell}$) and hardware slowdown factors on an Nvidia RTX 4080 GPU. Because ViT uses layer normalisation instead of batch normalisation, AdaBN is excluded; we instead evaluate SPA (Niu et al., 2025) and ZeroSIAM (Chen et al., 2026) (detailed in §C.1). Mirroring the ResNet-50 analysis, Tables 29, 30, and 31 dissect operational utility into its core mechanical drivers: batch availability ($\alpha$) for discrete, responsiveness ($\bar{\kappa}$) for continuous, and boundary states ($\beta$) for amortised scenarios.

*Table 26.* Offline accuracy (%) with ranks for the four corruptions in Figure 5. SPA ranks first across 13 of 15 and CMF across the remaining two; we show four for brevity.

| | Corruptions | | | |
|---|---|---|---|---|
| **Method** | Impulse | Brightness | Defocus | Contrast |
| Standard | 57.50 (9) | 77.67 (9) | 46.89 (9) | 32.64 (9) |
| LAME | 57.22 (10) | 77.36 (10) | 46.33 (10) | 25.69 (10) |
| NEO | 58.30 (8) | 78.14 (8) | 49.74 (8) | 58.57 (8) |
| Tent | 61.79 (5) | 79.16 (5) | 59.25 (5) | 67.26 (2) |
| ETA | 64.20 (3) | 80.27 (3) | 61.29 (2) | 66.41 (3) |
| SHOT-IM | 52.41 (11) | 77.21 (11) | 30.78 (11) | 1.59 (11) |
| SAR | 60.53 (7) | 78.64 (7) | 57.60 (7) | 64.27 (7) |
| SPA | **65.40** (1) | **80.86** (1) | 61.15 (3) | 65.27 (5) |
| DeYO | 61.15 (6) | 78.99 (6) | 57.83 (6) | 64.36 (6) |
| CMF | 64.75 (2) | 80.35 (2) | **62.18** (1) | **69.29** (1) |
| ZeroSIAM | 63.97 (4) | 79.99 (4) | 60.10 (4) | 66.31 (4) |

*Table 27.* Per-batch overhead decomposition ($\lambda = 105.3$ ms; notation per §3), averaged across 11,715 batches. All methods satisfy $\bar{e} < \lambda$, so the $\bar{e} - \lambda$ column is omitted.

| | Latency Breakdown (ms) | | | Slowdown |
|---|---|---|---|---|
| **Method** | $\bar{\delta}$ | $\bar{e}$ | $\bar{\ell}$ | $\bar{\delta}/\lambda$ |
| Standard | 98.5 | 98.5 | 0.0 | 0.94× |
| LAME | 99.7 | 99.7 | 0.1 | 0.95× |
| NEO | 98.8 | 98.8 | 0.1 | 0.94× |
| Tent | 230.9 | 98.5 | 132.4 | 2.19× |
| ETA | 230.4 | 98.5 | 131.9 | 2.19× |
| SHOT-IM | 255.7 | 98.5 | 157.2 | 2.43× |
| SAR | 463.3 | 98.8 | 364.5 | 4.40× |
| SPA | 588.2 | 98.3 | 489.9 | 5.59× |
| DeYO | 316.9 | 98.3 | 218.6 | 3.01× |
| CMF | 367.9 | 98.3 | 269.6 | 3.49× |
| ZeroSIAM | 331.6 | 98.6 | 232.9 | 3.15× |

*Table 28.* Offline accuracy (%) across all 15 ImageNet-C corruptions (ViT-B/16).

| Corruption | Gradient-Free | | | Gradient-Based | | | | | | | |
|---|---|---|---|---|---|---|---|---|---|---|---|
| | Standard | LAME | NEO | Tent | ETA | SHOT-IM | SAR | SPA | DeYO | CMF | ZeroSIAM |
| Gaussian noise | 56.76 | 56.42 | 57.51 | 60.19 | 62.85 | 49.53 | 59.17 | **63.79** | 59.08 | 63.50 | 62.39 |
| Shot noise | 56.79 | 56.48 | 57.59 | 61.56 | 64.19 | 51.84 | 60.38 | **65.34** | 61.13 | 64.89 | 63.80 |
| Impulse noise | 57.50 | 57.22 | 58.30 | 61.79 | 64.20 | 52.41 | 60.53 | **65.40** | 61.15 | 64.75 | 63.97 |
| Defocus blur | 46.89 | 46.33 | 49.74 | 59.25 | 61.29 | 30.78 | 57.60 | 61.15 | 57.83 | **62.18** | 60.10 |
| Glass blur | 35.58 | 34.78 | 38.20 | 56.69 | 61.86 | 46.01 | 56.01 | **63.78** | 59.08 | 61.97 | 60.70 |
| Motion blur | 53.13 | 52.74 | 54.84 | 63.46 | 66.77 | 49.20 | 61.79 | **69.21** | 64.52 | 66.84 | 66.11 |
| Zoom blur | 44.81 | 44.10 | 47.57 | 59.30 | 64.96 | 53.67 | 58.01 | **67.97** | 62.06 | 64.79 | 63.37 |
| Snow | 62.24 | 57.02 | 64.67 | 59.92 | 71.37 | 61.82 | 65.94 | **74.16** | 69.00 | 71.60 | 70.45 |
| Frost | 62.57 | 61.61 | 65.04 | 64.14 | 69.45 | 58.70 | 63.75 | **72.03** | 66.56 | 70.36 | 68.56 |
| Fog | 65.72 | 63.77 | 71.26 | 2.36 | 72.26 | 53.81 | 67.24 | **75.61** | 1.79 | 74.77 | 72.83 |
| Brightness | 77.67 | 77.36 | 78.14 | 79.16 | 80.27 | 77.21 | 78.64 | **80.86** | 78.99 | 80.35 | 79.99 |
| Contrast | 32.64 | 25.69 | 58.57 | 67.26 | 66.41 | 1.59 | 64.27 | 65.27 | 64.36 | **69.29** | 66.31 |
| Elastic transform | 46.04 | 44.35 | 49.99 | 61.27 | 70.93 | 66.18 | 61.30 | **74.15** | 68.50 | 70.54 | 69.62 |
| Pixelate | 66.97 | 66.58 | 67.69 | 72.70 | 75.80 | 70.51 | 72.20 | **77.67** | 74.24 | 75.76 | 75.46 |
| JPEG compression | 67.60 | 67.28 | 68.84 | 70.71 | 73.68 | 69.07 | 70.12 | **75.00** | 72.20 | 73.69 | 72.66 |
| **Mean** | 55.53 | 54.12 | 59.20 | 59.98 | 68.42 | 52.82 | 63.80 | **70.09** | 61.37 | 69.02 | 67.75 |

*Table 29.* Discrete utility decomposition at $\rho = 100\%$, aggregated across 15 corruptions (11,715 batches). Availability $\alpha$, the fraction of batches served, determines ranking.

| Method | $\alpha$ (%) | $|\mathcal{Q}|$ | $\bar{a}_{\text{served}}$ (%) | $U_{\text{discrete}}$ (%) |
|---|---|---|---|---|
| Standard | 100.0 | 11,715 | 55.5 | 55.5 |
| LAME | 100.0 | 11,715 | 54.1 | 54.1 |
| NEO | 100.0 | 11,715 | 59.2 | **59.2** |
| Tent | 46.0 | 5,385 | 58.7 | 27.0 |
| ETA | 49.7 | 5,827 | 57.9 | 28.8 |
| SHOT-IM | 41.5 | 4,867 | 56.5 | 23.5 |
| SAR | 23.1 | 2,701 | 57.7 | 13.3 |
| SPA | 18.2 | 2,129 | **67.1** | 12.2 |
| DeYO | 34.3 | 4,021 | 57.7 | 19.8 |
| CMF | 28.6 | 3,348 | 65.8 | 18.8 |
| ZeroSIAM | 32.0 | 3,753 | 65.3 | 20.9 |

*Table 30.* Continuous utility decomposition at $T = 200$ ms. Responsiveness $\bar{\kappa}$ determines ranking; alignment near unity indicates accuracy and responsiveness are approximately independent.

| Method | $\bar{a}$ (%) | $\bar{\kappa}$ (%) | $\text{Cov}(a, \kappa)$ | $U_{\text{continuous}}$ (%) |
|---|---|---|---|---|
| Standard | 55.53 | 100.0 | 0.0000 | 55.53 |
| LAME | 54.12 | 100.0 | 0.0000 | 54.12 |
| NEO | 59.20 | 100.0 | 0.0000 | **59.20** |
| Tent | 59.98 | 43.1 | −0.0001 | 25.83 |
| ETA | 68.42 | 43.3 | −0.0004 | 29.59 |
| SHOT-IM | 52.82 | 38.7 | −0.0000 | 20.44 |
| SAR | 63.80 | 21.0 | −0.0001 | 13.39 |
| SPA | **70.09** | 16.5 | −0.0002 | 11.55 |
| DeYO | 61.37 | 31.6 | 0.0003 | 18.57 |
| CMF | 69.02 | 26.6 | −0.0001 | 18.36 |
| ZeroSIAM | 67.75 | 29.6 | −0.0002 | 20.04 |

*Table 31.* Amortised utility decomposition at $B = 2.5$ s. For gradient-based methods, frozen accuracy determines rankings. For gradient-free methods, adapt accuracy determines ranking.

| Method | $\beta$ (%) | $\bar{a}_{\text{adapt}}$ (%) | $\bar{a}_{\text{frozen}}$ (%) | $U_{\text{amortised}}$ (%) |
|---|---|---|---|---|
| Standard | 100.0 | 55.53 | - | 55.53 |
| LAME | 100.0 | 54.12 | - | 54.12 |
| NEO | 100.0 | 59.20 | - | **59.20** |
| Tent | 2.4 | 55.36 | 56.18 | 56.16 |
| ETA | 2.9 | 56.40 | 57.94 | 57.89 |
| SHOT-IM | 2.0 | 58.83 | **58.73** | 58.74 |
| SAR | 0.9 | 55.65 | 56.02 | 56.02 |
| SPA | 0.8 | 57.29 | 58.19 | 58.19 |
| DeYO | 1.9 | 51.91 | 54.43 | 54.39 |
| CMF | 1.2 | 56.74 | 57.47 | 57.46 |
| ZeroSIAM | 1.4 | 54.80 | 53.34 | 53.36 |

While SPA is the majority offline winner (Table 28), it collapses under real-time temporal pressure. This failure is driven by its processing time under adaptation ($\bar{\ell} = 489.9$ ms), which leads to a $5.6\times$ slowdown relative to the non-adaptive baseline (Table 27). As a consequence, under discrete evaluation ($\rho = 100\%$), it achieves the highest served accuracy but the lowest

utility across all methods; its availability collapses to just 18.2%. Similarly, under continuous evaluation ($T = 200$ ms), the processing latency delays predictions so heavily that its responsiveness drops to 16.5%, the lowest among all methods. Lastly, under amortised evaluation, it only adapts on six batches of data ($< 0.8\%$ of the stream) and is outperformed by NEO, a gradient-based method that adapts on the entire stream. Across all three scenarios, rank instability persists.

## C.5. Corruption-Aggregated Parameter Sweeps (ViT-B/16)

Tables 32–34 report utility across all parameter values for each protocol, aggregated across 15 corruptions for the ViT-B/16 architecture. We omit standard deviation because it reflects corruption difficulty rather than method consistency; the former varies widely, and corruption-level breakdowns appear in the following section. Across 11,715 batches, latency variance is minimal, *i.e.,* $< 1$ ms for gradient-free and $< 3$ ms for gradient-based methods; this is negligible relative to the 130–490,ms mean differences between method families. The only exceptions are ETA ($\sigma = 8.9$ ms), CMF ($\sigma = 10.3$ ms), and DeYO ($\sigma = 32.5$ ms); this is due to their filtering mechanics, which skip an update if there are no valid samples.

*Table 32.* Discrete utility (%) across utilisation levels ($\rho$), aggregated across 15 corruptions (ViT-B/16).

| Method | $\rho = 100\%$ | $\rho = 70\%$ | $\rho = 50\%$ | $\rho = 35\%$ | $\rho = 25\%$ |
|---|---|---|---|---|---|
| Standard | 55.53 | 55.53 | 55.53 | 55.53 | 55.53 |
| NEO | **59.20** | **59.20** | 59.20 | 59.20 | 59.20 |
| LAME | 54.12 | 54.12 | 54.12 | 54.12 | 54.12 |
| Tent | 27.00 | 38.47 | 54.69 | 59.98 | 59.98 |
| ETA | 28.81 | 43.84 | **62.27** | **68.42** | 68.42 |
| SHOT-IM | 23.47 | 32.46 | 44.74 | 52.82 | 52.82 |
| DeYO | 19.81 | 28.06 | 39.66 | 56.22 | 61.37 |
| ZeroSIAM | 20.91 | 29.97 | 41.40 | 56.44 | 67.75 |
| CMF | 18.80 | 27.05 | 38.87 | 55.63 | **69.02** |
| SAR | 13.31 | 18.47 | 26.64 | 38.14 | 57.74 |
| SPA | 12.19 | 17.32 | 24.77 | 35.22 | 50.03 |

*Table 33.* Continuous utility (%) across HCI thresholds ($T$), aggregated across 15 corruptions (ViT-B/16).

| Method | $T = 200$ms | $T = 400$ms | $T = 1000$ms | $T = 2000$ms |
|---|---|---|---|---|
| Standard | 55.53 | 55.53 | 55.53 | 55.53 |
| NEO | **59.20** | **59.20** | 59.20 | 59.20 |
| LAME | 54.12 | 54.12 | 54.12 | 54.12 |
| Tent | 25.83 | 42.08 | 52.61 | 56.26 |
| ETA | 29.59 | 48.06 | **60.03** | **64.18** |
| SHOT-IM | 20.44 | 34.99 | 45.23 | 48.94 |
| DeYO | 18.57 | 35.19 | 49.28 | 54.99 |
| ZeroSIAM | 20.04 | 38.35 | 54.09 | 60.53 |
| CMF | 18.36 | 36.55 | 53.39 | 60.63 |
| SAR | 13.39 | 28.84 | 45.58 | 53.67 |
| SPA | 11.55 | 26.60 | 45.55 | 55.87 |

*Table 34.* Amortised utility (%) across overhead budgets ($B$), aggregated across 15 corruptions (ViT-B/16).

| Method | $B = 2.5$s | $B = 5$s | $B = 10$s | $B = 20$s | $B = 40$s | $B = 80$s |
|---|---|---|---|---|---|---|
| Standard | 55.53 | 55.53 | 55.53 | 55.53 | 55.53 | 55.53 |
| NEO | **59.20** | 59.20 | 59.20 | 59.20 | 59.20 | 59.20 |
| LAME | 54.12 | 54.12 | 54.12 | 54.12 | 54.12 | 54.12 |
| Tent | 56.16 | 55.77 | 56.92 | 58.44 | 59.49 | 60.00 |
| ETA | 57.89 | **61.95** | **64.94** | **66.94** | 68.06 | 68.40 |
| SHOT-IM | 58.74 | 56.07 | 56.32 | 55.79 | 55.18 | 53.55 |
| DeYO | 54.39 | 56.70 | 58.78 | 60.14 | 61.02 | 61.25 |
| ZeroSIAM | 53.36 | 59.66 | 63.42 | 65.41 | 66.70 | 67.41 |
| CMF | 57.46 | 59.31 | 62.67 | 65.13 | 67.49 | 68.55 |
| SAR | 56.02 | 56.30 | 55.08 | 56.16 | 56.71 | 58.50 |
| SPA | 58.19 | 60.37 | 63.91 | 66.64 | **68.48** | **69.30** |

Gradient-free values are constant across budgets ($\beta = 100\%$ at all $B$).

Table 35 summarises temporal decomposition terms under strict pressure; they are timing-derived and nearly constant across corruptions. Under amortised evaluation (B=2.5,s), gradient-based methods exhaust their adaptation budget within the first 6–23 batches of 781 total (<3% of the stream). This explains why stable frozen-state accuracy determines overall utility under tight resource pools. However, unlike the severe backward-pass latency collapses observed with ResNet-50, the structural stability of layer normalisation in the ViT-B/16 backbone allows high-overhead methods like SPA to steadily claw back utility and eventually rank first as the amortised budget expands to B=80,s.

*Table 35.* Temporal decomposition terms under strict pressure ($\rho=100\%$, $T=200$ ms, $B=2.5$ s), averaged across ImageNet-C (ViT-B/16).

| | Gradient-Free | | | Gradient-Based | | | | | | | |
|---|---|---|---|---|---|---|---|---|---|---|---|
| **Factor** | Standard | NEO | LAME | Tent | ETA | SHOT-IM | DeYO | ZeroSIAM | CMF | SAR | SPA |
| $\alpha$ (%) at $\rho=100\%$ | 100.0 | 100.0 | 100.0 | 46.0 | 49.7 | 41.5 | 34.3 | 32.0 | 28.6 | 23.1 | 18.2 |
| $\bar{\kappa}$ (%) at $T=200$ ms | 100.0 | 100.0 | 100.0 | 43.1 | 43.3 | 38.7 | 31.6 | 29.6 | 26.6 | 21.0 | 16.5 |
| $m$ at $B=2.5$ s | – | – | – | 19 | 23 | 16 | 15 | 11 | 9 | 7 | 6 |

## C.6. Per-Corruption Analysis (ViT)

Table 36 reports the winning method for each corruption across 15 scenarios. NEO wins 81 of 225 cases (36.0%), dominating high-pressure settings due to its minimal overhead, while ETA (29.7%) and SPA (19.1%) excel as constraints relax. Figure 5 shows this phase transition from gradient-free to gradient-based winners, with more methods underperforming standard inference than observed on ResNet-50. This drop is less pronounced under amortised evaluation, where the layer-normalised ViT backbone largely protects gradient-based methods from degrading below the standard baseline after budget exhaustion.

Tables 37–42 decompose utility under strict temporal pressure; relaxed scenarios move steadily toward offline performance and are less informative. For gradient-free methods, the decomposition is trivial because minimal overhead yields $\bar{a} \approx U$. Under amortised evaluation ($B = 2.5$ s), gradient-based strategies exhaust their entire adaptation budget within the first 6–23 batches, leaving over 91% of the remaining stream to run on frozen parameters.

Table 43 aggregates losses and the average utility deficit relative to the cell winner across all 225 cases. This risk profile is contextualised by Table 44, which tracks how often and where absolute utility drops below standard baseline inference. Together, these failure dynamics explain the shifting rank correlations shown in Table 45, which measures the Spearman correlation ($r$) between offline and temporal rankings. Under severe real-time pressure ($\rho = 100\%$, $T = 200$ ms), we observe strong negative correlations (mean $r = -0.52$ and $-0.48$), but these correlations shift to positive values of 0.57, 0.40, and 0.98 as constraints relax to $\rho = 25\%$, $T = 2000$ ms, and $B \geq 40$ s, respectively.

*Table 36.* Winning method per corruption across 16 scenarios; win counts in legend.

| | Offline | Discrete ($\rho$ %) | | | | | Continuous ($T$ ms) | | | | Amortised ($B$ s) | | | | | |
|---|---|---|---|---|---|---|---|---|---|---|---|---|---|---|---|---|
| Corruption | | 100 | 70 | 50 | 35 | 25 | 200 | 400 | 1k | 2k | 2.5 | 5 | 10 | 20 | 40 | 80 |
| Gaussian noise | P | N | N | N | E | C | N | N | N | E | P | E | E | P | P | P |
| Shot noise | P | N | N | E | E | C | N | N | N | E | P | P | E | P | P | P |
| Impulse noise | P | N | N | E | E | C | N | N | N | E | P | P | P | P | P | P |
| Defocus blur | C | N | N | E | E | C | N | N | E | E | Z | E | E | E | E | C |
| Glass blur | P | N | E | E | E | C | N | E | E | E | S | E | E | E | E | P |
| Motion blur | P | N | N | E | E | C | N | N | E | E | S | E | P | E | P | P |
| Zoom blur | P | N | N | E | E | E | N | N | E | E | S | S | E | E | E | P |
| Snow | P | N | N | E | E | C | N | N | N | E | N | E | P | P | P | P |
| Frost | P | N | N | N | E | C | N | N | IM | E | N | P | P | P | P | P |
| Fog | P | N | N | N | E | C | N | N | N | N | N | N | N | N | P | P |
| Brightness | P | N | N | N | E | C | N | N | N | N | S | E | E | P | P | P |
| Contrast | C | N | N | T | T | C | N | N | T | T | N | E | E | E | C | C |
| Elastic transform | P | N | N | E | E | E | N | N | E | E | S | S | S | P | P | P |
| Pixelate | P | N | N | E | E | E | N | N | N | E | S | S | S | P | P | P |
| JPEG compression | P | N | N | N | E | C | N | N | N | E | S | S | S | P | P | P |

**N** EO (81) **E** TA (67) S **P** A (56) **C** MF (17) **S** HOT-IM (14) **T** ent (4) **Z** eroSIAM (1)

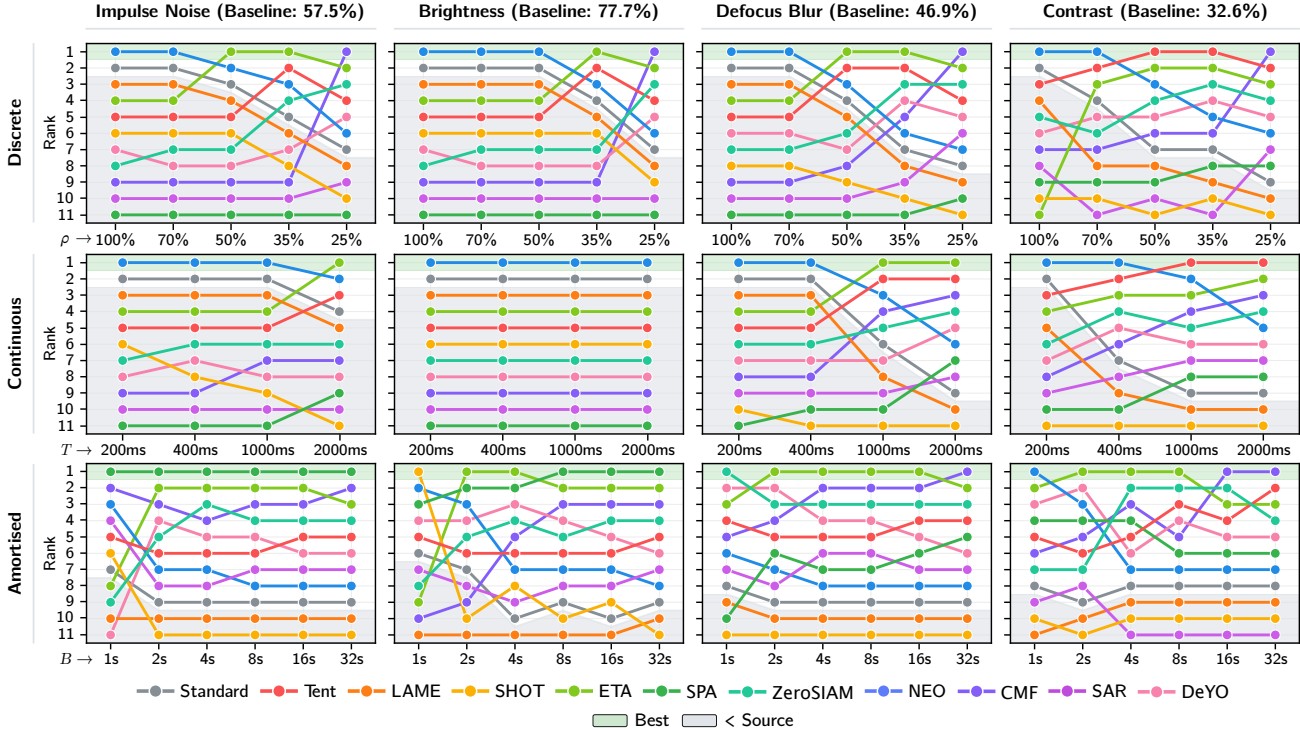

*Figure 5.* **Rank instability persists across temporal scenarios and corruption types on ViT-B/16.** Rows show discrete (utilisation $\rho$), continuous (threshold $T$), and amortised (budget $B$) evaluations; columns show one corruption from each category, spanning baseline accuracies from 32.6% to 77.7%. Green regions mark the best method; grey regions mark methods worse than standard inference. At relaxed thresholds, rankings converge towards offline; under pressure, instability increases. No method dominates. Best viewed in colour.

*Table 37.* Discrete utility (%) decomposition at $\rho = 100\%$ (gradient-free, ViT-B/16).

| | Standard | | LAME | | NEO | |
|---|---|---|---|---|---|---|
| **Corruption** | $\bar{a}_s$ | $U_d$ | $\bar{a}_s$ | $U_d$ | $\bar{a}_s$ | $U_d$ |
| Gaussian noise | 56.8 | 56.8 | 56.4 | 56.4 | 57.5 | 57.5 |
| Shot noise | 56.8 | 56.8 | 56.5 | 56.5 | 57.6 | 57.6 |
| Impulse noise | 57.5 | 57.5 | 57.2 | 57.2 | 58.3 | 58.3 |
| Defocus blur | 46.9 | 46.9 | 46.3 | 46.3 | 49.7 | 49.7 |
| Glass blur | 35.6 | 35.6 | 34.8 | 34.8 | 38.2 | 38.2 |
| Motion blur | 53.1 | 53.1 | 52.7 | 52.7 | 54.8 | 54.8 |
| Zoom blur | 44.8 | 44.8 | 44.1 | 44.1 | 47.6 | 47.6 |
| Snow | 62.2 | 62.2 | 57.0 | 57.0 | 64.7 | 64.7 |
| Frost | 62.6 | 62.6 | 61.6 | 61.6 | 65.0 | 65.0 |
| Fog | 65.7 | 65.7 | 63.8 | 63.8 | 71.3 | 71.3 |
| Brightness | 77.7 | 77.7 | 77.4 | 77.4 | 78.1 | 78.1 |
| Contrast | 32.6 | 32.6 | 25.7 | 25.7 | 58.6 | 58.6 |
| Elastic transform | 46.0 | 46.0 | 44.4 | 44.4 | 50.0 | 50.0 |
| Pixelate | 67.0 | 67.0 | 66.6 | 66.6 | 67.7 | 67.7 |
| JPEG compression | 67.6 | 67.6 | 67.3 | 67.3 | 68.8 | 68.8 |

*Table 38.* Continuous utility (%) decomposition at $T = 200$ ms (gradient-free, ViT-B/16).

| | Standard | | LAME | | NEO | |
|---|---|---|---|---|---|---|
| **Corruption** | $\bar{a}$ | $U_c$ | $\bar{a}$ | $U_c$ | $\bar{a}$ | $U_c$ |
| Gaussian noise | 56.8 | 56.8 | 56.4 | 56.4 | 57.5 | 57.5 |
| Shot noise | 56.8 | 56.8 | 56.5 | 56.5 | 57.6 | 57.6 |
| Impulse noise | 57.5 | 57.5 | 57.2 | 57.2 | 58.3 | 58.3 |
| Defocus blur | 46.9 | 46.9 | 46.3 | 46.3 | 49.7 | 49.7 |
| Glass blur | 35.6 | 35.6 | 34.8 | 34.8 | 38.2 | 38.2 |
| Motion blur | 53.1 | 53.1 | 52.7 | 52.7 | 54.8 | 54.8 |
| Zoom blur | 44.8 | 44.8 | 44.1 | 44.1 | 47.6 | 47.6 |
| Snow | 62.2 | 62.2 | 57.0 | 57.0 | 64.7 | 64.7 |
| Frost | 62.6 | 62.6 | 61.6 | 61.6 | 65.0 | 65.0 |
| Fog | 65.7 | 65.7 | 63.8 | 63.8 | 71.3 | 71.3 |
| Brightness | 77.7 | 77.7 | 77.4 | 77.4 | 78.1 | 78.1 |
| Contrast | 32.6 | 32.6 | 25.7 | 25.7 | 58.6 | 58.6 |
| Elastic transform | 46.0 | 46.0 | 44.4 | 44.4 | 50.0 | 50.0 |
| Pixelate | 67.0 | 67.0 | 66.6 | 66.6 | 67.7 | 67.7 |
| JPEG compression | 67.6 | 67.6 | 67.3 | 67.3 | 68.8 | 68.8 |

*Table 39.* Discrete utility (%) decomposition at $\rho=100\%$ (gradient-based, ViT-B/16).

| Corruption | Tent | | ETA | | SHOT-IM | | DeYO | | ZeroSIAM | | CMF | | SAR | | SPA | |
|---|---|---|---|---|---|---|---|---|---|---|---|---|---|---|---|---|
| | $\bar{a}_s$ | $U_d$ | $\bar{a}_s$ | $U_d$ | $\bar{a}_s$ | $U_d$ | $\bar{a}_s$ | $U_d$ | $\bar{a}_s$ | $U_d$ | $\bar{a}_s$ | $U_d$ | $\bar{a}_s$ | $U_d$ | $\bar{a}_s$ | $U_d$ |
| Gaussian noise | 59.2 | 27.2 | 61.2 | 28.1 | 53.2 | 22.1 | 58.5 | 19.5 | 59.7 | 19.2 | 60.8 | 17.4 | 58.2 | 13.4 | 61.9 | 11.3 |
| Shot noise | 60.2 | 27.7 | 62.8 | 28.9 | 56.5 | 23.4 | 60.3 | 20.0 | 61.6 | 19.8 | 62.0 | 17.7 | 59.5 | 13.7 | 63.6 | 11.6 |
| Impulse noise | 60.3 | 27.7 | 63.1 | 29.0 | 56.5 | 23.4 | 60.5 | 20.0 | 61.6 | 19.7 | 62.8 | 18.0 | 59.1 | 13.6 | 63.5 | 11.5 |
| Defocus blur | 57.3 | 26.4 | 60.0 | 27.6 | 42.5 | 17.7 | 57.3 | 19.1 | 58.6 | 18.8 | 59.2 | 17.0 | 54.5 | 12.6 | 56.4 | 10.3 |
| Glass blur | 52.9 | 24.3 | 59.3 | 27.2 | 49.1 | 20.4 | 56.7 | 18.9 | 57.6 | 18.4 | 57.3 | 16.4 | 49.4 | 11.4 | 58.1 | 10.6 |
| Motion blur | 61.2 | 28.1 | 65.0 | 29.9 | 56.8 | 23.6 | 62.9 | 20.7 | 63.3 | 20.3 | 63.7 | 18.3 | 58.4 | 13.5 | 65.9 | 12.0 |
| Zoom blur | 56.7 | 26.1 | 61.9 | 28.4 | 57.4 | 23.8 | 58.5 | 19.4 | 60.3 | 19.3 | 59.4 | 16.8 | 52.7 | 12.1 | 62.6 | 11.4 |
| Snow | 64.0 | 29.4 | 69.2 | 31.9 | 64.2 | 26.6 | 67.5 | 22.1 | 68.2 | 21.8 | 68.4 | 19.1 | 64.2 | 14.8 | 71.6 | 13.0 |
| Frost | 62.2 | 28.6 | 66.9 | 30.8 | 62.1 | 25.8 | 64.1 | 21.2 | 65.3 | 20.9 | 67.6 | 19.1 | 59.9 | 13.8 | 70.1 | 12.7 |
| Fog | 5.0 | 2.3 | 69.5 | 34.2 | 56.2 | 23.4 | 3.0 | 1.6 | 69.1 | 22.1 | 70.6 | 20.2 | 26.8 | 6.2 | 73.0 | 13.3 |
| Brightness | 79.1 | 36.4 | 79.7 | 36.6 | 77.8 | 32.3 | 78.8 | 25.5 | 79.3 | 25.4 | 79.4 | 23.0 | 78.1 | 18.0 | 80.4 | 14.6 |
| Contrast | 65.2 | 30.0 | 0.6 | 0.6 | 5.7 | 2.4 | 61.6 | 20.5 | 65.4 | 20.9 | 66.2 | 19.0 | 63.8 | 12.5 | 61.3 | 11.1 |
| Elastic transform | 56.7 | 26.1 | 68.2 | 31.3 | 66.8 | 27.7 | 65.8 | 21.7 | 64.8 | 20.7 | 64.6 | 18.4 | 53.8 | 12.4 | 68.5 | 12.4 |
| Pixelate | 71.1 | 32.7 | 74.7 | 34.3 | 72.3 | 30.0 | 73.1 | 23.9 | 73.2 | 23.4 | 73.2 | 21.1 | 69.1 | 15.9 | 75.7 | 13.8 |
| JPEG compression | 69.9 | 32.1 | 72.4 | 33.3 | 70.6 | 29.4 | 70.6 | 23.1 | 70.9 | 22.7 | 71.1 | 20.5 | 68.5 | 15.8 | 73.2 | 13.3 |

*Table 40.* Continuous utility (%) decomposition at $T=200$ ms (gradient-based, ViT-B/16).

| Corruption | Tent | | ETA | | SHOT-IM | | DeYO | | ZeroSIAM | | CMF | | SAR | | SPA | |
|---|---|---|---|---|---|---|---|---|---|---|---|---|---|---|---|---|
| | $\bar{a}$ | $U_c$ | $\bar{a}$ | $U_c$ | $\bar{a}$ | $U_c$ | $\bar{a}$ | $U_c$ | $\bar{a}$ | $U_c$ | $\bar{a}$ | $U_c$ | $\bar{a}$ | $U_c$ | $\bar{a}$ | $U_c$ |
| Gaussian noise | 60.2 | 26.0 | 62.9 | 27.1 | 49.5 | 19.2 | 59.1 | 18.0 | 62.4 | 18.5 | 63.5 | 16.9 | 59.2 | 12.5 | 63.8 | 10.5 |
| Shot noise | 61.6 | 26.6 | 64.2 | 27.7 | 51.8 | 20.1 | 61.1 | 18.6 | 63.8 | 18.9 | 64.9 | 17.3 | 60.4 | 12.7 | 65.3 | 10.8 |
| Impulse noise | 61.8 | 26.7 | 64.2 | 27.8 | 52.4 | 20.3 | 61.2 | 18.5 | 64.0 | 18.9 | 64.8 | 17.3 | 60.5 | 12.7 | 65.4 | 10.8 |
| Defocus blur | 59.2 | 25.5 | 61.3 | 26.4 | 30.8 | 11.9 | 57.8 | 17.5 | 60.1 | 17.8 | 62.2 | 16.6 | 57.6 | 12.1 | 61.1 | 10.1 |
| Glass blur | 56.7 | 24.4 | 61.9 | 26.6 | 46.0 | 17.8 | 59.1 | 17.9 | 60.7 | 17.9 | 62.0 | 16.5 | 56.0 | 11.8 | 63.8 | 10.5 |
| Motion blur | 63.5 | 27.3 | 66.8 | 28.7 | 49.2 | 19.0 | 64.5 | 19.4 | 66.1 | 19.5 | 66.8 | 17.8 | 61.8 | 13.0 | 69.2 | 11.4 |
| Zoom blur | 59.3 | 25.5 | 65.0 | 27.9 | 53.7 | 20.7 | 62.1 | 18.7 | 63.4 | 18.7 | 64.8 | 17.1 | 58.0 | 12.2 | 68.0 | 11.2 |
| Snow | 59.9 | 25.8 | 71.4 | 30.7 | 61.8 | 23.9 | 69.0 | 20.7 | 70.5 | 20.8 | 71.6 | 18.9 | 65.9 | 13.8 | 74.2 | 12.2 |
| Frost | 64.1 | 27.6 | 69.5 | 29.9 | 58.7 | 22.7 | 66.6 | 20.1 | 68.6 | 20.3 | 70.4 | 18.6 | 63.8 | 13.4 | 72.0 | 11.9 |
| Fog | 2.4 | 1.1 | 72.3 | 33.3 | 53.8 | 20.8 | 1.8 | 1.7 | 72.8 | 21.5 | 74.8 | 19.9 | 67.2 | 14.1 | 75.6 | 12.5 |
| Brightness | 79.2 | 34.1 | 80.3 | 34.5 | 77.2 | 29.9 | 79.0 | 23.6 | 80.0 | 23.7 | 80.3 | 21.5 | 78.6 | 16.5 | 80.9 | 13.3 |
| Contrast | 67.3 | 28.9 | 66.4 | 28.6 | 1.6 | 0.6 | 64.4 | 19.4 | 66.3 | 19.6 | 69.3 | 18.5 | 64.3 | 13.5 | 65.3 | 10.7 |
| Elastic transform | 61.3 | 26.4 | 70.9 | 30.5 | 66.2 | 25.7 | 68.5 | 20.6 | 69.6 | 20.6 | 70.5 | 18.7 | 61.3 | 12.9 | 74.1 | 12.2 |
| Pixelate | 72.7 | 31.3 | 75.8 | 32.6 | 70.5 | 27.3 | 74.2 | 22.2 | 75.5 | 22.3 | 75.8 | 20.2 | 72.2 | 15.1 | 77.7 | 12.8 |
| JPEG compression | 70.7 | 30.4 | 73.7 | 31.7 | 69.1 | 26.7 | 72.2 | 21.6 | 72.7 | 21.5 | 73.7 | 19.6 | 70.1 | 14.7 | 75.0 | 12.4 |

*Table 41.* Amortised utility (%) decomposition at $B=2500$ ms (gradient-free, ViT-B/16).

| Corruption | Standard | | | LAME | | | NEO | | |
|---|---|---|---|---|---|---|---|---|---|
| | $\bar{a}_a$ | $\bar{a}_f$ | $U_a$ | $\bar{a}_a$ | $\bar{a}_f$ | $U_a$ | $\bar{a}_a$ | $\bar{a}_f$ | $U_a$ |
| Gaussian noise | 56.8 | – | 56.8 | 56.4 | – | 56.4 | 57.5 | – | 57.5 |
| Shot noise | 56.8 | – | 56.8 | 56.5 | – | 56.5 | 57.6 | – | 57.6 |
| Impulse noise | 57.5 | – | 57.5 | 57.2 | – | 57.2 | 58.3 | – | 58.3 |
| Defocus blur | 46.9 | – | 46.9 | 46.3 | – | 46.3 | 49.7 | – | 49.7 |
| Glass blur | 35.6 | – | 35.6 | 34.8 | – | 34.8 | 38.2 | – | 38.2 |
| Motion blur | 53.1 | – | 53.1 | 52.7 | – | 52.7 | 54.8 | – | 54.8 |
| Zoom blur | 44.8 | – | 44.8 | 44.1 | – | 44.1 | 47.6 | – | 47.6 |
| Snow | 62.2 | – | 62.2 | 57.0 | – | 57.0 | 64.7 | – | 64.7 |
| Frost | 62.6 | – | 62.6 | 61.6 | – | 61.6 | 65.0 | – | 65.0 |
| Fog | 65.7 | – | 65.7 | 63.8 | – | 63.8 | 71.3 | – | 71.3 |
| Brightness | 77.7 | – | 77.7 | 77.4 | – | 77.4 | 78.1 | – | 78.1 |
| Contrast | 32.6 | – | 32.6 | 25.7 | – | 25.7 | 58.6 | – | 58.6 |
| Elastic transform | 46.0 | – | 46.0 | 44.4 | – | 44.4 | 50.0 | – | 50.0 |
| Pixelate | 67.0 | – | 67.0 | 66.6 | – | 66.6 | 67.7 | – | 67.7 |
| JPEG compression | 67.6 | – | 67.6 | 67.3 | – | 67.3 | 68.8 | – | 68.8 |

Gradient-free methods never exhaust the budget; $\bar{a}_f$ is undefined and utility follows offline rankings.

*Table 42.* Amortised utility (%) decomposition at $B=2500$ ms (gradient-based, ViT-B/16).

| Corruption | Tent | | | ETA | | | SHOT-IM | | | DeYO | | | ZeroSIAM | | | CMF | | | SAR | | | SPA | | |
|---|---|---|---|---|---|---|---|---|---|---|---|---|---|---|---|---|---|---|---|---|---|---|---|---|
| | $\bar{a}_a$ | $\bar{a}_f$ | $U_a$ | $\bar{a}_a$ | $\bar{a}_f$ | $U_a$ | $\bar{a}_a$ | $\bar{a}_f$ | $U_a$ | $\bar{a}_a$ | $\bar{a}_f$ | $U_a$ | $\bar{a}_a$ | $\bar{a}_f$ | $U_a$ | $\bar{a}_a$ | $\bar{a}_f$ | $U_a$ | $\bar{a}_a$ | $\bar{a}_f$ | $U_a$ | $\bar{a}_a$ | $\bar{a}_f$ | $U_a$ |
| Gaussian noise | 54.2 | 56.6 | 56.5 | 54.6 | 56.5 | 56.4 | 56.1 | 56.0 | 56.0 | 54.6 | 55.8 | 55.8 | 54.0 | 55.6 | 55.5 | 55.9 | 56.9 | 56.9 | 54.7 | 57.2 | 57.1 | 56.0 | 58.5 | 58.5 |
| Shot noise | 56.2 | 57.9 | 57.8 | 56.0 | 57.7 | 57.7 | 57.7 | 58.1 | 58.1 | 55.5 | 58.0 | 58.0 | 56.2 | 56.9 | 56.9 | 57.3 | 58.0 | 58.0 | 56.2 | 57.3 | 57.3 | 57.3 | 59.9 | 59.9 |
| Impulse noise | 56.8 | 58.0 | 58.0 | 56.9 | 57.5 | 57.5 | 59.1 | 57.5 | 57.6 | 57.7 | 57.0 | 57.0 | 58.1 | 57.2 | 57.2 | 59.9 | 58.3 | 58.3 | 58.7 | 58.1 | 58.1 | 59.9 | 60.5 | 60.5 |
| Defocus blur | 48.8 | 50.6 | 50.5 | 50.7 | 53.5 | 53.4 | 49.8 | 45.1 | 45.1 | 50.8 | 53.8 | 53.7 | 49.6 | 54.2 | 54.1 | 51.7 | 50.0 | 50.0 | 49.8 | 47.0 | 47.0 | 48.7 | 45.8 | 45.9 |
| Glass blur | 38.6 | 40.0 | 39.9 | 41.7 | 47.2 | 47.1 | 51.2 | 53.0 | 52.9 | 41.3 | 46.3 | 46.2 | 41.2 | 43.7 | 43.6 | 41.2 | 42.3 | 42.2 | 40.0 | 37.7 | 37.7 | 41.7 | 39.8 | 39.8 |
| Motion blur | 53.8 | 55.9 | 55.8 | 55.3 | 58.1 | 58.0 | 57.0 | 58.1 | 58.1 | 52.3 | 57.1 | 57.0 | 53.4 | 57.7 | 57.6 | 54.9 | 55.5 | 55.5 | 54.9 | 53.9 | 53.9 | 54.4 | 55.6 | 55.5 |
| Zoom blur | 46.1 | 48.4 | 48.3 | 46.2 | 50.9 | 50.8 | 54.0 | 57.4 | 57.3 | 45.7 | 48.7 | 48.7 | 47.9 | 50.2 | 50.2 | 47.7 | 48.1 | 48.1 | 46.4 | 46.4 | 46.4 | 49.0 | 47.5 | 47.5 |
| Snow | 60.9 | 62.4 | 62.4 | 61.5 | 63.0 | 62.9 | 57.1 | 60.1 | 60.0 | 61.6 | 62.8 | 62.7 | 61.6 | 62.7 | 62.7 | 61.6 | 62.5 | 62.5 | 60.7 | 62.3 | 62.3 | 61.7 | 62.3 | 62.3 |
| Frost | 59.0 | 58.1 | 58.1 | 57.2 | 56.0 | 56.0 | 59.0 | 58.6 | 58.6 | 57.6 | 56.1 | 56.1 | 57.5 | 54.6 | 54.7 | 59.9 | 57.5 | 57.5 | 59.6 | 62.4 | 62.3 | 60.2 | 64.1 | 64.0 |
| Fog | 52.7 | 44.1 | 44.3 | 61.6 | 47.2 | 48.5 | 60.4 | 57.0 | 57.0 | 35.7 | 0.1 | 1.8 | 44.6 | 6.9 | 7.5 | 60.6 | 64.5 | 64.4 | 64.5 | 66.1 | 66.1 | 64.3 | 63.9 | 63.9 |
| Brightness | 76.1 | 77.7 | 77.7 | 75.8 | 77.6 | 77.6 | 77.0 | 78.5 | 78.5 | 75.1 | 77.8 | 77.8 | 75.4 | 77.7 | 77.7 | 75.2 | 77.5 | 77.5 | 74.1 | 77.7 | 77.7 | 75.5 | 77.9 | 77.9 |
| Contrast | 44.6 | 48.8 | 48.7 | 36.4 | 55.5 | 55.0 | 42.7 | 30.1 | 30.3 | 44.6 | 54.9 | 54.7 | 41.3 | 35.4 | 35.5 | 42.3 | 46.5 | 46.5 | 32.6 | 32.4 | 32.4 | 45.6 | 51.3 | 51.2 |
| Elastic transform | 48.8 | 48.3 | 48.3 | 49.5 | 51.3 | 51.2 | 61.5 | 67.2 | 67.1 | 49.5 | 50.4 | 50.4 | 48.2 | 50.7 | 50.6 | 49.8 | 49.0 | 49.0 | 47.3 | 46.9 | 46.9 | 47.4 | 49.4 | 49.4 |
| Pixelate | 66.9 | 67.6 | 67.6 | 66.4 | 68.0 | 67.9 | 70.5 | 73.3 | 73.2 | 66.0 | 67.4 | 67.3 | 66.3 | 68.0 | 68.0 | 68.2 | 67.4 | 67.4 | 67.6 | 67.2 | 67.2 | 67.4 | 68.2 | 68.2 |
| JPEG compression | 67.0 | 68.3 | 68.3 | 67.2 | 68.5 | 68.4 | 69.4 | 71.0 | 71.0 | 66.9 | 68.7 | 68.6 | 66.6 | 68.5 | 68.5 | 67.5 | 68.1 | 68.1 | 67.6 | 67.8 | 67.8 | 70.3 | 68.2 | 68.3 |

Methods exhaust budget within 6–68 batches of 781; > 91% of the stream runs on frozen inference.

*Table 43.* Performance deficit profile across 225 evaluation cells for ImageNet-C with ViT-B/16. We report the number of such cells where a method fails to rank first (losses) and its average utility deficit to the cell winner (Mean Δ, pp). Lower is better for both.

| Method | Overall Losses | Overall Mean Δ | Discrete Losses | Discrete Mean Δ | Continuous Losses | Continuous Mean Δ | Amortised Losses | Amortised Mean Δ |
|---|---|---|---|---|---|---|---|---|
| Standard | 225 | 8.80 | 75 | 8.33 | 60 | 5.88 | 90 | 11.14 |
| NEO | 144 | 8.02 | 41 | 8.53 | 20 | 6.63 | 83 | 8.10 |
| LAME | 225 | 10.21 | 75 | 9.74 | 60 | 7.29 | 90 | 12.55 |
| Tent | 221 | 13.66 | 73 | 16.27 | 58 | 17.80 | 90 | 8.87 |
| ETA | 158 | 9.79 | 48 | 14.85 | 42 | 15.63 | 68 | 2.61 |
| SHOT-IM | 211 | 19.43 | 75 | 22.60 | 60 | 24.01 | 76 | 12.70 |
| DeYO | 225 | 16.63 | 75 | 22.83 | 60 | 21.90 | 90 | 7.96 |
| ZeroSIAM | 224 | 13.36 | 75 | 20.56 | 60 | 18.15 | 89 | 4.05 |
| CMF | 210 | 14.72 | 63 | 26.17 | 60 | 19.17 | 87 | 3.35 |
| SAR | 225 | 22.03 | 75 | 33.00 | 60 | 26.04 | 90 | 10.21 |
| SPA | 182 | 24.64 | 75 | 35.95 | 60 | 26.51 | 47 | 4.19 |

*Table 44.* Number of temporal evaluations in which each method's utility falls below standard inference, and the share of those misses attributable to each evaluation category. ImageNet-C with ViT-B/16.

| Method | Total (225) | Discrete (75) | Continuous (60) | Amortised (90) |
|---|---|---|---|---|
| NEO | 0 | – | – | – |
| LAME | 225 | 33% | 27% | 40% |
| Tent | 98 | 43% | 44% | 13% |
| ETA | 72 | 42% | 49% | 10% |
| SHOT-IM | 161 | 39% | 34% | 28% |
| DeYO | 110 | 46% | 45% | 9% |
| ZeroSIAM | 101 | 48% | 45% | 8% |
| CMF | 107 | 50% | 43% | 7% |
| SAR | 138 | 50% | 38% | 12% |
| SPA | 125 | 57% | 41% | 2% |

*Table 45.* Spearman rank correlation ($r$) between offline and temporal rankings across 15 ImageNet-C corruptions on ViT-B/16. Due to layer normalisation, gradient-based methods are more stable under tight amortised budgets than they are on ResNet-50.

| Corruption | Discrete ($\rho$, %) 100 | 70 | 50 | 35 | 25 | Continuous ($T$, ms) 200 | 400 | 1000 | 2000 | Amortised ($B$, s) 2.5 | 5 | 10 | 20 | 40 | 80 |
|---|---|---|---|---|---|---|---|---|---|---|---|---|---|---|---|
| *Noise* | | | | | | | | | | | | | | | |
| Gaussian Noise | -0.61 | -0.58 | -0.46 | -0.13 | 0.41 | -0.58 | -0.40 | -0.35 | 0.05 | 0.26 | 0.73 | 0.86 | 0.93 | 0.98 | 0.99 |
| Shot Noise | -0.58 | -0.56 | -0.40 | 0.00 | 0.45 | -0.56 | -0.45 | -0.34 | 0.35 | 0.44 | 0.95 | 0.92 | 0.98 | 0.98 | 1.00 |
| Impulse Noise | -0.58 | -0.56 | -0.40 | -0.08 | 0.45 | -0.56 | -0.45 | -0.34 | 0.09 | 0.37 | 0.95 | 0.95 | 0.98 | 0.99 | 1.00 |
| *Blur* | | | | | | | | | | | | | | | |
| Defocus Blur | -0.39 | -0.39 | 0.02 | 0.49 | 0.75 | -0.25 | -0.17 | 0.49 | 0.84 | 0.55 | 0.83 | 0.89 | 0.89 | 0.94 | 0.97 |
| Glass Blur | -0.71 | -0.47 | -0.10 | 0.32 | 0.75 | -0.64 | -0.28 | 0.77 | 0.85 | 0.53 | 0.65 | 0.71 | 0.91 | 0.97 | 0.99 |
| Motion Blur | -0.60 | -0.59 | -0.35 | 0.23 | 0.59 | -0.47 | -0.39 | 0.09 | 0.72 | 0.27 | 0.77 | 0.95 | 0.90 | 0.96 | 0.99 |
| Zoom Blur | -0.66 | -0.65 | -0.28 | 0.28 | 0.74 | -0.65 | -0.44 | 0.48 | 0.85 | 0.46 | 0.62 | 0.84 | 0.97 | 0.98 | 1.00 |
| *Weather* | | | | | | | | | | | | | | | |
| Snow | -0.62 | -0.61 | -0.45 | 0.03 | 0.41 | -0.61 | -0.61 | -0.12 | 0.50 | 0.51 | 0.73 | 0.89 | 0.87 | 0.91 | 0.98 |
| Frost | -0.48 | -0.48 | -0.35 | -0.17 | 0.45 | -0.47 | -0.47 | -0.18 | 0.22 | -0.26 | 0.12 | 0.61 | 0.86 | 0.93 | 0.99 |
| Fog | 0.21 | 0.21 | 0.21 | 0.16 | 0.72 | 0.21 | 0.27 | 0.46 | 0.60 | 0.35 | 0.31 | 0.72 | 0.88 | 0.94 | 0.97 |
| Brightness | -0.58 | -0.56 | -0.56 | -0.29 | 0.41 | -0.56 | -0.56 | -0.56 | -0.56 | -0.23 | 0.58 | 0.82 | 0.95 | 0.95 | 0.99 |
| *Digital* | | | | | | | | | | | | | | | |
| Contrast | -0.15 | 0.31 | 0.61 | 0.67 | 0.94 | 0.05 | 0.52 | 0.75 | 0.89 | 0.50 | 0.60 | 0.81 | 0.77 | 0.86 | 0.90 |
| Elastic transform | -0.61 | -0.58 | -0.25 | 0.27 | 0.74 | -0.68 | -0.45 | 0.55 | 0.85 | 0.60 | 0.79 | 0.79 | 0.90 | 0.98 | 0.99 |
| Pixelate | -0.66 | -0.66 | -0.44 | -0.05 | 0.41 | -0.65 | -0.65 | -0.65 | 0.05 | 0.55 | 0.66 | 0.69 | 0.94 | 0.99 | 0.99 |
| JPEG compression | -0.71 | -0.71 | -0.71 | -0.35 | 0.41 | -0.71 | -0.71 | -0.70 | -0.31 | 0.17 | 0.44 | 0.65 | 0.79 | 0.95 | 0.98 |
| **Mean** | -0.52 | -0.46 | -0.26 | 0.09 | 0.57 | -0.48 | -0.35 | 0.02 | 0.40 | 0.34 | 0.65 | 0.81 | 0.90 | 0.95 | 0.98 |
| **Std** | 0.24 | 0.30 | 0.33 | 0.29 | 0.18 | 0.27 | 0.33 | 0.52 | 0.46 | 0.27 | 0.23 | 0.11 | 0.06 | 0.04 | 0.02 |

## C.7. Additional Datasets (ImageNet-V2 & ImageNet-R)

To show the generality of our observations beyond just ImageNet-C corruptions, we extend our evaluation to natural distribution shifts and style variations using the ImageNet-V2 (Recht et al., 2019) and ImageNet-R (Hendrycks et al., 2021) datasets. Under ImageNet-V2, standard inference achieves relatively high accuracies of 63.2% on ResNet-50 and 75.5% on ViT-B/16. Conversely, on ImageNet-R, artistic abstractions and stylistic variations cause a severe baseline drop to 36.2% on ResNet-50 and 59.5% on ViT-B/16. We report the complete empirical results for these datasets across both architectures as follows: Table 46 lists offline accuracies for ResNet-50; Table 48 combines the discrete, continuous, and amortised utility decompositions under strict operational limits for ResNet-50; and Tables 50, 52, and 54 present the full parameter sweeps across all utilisation levels, temporal thresholds, and overhead budgets for ResNet-50. For the ViT-B/16 backbone, Table 47 details the offline accuracies; Table 49 combines the corresponding discrete, continuous, and amortised utility decompositions under strict operational limits for ViT-B/16; and Tables 51, 53, and 55 provide the complete sweeps across all operational scenarios. Crucially, rank instability persists across the 62 distinct temporal evaluations on these datasets, comprising $2 \times 16$ evaluations for ResNet-50 and $2 \times 15$ for ViT-B/16.

*Table 46.* Offline accuracy (%) on ImageNet-R and ImageNet-V2 (ResNet-50). Bold = best per dataset; underline = second-best.

| | *Gradient-free* | | | | *Gradient-based* | | | | | |
| Dataset | Standard | AdaBN | LAME | NEO | Tent | ETA | SHOT-IM | DeYO | CMF | SAR |
|---|---|---|---|---|---|---|---|---|---|---|
| ImageNet-R | 36.2 | 39.6 | 36.1 | 39.0 | 42.1 | 44.5 | 41.4 | 45.5 | **46.0** | 42.9 |
| ImageNet-V2 | 63.2 | 62.6 | 63.1 | **63.2** | 62.9 | 63.1 | 62.9 | 63.2 | 62.9 | 62.9 |

*Table 47.* Offline accuracy (%) on ImageNet-R and ImageNet-V2 (ViT-B/16). Bold = best per dataset; underline = second-best.

| | *Gradient-free* | | | | *Gradient-based* | | | | | | |
| Dataset | Standard | LAME | NEO | Tent | ETA | SHOT-IM | DeYO | ZeroSIAM | CMF | SAR | SPA |
|---|---|---|---|---|---|---|---|---|---|---|---|
| ImageNet-R | 59.5 | 59.3 | 60.5 | 62.2 | 65.6 | 52.7 | 65.9 | 63.5 | **66.2** | 62.2 | 65.7 |
| ImageNet-V2 | 75.5 | 75.3 | 75.5 | 75.2 | 74.6 | 72.4 | 73.8 | 74.9 | 74.6 | 74.9 | **75.5** |

*Table 48.* Utility decompositions across ImageNet-R and ImageNet-V2 datasets using a ResNet-50 backbone under strict constraints.

| | Discrete ($\rho = 100\%$) | | | Continuous ($T = 50\,\mathrm{ms}$) | | | | Amortised ($B = 1\,\mathrm{s}$) | | | |
| Method | $\alpha$ | $\bar{a}_s$ | $U_\mathrm{d}$ | $\bar{a}$ | $\bar{\kappa}$ | Cov | $U_\mathrm{c}$ | $\beta$ | $\bar{a}_\mathrm{a}$ | $\bar{a}_\mathrm{f}$ | $U_\mathrm{a}$ |
|---|---|---|---|---|---|---|---|---|---|---|---|
| *ImageNet-R* | | | | | | | | | | | |
| Standard | 99.8 | 36.2 | 36.1 | 36.16 | 98.6 | -0.0000 | 35.67 | 100.0 | 36.2 | – | 36.2 |
| AdaBN | 94.2 | 39.7 | 37.4 | 39.62 | 80.2 | 0.0000 | 31.77 | 100.0 | 39.7 | – | 39.7 |
| LAME | 96.4 | 36.1 | 34.8 | 36.10 | 86.7 | 0.0002 | 31.31 | 100.0 | 36.1 | – | 36.1 |
| NEO | 99.6 | 39.0 | 38.8 | 38.99 | 98.3 | 0.0000 | 38.33 | 100.0 | 39.0 | – | 39.0 |
| Tent | 41.5 | 41.1 | 17.0 | 42.05 | 15.2 | -0.0000 | 6.38 | 3.8 | 39.3 | 0.5 | 1.9 |
| ETA | 41.0 | 44.1 | 18.1 | 44.50 | 15.1 | -0.0001 | 6.69 | 3.8 | 40.0 | 0.5 | 2.0 |
| SHOT-IM | 33.3 | 42.1 | 14.0 | 41.36 | 11.3 | -0.0000 | 4.66 | 2.8 | 40.4 | 40.6 | 40.6 |
| DeYO | 32.7 | 44.6 | 14.6 | 45.52 | 10.8 | -0.0001 | 4.90 | 2.8 | 40.4 | 0.4 | 1.6 |
| CMF | 25.6 | 44.3 | 11.4 | 46.00 | 8.2 | -0.0001 | 3.74 | 1.9 | 41.5 | 0.4 | 1.2 |
| SAR | 20.7 | 41.3 | 8.6 | 42.88 | 6.3 | -0.0000 | 2.70 | 1.5 | 42.2 | 0.4 | 1.1 |
| *ImageNet-V2* | | | | | | | | | | | |
| Standard | 100.0 | 63.2 | 63.2 | 63.19 | 99.0 | 0.0000 | 62.54 | 100.0 | 63.2 | – | 63.2 |
| AdaBN | 94.2 | 62.9 | 59.3 | 62.62 | 80.4 | 0.0000 | 50.33 | 100.0 | 62.6 | – | 62.6 |
| LAME | 98.1 | 63.0 | 61.8 | 63.05 | 92.3 | 0.0003 | 58.24 | 100.0 | 63.1 | – | 63.1 |
| NEO | 100.0 | 63.2 | 63.2 | 63.21 | 98.4 | -0.0000 | 62.17 | 100.0 | 63.2 | – | 63.2 |
| Tent | 41.7 | 62.3 | 26.0 | 62.92 | 15.5 | 0.0001 | 9.77 | 11.5 | 63.2 | 0.1 | 7.4 |
| ETA | 41.7 | 63.1 | 26.3 | 63.06 | 15.4 | 0.0000 | 9.70 | 11.5 | 63.4 | 0.1 | 7.4 |
| SHOT-IM | 34.0 | 61.3 | 20.8 | 62.87 | 11.6 | 0.0001 | 7.31 | 8.3 | 62.3 | 63.6 | 63.5 |
| DeYO | 30.1 | 63.7 | 19.2 | 63.18 | 10.0 | 0.0000 | 6.35 | 7.1 | 62.8 | 0.1 | 4.5 |
| CMF | 25.6 | 63.5 | 16.3 | 62.93 | 8.4 | 0.0001 | 5.29 | 5.8 | 62.8 | 0.1 | 3.7 |
| SAR | 21.2 | 61.3 | 13.0 | 62.88 | 6.7 | 0.0001 | 4.19 | 4.5 | 62.1 | 0.1 | 2.9 |

*Table 49.* Utility decompositions across ImageNet-R and ImageNet-V2 datasets using a ViT-B/16 backbone under strict scenarios.

| Method | Discrete ($\rho = 100\%$) | | | Continuous ($T = 200\,\text{ms}$) | | | | Amortised ($B = 2.5\,\text{s}$) | | | |
| --- | --- | --- | --- | --- | --- | --- | --- | --- | --- | --- | --- |
| | $\alpha$ | $\bar{a}_s$ | $U_\text{d}$ | $\bar{a}$ | $\bar{\kappa}$ | Cov | $U_\text{c}$ | $\beta$ | $\bar{a}_\text{a}$ | $\bar{a}_\text{f}$ | $U_\text{a}$ |
| *ImageNet-R* | | | | | | | | | | | |
| Standard | 100.0 | 59.5 | 59.5 | 59.47 | 100.0 | 0.0000 | 59.47 | 100.0 | 59.5 | – | 59.5 |
| LAME | 100.0 | 59.3 | 59.3 | 59.30 | 100.0 | 0.0000 | 59.30 | 100.0 | 59.3 | – | 59.3 |
| NEO | 100.0 | 60.5 | 60.5 | 60.46 | 100.0 | 0.0000 | 60.46 | 100.0 | 60.5 | – | 60.5 |
| Tent | 46.2 | 61.0 | 28.2 | 62.22 | 43.5 | -0.0000 | 27.06 | 4.1 | 58.8 | 59.8 | 59.8 |
| ETA | 46.4 | 63.6 | 29.5 | 65.55 | 43.5 | -0.0000 | 28.51 | 4.1 | 58.5 | 60.3 | 60.2 |
| SHOT-IM | 41.9 | 60.9 | 25.5 | 52.71 | 39.1 | 0.0001 | 20.60 | 3.4 | 60.4 | 65.0 | 64.9 |
| DeYO | 34.2 | 64.5 | 22.0 | 65.89 | 31.0 | -0.0002 | 20.42 | 2.8 | 58.3 | 60.3 | 60.2 |
| ZeroSIAM | 32.3 | 62.8 | 20.3 | 63.50 | 29.8 | -0.0000 | 18.94 | 2.4 | 58.9 | 60.1 | 60.1 |
| CMF | 26.9 | 61.7 | 16.6 | 66.22 | 25.7 | 0.0001 | 17.03 | 1.9 | 58.9 | 59.8 | 59.8 |
| SAR | 23.3 | 59.8 | 13.9 | 62.21 | 21.3 | 0.0000 | 13.24 | 1.5 | 58.3 | 59.6 | 59.6 |
| SPA | 18.4 | 64.4 | 11.8 | 65.74 | 16.7 | -0.0001 | 10.95 | 1.3 | 58.1 | 60.1 | 60.0 |
| *ImageNet-V2* | | | | | | | | | | | |
| Standard | 100.0 | 75.5 | 75.5 | 75.50 | 100.0 | 0.0000 | 75.50 | 100.0 | 75.5 | – | 75.5 |
| LAME | 100.0 | 75.3 | 75.3 | 75.34 | 100.0 | 0.0000 | 75.34 | 100.0 | 75.3 | – | 75.3 |
| NEO | 100.0 | 75.5 | 75.5 | 75.45 | 100.0 | 0.0000 | 75.45 | 100.0 | 75.5 | – | 75.5 |
| Tent | 46.8 | 74.9 | 35.0 | 75.15 | 43.7 | -0.0001 | 32.83 | 12.2 | 75.1 | 75.4 | 75.3 |
| ETA | 46.8 | 74.8 | 35.0 | 74.57 | 43.6 | -0.0001 | 32.52 | 12.2 | 75.0 | 74.9 | 74.9 |
| SHOT-IM | 42.3 | 73.2 | 30.9 | 72.42 | 39.3 | -0.0000 | 28.49 | 10.9 | 73.9 | 73.1 | 73.1 |
| DeYO | 33.3 | 74.4 | 24.8 | 73.81 | 30.5 | -0.0001 | 22.51 | 7.7 | 74.7 | 75.3 | 75.2 |
| ZeroSIAM | 32.7 | 75.4 | 24.6 | 74.92 | 30.1 | -0.0001 | 22.57 | 7.1 | 74.3 | 75.3 | 75.2 |
| CMF | 29.5 | 75.3 | 22.2 | 74.62 | 27.2 | -0.0001 | 20.30 | 5.8 | 74.0 | 75.2 | 75.1 |
| SAR | 23.7 | 75.5 | 17.9 | 74.91 | 21.6 | -0.0002 | 16.17 | 4.5 | 73.0 | 75.6 | 75.5 |
| SPA | 18.6 | 74.2 | 13.8 | 75.51 | 17.0 | -0.0002 | 12.83 | 3.8 | 71.9 | 75.7 | 75.5 |

*Table 50.* Discrete utility (%) across utilisation levels ($\rho$), per dataset (ResNet-50).

| Dataset | Method | $\rho$=100% | $\rho$=70% | $\rho$=50% | $\rho$=35% | $\rho$=25% |
|---|---|---|---|---|---|---|
| ImageNet-R | Standard | 36.1 | 36.2 | 36.2 | 36.2 | 36.2 |
| | AdaBN | 37.4 | **39.6** | **39.6** | 39.6 | 39.6 |
| | LAME | 34.8 | 36.1 | 36.1 | 36.1 | 36.1 |
| | NEO | **38.8** | 39.0 | 39.0 | 39.0 | 39.0 |
| | Tent | 17.0 | 24.2 | 34.5 | 42.1 | 42.1 |
| | ETA | 18.1 | 25.0 | 35.8 | **44.5** | 44.5 |
| | SHOT-IM | 14.0 | 19.6 | 27.5 | 38.8 | 41.3 |
| | DeYO | 14.6 | 20.3 | 28.8 | 41.1 | 45.5 |
| | CMF | 11.4 | 16.3 | 23.0 | 32.8 | **46.0** |
| | SAR | 8.6 | 12.0 | 17.4 | 24.7 | 34.8 |
| ImageNet-V2 | Standard | 63.2 | 63.2 | 63.2 | 63.2 | 63.2 |
| | AdaBN | 59.3 | 62.6 | 62.6 | 62.6 | 62.6 |
| | LAME | 61.8 | 63.1 | 63.1 | 63.1 | 63.1 |
| | NEO | **63.2** | **63.2** | **63.2** | **63.2** | **63.2** |
| | Tent | 26.0 | 37.1 | 51.9 | 62.9 | 62.9 |
| | ETA | 26.3 | 36.8 | 52.2 | 63.1 | 63.1 |
| | SHOT-IM | 20.8 | 30.0 | 41.3 | 58.8 | 62.9 |
| | DeYO | 19.2 | 27.2 | 37.7 | 52.6 | 63.2 |
| | CMF | 16.3 | 23.0 | 31.8 | 44.9 | 62.9 |
| | SAR | 13.0 | 18.8 | 26.3 | 36.6 | 51.8 |

*Table 51.* Discrete utility (%) across utilisation levels ($\rho$), per dataset (ViT-B/16).

| Dataset | Method | $\rho$=100% | $\rho$=70% | $\rho$=50% | $\rho$=35% | $\rho$=25% |
|---|---|---|---|---|---|---|
| ImageNet-R | Standard | 59.5 | 59.5 | 59.5 | 59.5 | 59.5 |
| | LAME | 59.3 | 59.3 | 59.3 | 59.3 | 59.3 |
| | NEO | **60.5** | **60.5** | **60.5** | 60.5 | 60.5 |
| | Tent | 28.2 | 40.0 | 57.0 | 62.2 | 62.2 |
| | ETA | 29.5 | 42.0 | 60.3 | **65.6** | 65.6 |
| | SHOT-IM | 25.5 | 33.7 | 46.0 | 52.7 | 52.7 |
| | DeYO | 22.0 | 31.0 | 43.8 | 61.6 | 65.9 |
| | ZeroSIAM | 20.3 | 29.0 | 40.8 | 57.0 | 63.5 |
| | CMF | 16.6 | 24.4 | 35.4 | 50.8 | **66.2** |
| | SAR | 13.9 | 19.5 | 28.3 | 40.3 | 56.9 |
| | SPA | 11.8 | 16.5 | 23.9 | 32.9 | 46.8 |
| ImageNet-V2 | Standard | **75.5** | **75.5** | **75.5** | **75.5** | **75.5** |
| | LAME | 75.3 | 75.3 | 75.3 | 75.3 | 75.3 |
| | NEO | 75.5 | 75.5 | 75.5 | 75.5 | 75.5 |
| | Tent | 35.0 | 49.2 | 69.4 | 75.2 | 75.2 |
| | ETA | 35.0 | 48.7 | 68.7 | 74.6 | 74.6 |
| | SHOT-IM | 30.9 | 43.5 | 61.0 | 72.4 | 72.4 |
| | DeYO | 24.8 | 34.8 | 48.4 | 68.1 | 73.8 |
| | ZeroSIAM | 24.6 | 34.7 | 48.5 | 67.4 | 74.9 |
| | CMF | 22.2 | 31.2 | 43.2 | 61.0 | 74.6 |
| | SAR | 17.9 | 24.8 | 35.1 | 49.1 | 68.8 |
| | SPA | 13.8 | 19.8 | 27.6 | 38.5 | 54.5 |

*Table 52.* Continuous utility (%) across HCI thresholds ($T$), per dataset (ResNet-50).

| Dataset | Method | $T$=50 ms | $T$=100 ms | $T$=200 ms | $T$=400 ms | $T$=1000 ms |
|---|---|---|---|---|---|---|
| ImageNet-R | Standard | 35.67 | 36.08 | 36.13 | 36.15 | 36.16 |
| | AdaBN | 31.77 | 38.04 | **39.01** | **39.35** | 39.52 |
| | LAME | 31.31 | 35.18 | 35.75 | 35.94 | 36.04 |
| | NEO | **38.33** | **38.87** | 38.94 | 38.97 | 38.98 |
| | Tent | 6.38 | 21.60 | 31.01 | 36.31 | 39.70 |
| | ETA | 6.69 | 22.74 | 32.73 | 38.36 | **41.98** |
| | SHOT-IM | 4.66 | 17.69 | 27.51 | 33.80 | 38.16 |
| | DeYO | 4.90 | 18.90 | 29.76 | 36.84 | 41.82 |
| | CMF | 3.74 | 15.71 | 26.66 | 34.77 | 41.03 |
| | SAR | 2.70 | 12.04 | 21.83 | 30.00 | 36.93 |
| ImageNet-V2 | Standard | **62.54** | **63.08** | **63.15** | 63.17 | 63.18 |
| | AdaBN | 50.33 | 60.14 | 61.67 | 62.19 | 62.46 |
| | LAME | 58.24 | 62.18 | 62.72 | 62.90 | 63.00 |
| | NEO | 62.17 | 63.03 | 63.14 | **63.18** | **63.20** |
| | Tent | 9.77 | 32.47 | 46.49 | 54.37 | 59.41 |
| | ETA | 9.70 | 32.37 | 46.46 | 54.41 | 59.51 |
| | SHOT-IM | 7.31 | 27.04 | 41.91 | 51.42 | 58.02 |
| | DeYO | 6.35 | 24.58 | 39.65 | 49.97 | 57.48 |
| | CMF | 5.29 | 21.48 | 36.37 | 47.47 | 56.07 |
| | SAR | 4.19 | 17.86 | 32.15 | 44.08 | 54.20 |

*Table 53.* Continuous utility (%) across HCI thresholds ($T$), per dataset (ViT-B/16).

| Dataset | Method | $T$=200 ms | $T$=400 ms | $T$=1000 ms | $T$=2000 ms |
|---|---|---|---|---|---|
| ImageNet-R | Standard | 59.47 | 59.47 | 59.47 | 59.47 |
| | LAME | 59.30 | 59.30 | 59.30 | 59.30 |
| | NEO | **60.46** | **60.46** | **60.46** | 60.46 |
| | Tent | 27.06 | 43.87 | 54.68 | 58.41 |
| | ETA | 28.51 | 46.22 | 57.61 | **61.54** |
| | SHOT-IM | 20.60 | 35.09 | 45.22 | 48.89 |
| | DeYO | 20.42 | 38.35 | 53.27 | 59.26 |
| | ZeroSIAM | 18.94 | 36.11 | 50.80 | 56.79 |
| | CMF | 17.03 | 34.27 | 50.64 | 57.82 |
| | SAR | 13.24 | 28.34 | 44.61 | 52.44 |
| | SPA | 10.95 | 25.10 | 42.84 | 52.48 |
| ImageNet-V2 | Standard | **75.50** | **75.50** | **75.50** | **75.50** |
| | LAME | 75.34 | 75.34 | 75.34 | 75.34 |
| | NEO | 75.45 | 75.45 | 75.45 | 75.45 |
| | Tent | 32.83 | 53.06 | 66.07 | 70.57 |
| | ETA | 32.52 | 52.59 | 65.53 | 70.01 |
| | SHOT-IM | 28.49 | 48.32 | 62.18 | 67.19 |
| | DeYO | 22.51 | 42.43 | 59.31 | 66.17 |
| | ZeroSIAM | 22.57 | 42.74 | 59.99 | 67.04 |
| | CMF | 20.30 | 39.91 | 57.95 | 65.69 |
| | SAR | 16.17 | 34.27 | 53.80 | 63.18 |
| | SPA | 12.83 | 29.02 | 49.31 | 60.35 |

*Table 54.* Amortised utility (%) across overhead budgets ($B$), per dataset (ResNet-50).

| Dataset | Method | $B$=1 s | $B$=2 s | $B$=4 s | $B$=8 s | $B$=16 s | $B$=32 s |
|---------|--------|---------|---------|---------|---------|----------|----------|
| ImageNet-R | Standard | 36.16 | 36.16 | 36.16 | 36.16 | 36.16 | 36.16 |
| | AdaBN | 39.68 | 39.62 | 39.62 | 39.62 | 39.62 | 39.62 |
| | LAME | 36.10 | 36.10 | 36.10 | 36.10 | 36.10 | 36.10 |
| | NEO | 38.99 | 38.99 | 38.99 | 38.99 | 38.99 | 38.99 |
| | Tent | 1.95 | 3.60 | 32.85 | 42.20 | 42.33 | 42.05 |
| | ETA | 1.97 | 3.49 | 32.63 | 44.82 | 44.86 | 44.50 |
| | SHOT-IM | **40.61** | **40.87** | 41.85 | 42.28 | 41.96 | 41.53 |
| | DeYO | 1.56 | 2.77 | **42.55** | 43.87 | 44.95 | 45.55 |
| | CMF | 1.23 | 2.08 | 32.36 | **45.59** | **46.07** | **46.40** |
| | SAR | 1.07 | 1.56 | 2.66 | 41.51 | 42.25 | 42.55 |
| ImageNet-V2 | Standard | 63.19 | 63.19 | 63.19 | 63.19 | 63.19 | 63.19 |
| | AdaBN | 62.62 | 62.62 | 62.62 | 62.62 | 62.62 | 62.62 |
| | LAME | 63.05 | 63.05 | 63.05 | 63.05 | 63.05 | 63.05 |
| | NEO | 63.21 | 63.21 | 63.21 | 63.21 | 63.21 | 63.21 |
| | Tent | 7.38 | 14.35 | 57.73 | 63.02 | 62.92 | 62.92 |
| | ETA | 7.40 | 13.98 | 56.73 | 63.15 | 63.06 | 63.06 |
| | SHOT-IM | **63.48** | **63.36** | **63.28** | 63.15 | 62.87 | 62.87 |
| | DeYO | 4.53 | 8.51 | 61.37 | 62.73 | **63.18** | 63.18 |
| | CMF | 3.73 | 7.01 | 50.72 | **63.26** | 63.09 | **62.93** |
| | SAR | 2.88 | 5.27 | 10.42 | 63.06 | 63.00 | 62.88 |

*Table 55.* Amortised utility (%) across overhead budgets ($B$), per dataset (ViT-B/16).

| Dataset | Method | $B$=2.5 s | $B$=5 s | $B$=10 s | $B$=20 s | $B$=40 s | $B$=80 s |
|---------|--------|-----------|---------|----------|----------|----------|----------|
| ImageNet-R | Standard | 59.47 | 59.47 | 59.47 | 59.47 | 59.47 | 59.47 |
| | LAME | 59.30 | 59.30 | 59.30 | 59.30 | 59.30 | 59.30 |
| | NEO | 60.46 | 60.46 | 60.46 | 60.46 | 60.46 | 60.46 |
| | Tent | 59.78 | 60.11 | 60.62 | 61.34 | 62.06 | 62.22 |
| | ETA | 60.21 | 61.11 | 62.72 | **65.14** | **65.74** | 65.55 |
| | SHOT-IM | **64.86** | **64.46** | **63.05** | 58.67 | 54.78 | 52.71 |
| | DeYO | 60.23 | 61.25 | 62.82 | 64.88 | 65.72 | 65.85 |
| | ZeroSIAM | 60.12 | 60.92 | 62.15 | 63.31 | 64.24 | 63.66 |
| | CMF | 59.82 | 60.32 | 61.26 | 62.84 | 64.98 | **66.01** |
| | SAR | 59.62 | 59.69 | 60.00 | 60.37 | 61.00 | 61.77 |
| | SPA | 60.05 | 60.85 | 61.73 | 63.32 | 64.81 | 65.19 |
| ImageNet-V2 | Standard | 75.50 | **75.50** | **75.50** | **75.50** | **75.50** | 75.50 |
| | LAME | 75.34 | 75.34 | 75.34 | 75.34 | 75.34 | 75.34 |
| | NEO | 75.45 | 75.45 | 75.45 | 75.45 | 75.45 | 75.45 |
| | Tent | 75.32 | 75.31 | 75.16 | 75.15 | 75.32 | 75.32 |
| | ETA | 74.91 | 74.79 | 74.66 | 74.57 | 74.91 | 74.91 |
| | SHOT-IM | 73.15 | 72.68 | 72.74 | 72.43 | 72.42 | 73.15 |
| | DeYO | 75.22 | 75.01 | 74.60 | 73.92 | 73.81 | 75.22 |
| | ZeroSIAM | 75.23 | 74.94 | 75.00 | 74.89 | 74.92 | 75.23 |
| | CMF | 75.14 | 74.83 | 74.76 | 74.68 | 74.62 | 74.62 |
| | SAR | 75.45 | 75.40 | 75.26 | 75.11 | 74.94 | 74.91 |
| | SPA | **75.51** | 75.37 | 75.20 | 75.38 | 75.45 | **75.51** |

## C.8. Additional Hardware (Raspberry Pi 5)

We evaluate ResNet-18 on CIFAR-10-C (Hendrycks & Dietterich, 2019) using the discrete and amortised protocols but now compare results across two hardware platforms: a Raspberry Pi 5 16 GB (CPU, $\lambda = 486.6$ ms) and an Nvidia RTX 4080 (GPU, $\lambda = 1.5$ ms). The continuous protocol does not enable a fair comparison, as the standard inference latencies of both platforms sit at opposite extremes; there is no common ground to pick a reasonable HCI threshold. Table 56 reports per-method adaptation overhead and offline accuracy. Offline accuracy is hardware-invariant, but adaptation time varies substantially: gradient-based methods, such as Tent and SAR, are around 2.7–6.1× the inference budget on CPU, compared to roughly 1–3× on GPU; this is expected due to both hardware limitations and less optimised runtime kernels on CPU.

Table 57 decomposes discrete utility at full utilisation ($\rho = 100\%$). On CPU, the comparatively high overhead of gradient-based methods collapses their availability $\alpha$ below 31%. AdaBN, which requires no gradient computation, dominates on CPU at this utilisation level. Table 58 sweeps discrete utility $U_d$ across utilisation levels $\rho$. The pattern is consistent across hardware: at lower utilisation, gradient-based methods recover because the system has slack to absorb their cost, and Tent eventually reaches its offline accuracy ceiling. Table 59 decomposes amortised utility at a budget of $B \approx 25\lambda$. We see the same pattern here as with ResNet-50, owing to how gradient-based methods interact with the running statistics of batch normalisation layers. Table 60 sweeps amortised utility across budget levels.

The ordinal ranking of methods by adaptation cost (Table 56) is stable across both platforms, with only Tent and ETA swapping within their tier due to measurement noise. Across both hardware platforms and all 15 corruptions, with 5 discrete and 4 amortised scenarios, this group encompasses a total of 270 distinct temporal evaluations (*i.e.*, $2 \times 15 \times 5 + 2 \times 15 \times 4$). Winners change across hardware as gradient-based methods are penalised more heavily on CPU; conclusions about method choice should be drawn in the context of the intended deployment hardware.

*Table 56.* Per-method overhead and offline accuracy for ResNet-18 on CIFAR-10-C. $\bar{\ell}$ is the mean adaptation time per batch (ms). $\bar{\ell}/\lambda$ expresses overhead in units of inference time; this governs availability $\alpha$ in discrete and adapt fraction $\beta$ in amortised evaluation. Offline accuracy is the mean over 15 corruptions (hardware-invariant).

| | | CPU | | GPU | |
|---|---|---|---|---|---|
| Method | Offline (%) | $\bar{\ell}$ (ms) | $\bar{\ell}/\lambda$ | $\bar{\ell}$ (ms) | $\bar{\ell}/\lambda$ |
| Standard | 61.3 | 0.0 | 0.00 | 0.0 | 0.01 |
| AdaBN | 77.2 | 0.0 | 0.00 | 0.0 | 0.01 |
| LAME | 59.9 | 0.1 | 0.00 | 0.0 | 0.02 |
| NEO | 64.2 | 0.1 | 0.00 | 0.0 | 0.02 |
| Tent | **79.7** | 1330.9 | 2.74 | 1.5 | 0.99 |
| ETA | 79.7 | 1328.6 | 2.73 | 1.9 | 1.26 |
| SHOT-IM | 65.4 | 2916.1 | 5.99 | 3.2 | 2.16 |
| SAR | 78.2 | 2975.6 | 6.12 | 5.1 | 3.37 |

*Table 57.* Discrete utility decomposition ($\rho = 100\%$) on CIFAR-10-C (ResNet-18).

| | CPU ($\lambda = 486.6$ ms) | | | GPU ($\lambda = 1.5$ ms) | | |
|---|---|---|---|---|---|---|
| Method | $\alpha$ (%) | $\bar{a}_s$ (%) | $U_d$ (%) | $\alpha$ (%) | $\bar{a}_s$ (%) | $U_d$ (%) |
| Standard | 100.0 | 61.3 | 61.3 | 100.0 | 61.3 | 61.3 |
| AdaBN | 100.0 | 77.2 | **77.2** | 100.0 | 77.2 | **77.2** |
| LAME | 100.0 | 59.9 | 59.9 | 74.4 | 61.8 | 46.0 |
| NEO | 100.0 | 64.2 | 64.2 | 100.0 | 64.2 | 64.2 |
| Tent | 30.4 | **78.9** | 24.0 | 52.4 | 78.8 | 41.3 |
| ETA | 30.6 | 78.7 | 24.1 | 46.4 | **79.1** | 36.7 |
| SHOT-IM | 15.9 | 75.1 | 12.0 | 33.7 | 73.1 | 24.6 |
| SAR | 15.4 | 78.0 | 12.0 | 25.0 | 78.0 | 19.5 |

*Table 58.* Discrete utility $U_d$ (%) across utilisation levels $\rho$ on CIFAR-10-C (ResNet-18).

| Method | $\rho = 100\%$ CPU | GPU | $\rho = 70\%$ CPU | GPU | $\rho = 50\%$ CPU | GPU | $\rho = 35\%$ CPU | GPU | $\rho = 25\%$ CPU | GPU |
|---|---|---|---|---|---|---|---|---|---|---|
| Standard | 61.32 | 61.32 | 61.32 | 61.32 | 61.32 | 61.32 | 61.32 | 61.32 | 61.32 | 61.32 |
| AdaBN | **77.19** | **77.19** | **77.19** | **77.19** | **77.19** | 77.19 | **77.19** | 77.19 | 77.19 | 77.19 |
| LAME | 59.88 | 45.98 | 59.88 | 58.83 | 59.88 | 59.84 | 59.88 | 59.88 | 59.88 | 59.88 |
| NEO | 64.21 | 64.21 | 64.21 | 64.21 | 64.21 | 64.21 | 64.21 | 64.21 | 64.21 | 64.21 |
| Tent | 23.97 | 41.31 | 34.00 | 61.49 | 47.44 | **79.67** | 67.85 | 79.67 | **79.71** | 79.67 |
| ETA | 24.05 | 36.73 | 33.97 | 52.25 | 47.36 | 78.12 | 66.89 | **79.80** | 79.67 | **79.80** |
| SHOT-IM | 11.96 | 24.62 | 16.59 | 33.74 | 22.31 | 46.28 | 30.59 | 62.48 | 41.69 | 64.27 |
| SAR | 12.03 | 19.50 | 16.93 | 27.00 | 24.02 | 38.54 | 33.49 | 53.02 | 47.00 | 72.77 |

*Table 59.* Amortised utility decomposition ($B \approx 25\lambda$) on CIFAR-10-C (ResNet-18).

| Method | **CPU** $(B = 12\,\text{s} \approx 25\lambda)$ $\beta$ (%) | $\bar{a}_a$ (%) | $\bar{a}_f$ (%) | $U_a$ (%) | **GPU** $(B = 37\,\text{ms} \approx 25\lambda)$ $\beta$ (%) | $\bar{a}_a$ (%) | $\bar{a}_f$ (%) | $U_a$ (%) |
|---|---|---|---|---|---|---|---|---|
| Standard | 100.0 | 61.3 | – | 61.3 | 100.0 | 61.3 | – | 61.3 |
| AdaBN | 100.0 | 77.2 | – | **77.2** | 100.0 | 77.2 | – | **77.2** |
| LAME | 100.0 | 59.9 | – | 59.9 | 100.0 | 60.8 | – | 60.8 |
| NEO | 100.0 | 64.2 | – | 64.2 | 100.0 | 64.2 | – | 64.2 |
| Tent | 6.4 | 78.3 | 10.6 | 15.0 | 15.8 | 78.5 | 13.2 | 23.5 |
| ETA | 6.3 | 78.4 | 10.6 | 14.9 | 12.6 | 78.0 | 11.6 | 20.0 |
| SHOT-IM | 3.2 | **79.7** | **70.2** | 70.5 | 7.5 | 77.4 | **73.8** | 74.1 |
| SAR | 3.2 | 79.7 | 10.1 | 12.3 | 5.3 | **78.7** | 10.3 | 14.0 |

*Table 60.* Amortised utility $U_a$ (%) across budget levels on CIFAR-10-C (ResNet-18). Budgets are expressed relative to inference time $\lambda$; CPU: $\{12, 24, 48, 96\}$ s; GPU: $\{37, 74, 148, 296\}$ ms.

| Method | $B \approx 25\lambda$ CPU | GPU | $B \approx 50\lambda$ CPU | GPU | $B \approx 100\lambda$ CPU | GPU | $B \approx 200\lambda$ CPU | GPU |
|---|---|---|---|---|---|---|---|---|
| Standard | 61.32 | 61.32 | 61.32 | 61.32 | 61.32 | 61.32 | 61.32 | 61.32 |
| AdaBN | **77.19** | **77.19** | **77.19** | **77.19** | **77.19** | 77.19 | **77.19** | 77.19 |
| LAME | 59.88 | 60.79 | 59.88 | 60.33 | 59.88 | 59.91 | 59.88 | 59.88 |
| NEO | 64.21 | 64.21 | 64.21 | 64.21 | 64.21 | 64.21 | 64.21 | 64.21 |
| Tent | 14.98 | 23.47 | 19.92 | 37.53 | 29.48 | **79.80** | 48.69 | 79.67 |
| ETA | 14.86 | 19.97 | 19.46 | 30.55 | 28.65 | 51.58 | 47.42 | **79.80** |
| SHOT-IM | 70.54 | 74.09 | 72.97 | 72.62 | 74.10 | 69.25 | 71.24 | 66.21 |
| SAR | 12.34 | 13.96 | 14.19 | 17.68 | 18.04 | 24.33 | 25.72 | 43.71 |

