# OpenReview forum: "Tempora: Characterising the Time-Contingent Utility of Online Test-Time Adaptation"
_ICML.cc/2026/Conference — ICML 2026 regular_

### Official Review · Reviewer_VMv7 · 2026-02-27

**Soundness:** 3
**Presentation:** 2
**Significance:** 2
**Originality:** 3
**Overall Recommendation:** 4
**Confidence:** 3

**Summary:**

This paper introduces Tempora, a framework for evaluating test-time adaptation (TTA) methods under temporal constraints. The authors argue that conventional TTA evaluations unrealistically assume unbounded processing time, ignoring the accuracy-latency trade-off critical in real deployments. Tempora comprises three time-contingent utility metrics: discrete utility for hard-deadline asynchronous streams, continuous utility for interactive settings with latency-penalised value decay, and amortised utility for budget-constrained deployments.

**Compliance With Llm Reviewing Policy:**

Affirmed.

**Final Justification:**

No concerns, keeping the recommendation as weak accept.

**Key Questions For Authors:**

See weaknesses. The response may influence the final rating.

**Limitations:**

yes

**Strengths And Weaknesses:**

Strenghths:
- Points out the unreasonableness of the "unbounded processing time" assumption in traditional TTA evaluations, explicitly clarifying the trade-off between accuracy and latency.

- Instead of a single-dimensional penalty, the Tempora framework systematically proposes three complementary utility metrics: discrete, continuous, and amortised.

- Evaluated seven mainstream TTA methods on ImageNet-C, covering 240 evaluations across 16 temporal scenarios.


Weakness:
- Calling ETA a state-of-the-art method in 2026 is unconvincing. The authors should include more recent TTA methods, such as DeYO [1] or PASLE [2], as baselines.
- All evaluations are restricted to a single architecture (ResNet-50), a single dataset (ImageNet-C), a single severity level (5), and a single batch size (64). Although the abstract claims that Tempora is a general framework for "diverse temporal constraints," the impact of varying architectures, datasets, and batch sizes remains unclear.
- In Section 2, $\mathcal{P}_{\mathcal{S}}(X)$ is used to denote the probability distribution, whereas Equation 1 and its surrounding context use $\mathcal{P}$ to represent the set of served batch indices. These symbols should be disambiguated to avoid mathematical confusion.
- LAME is a method specifically designed for non-stationary data streams. Does limiting the evaluation strictly to independent and identically distributed (i.i.d.) scenarios in this paper result in an unfair comparison for this method?
- The paper solely considers image classification tasks. Would the Tempora framework remain effective and applicable for dense prediction tasks, such as semantic segmentation, where the baseline latency $\lambda$ is significantly larger?

[1] Entropy is not enough for test-time adaptation: From the perspective of disentangled factors. ICLR 2024.

[2] SELECTIVE  LABEL  ENHANCEMENT  LEARNING  FOR TEST-TIME ADAPTATION. ICLR 2025.

---

> ### Author Rebuttal · Authors · 2026-03-31
>
> **W1: Adding recent methods.**
>
> We will update our framing to remove that mention, but it remains the dominant method under our temporal evaluation with ResNet-50 and ImageNet-c. Please see our responses to reviewer QNYa-W3 for full details, including how newer methods fare on ViT. We implemented PASLE but note that the authors do not evaluate on ImageNet-C; as such, we do not have suitable hyperparameters to conduct a meaningful evaluation.
>
> **W2: Dataset and architecture diversity.**
>
> Below, we show that rank instability persists across both new datasets and architectures. On ImageNet-R (Tab. 1), winners shift across all three archetypes. On ImageNet-V2 with ViT-B/16 (Tab. 2), the non-adaptive baseline dominates under temporal pressure. Amortised budgets are shown as ResNet-50/ViT-B (ms).
>
> Table 1: Winners per scenario on ImageNet-R.
> |Scenario|RN-50|ViT|
> |---|---|---|
> |Offline|CMF|CMF|
> |Discrete (100%)|NEO|NEO|
> |Discrete (70%)|AdaBN|NEO|
> |Discrete (50%)|AdaBN|NEO|
> |Discrete (35%)|ETA|ETA|
> |Discrete (25%)|CMF|CMF|
> |Continuous (50ms)|NEO|--|
> |Continuous (100ms)|NEO|--|
> |Continuous (200ms)|AdaBN|NEO|
> |Continuous (400ms)|AdaBN|NEO|
> |Continuous (1000ms)|ETA|NEO|
> |Continuous (2000ms)|--|ETA|
> |Amortised (1k/2.5k)|SHOT|SHOT|
> |Amortised (2k/5k)|SHOT|SHOT|
> |Amortised (4k/10k)|DeYO|SHOT|
> |Amortised (8k/20k)|CMF|ETA|
> |Amortised (16k/40k)|CMF|ETA|
>
> Table 2: Winners per scenario on ImageNet-V2.
> |Scenario|RN-50|ViT|
> |---|---|---|
> |Offline|NEO|SPA|
> |Discrete (100%)|NEO|Basic|
> |Discrete (70%)|NEO|Basic|
> |Discrete (50%)|NEO|Basic|
> |Discrete (35%)|NEO|Basic|
> |Discrete (25%)|NEO|Basic|
> |Continuous (50ms)|Basic|--|
> |Continuous (100ms)|Basic|--|
> |Continuous (200ms)|Basic|Basic|
> |Continuous (400ms)|NEO|Basic|
> |Continuous (1000ms)|NEO|Basic|
> |Continuous (2000ms)|--|Basic|
> |Amortised (1k/2.5k)|SHOT|SPA|
> |Amortised (2k/5k)|SHOT|SAR|
> |Amortised (4k/10k)|SHOT|SAR|
>
> Please see our response to reviewer QNYa-W4 for full details.
>
> **W3: Overloaded symbols.**
>
> We will fix this in our revised manuscript.
>
> **W4: Evaluation setup for LAME**
>
> We do not view the evaluation as unfair. If a practitioner has this setup, then it simply reveals that the method is unsuitable, which is the intention. There is no fair comparison in TTA literature, which is why we focus on interpretable factors that contribute to the results; a practitioner can determine what trade-offs they find acceptable. We acknowledge that extensive evaluation in tandem with distribution realism benchmarks may further reveal each method's favourable operating conditions.
>
> **W5: Extension to other tasks.**
>
> Yes, indeed it would. Specific thresholds, particularly for continuous and amortised evaluations, would likely need to be adjusted; we detail how practitioners might do this in a response to reviewer QNYa-Q4. Even within image classification, the baseline latency can be made larger by (1) picking a larger model, (2) using a dataset with a larger image size, (3) increasing the batch size, or (4) slowing down the hardware. Each evaluation setup may produce unique trade-offs and method rankings that are most relevant to practitioners or method designers in that setting; it is by design that the rankings do not hold outside it.

---

> > ### Author Rebuttal · Reviewer_VMv7 · 2026-04-02
> >
> > Thank you for addressing my comments. I have no further questions and I will keep my score.

---

> > > ### Author Response · Authors · 2026-04-08
> > >
> > > We are glad that the reviewer’s concerns were resolved. We invite them to read our detailed reply to reviewer d95w, which clarifies the conceptual and empirical distance from prior work across protocol design, evaluative purpose, and methodological correctness, supported by new empirical evidence in Table 1. Together with the novel continuous and amortised protocols, the decomposition framing, and the extensive additional experiments provided in this rebuttal, we believe our work makes a meaningful contribution with practical significance for the wider community and would appreciate if the reviewer considers updating their score in light of the full rebuttal.

---

### Official Review · Reviewer_QNYa · 2026-02-28

**Soundness:** 2
**Presentation:** 4
**Significance:** 3
**Originality:** 2
**Overall Recommendation:** 3
**Confidence:** 4

**Summary:**

This paper investigates an important problem in the test-time adaptation (TTA) community: many existing TTA methods focus primarily on the total adaptation time, while overlooking real-time effectiveness under explicit computation budgets. Building on this observation, the authors develop Tempora, an evaluation framework that quantifies the accuracy–latency trade-off using three metrics. Under Tempora, the authors report three findings using ResNet-50 on ImageNet-C: Rank instability, corruption-specific trade-offs, and pressure-specific trade-offs. These observations highlight the importance of selecting appropriate adaptation strategies under different budget constraints.

**Compliance With Llm Reviewing Policy:**

Affirmed.

**Final Justification:**

After reading the rebuttal, I appreciate the added clarifications and experiments. The authors provided substantial additional validations during the rebuttal, many of which were missing from the original manuscript, and these results offer stronger support for the paper’s conclusions. However, like Reviewer d95w, I still have concerns about the novelty relative to Alfarra et al. (2024). Taking the full discussion into account, I have decided to increase my score to 3, rather than 4.

**Key Questions For Authors:**

1. Do different TTA families show systematically different trade-offs? (The taxonomy in the Adaptation Overhead subsection could serve as the primary grouping; a category-wise analysis would be very helpful.)

2. Are the main conclusions robust when including a broader and more representative set of TTA methods? (please consider adding more established and recent baselines).

3. Do the reported findings transfer across architectures, datasets, and task types? [OPTIMAL]

4. How should practitioners map real deployment constraints to Tempora’s “budget/pressure” settings, and how should the three metrics be interpreted in practice?

**Limitations:**

The paper does not include an explicit Limitations section. I believe the authors should at minimum discuss the limitations regarding task type (image classification) and model architecture (ResNet-50), so that readers can clearly understand the scope under which the conclusions are expected to hold and avoid over-generalizing the findings.

**Strengths And Weaknesses:**

Strengths

1. The paper is well written and easy to follow.
2. Plotting and visualization are clear.
3. The work tackles an important research subject, namely the accuracy–latency trade-off of TTA methods. Choosing an appropriate adaptation mode for a specific downstream task is practically important and aligned with real deployment settings.

Weakness

1. The motivation and contributions appear quite close to [1]. In particular, rank instability was already discussed in that work.
[1] Evaluation of test-time adaptation under computational time constraints, ICML 2024

2. I suggest the authors take inspiration from [1] and further analyze whether different types of methods exhibit clearly different behaviors under the proposed evaluation. For instance, the Adaptation Overhead section already provides a categorization; it would be helpful to explicitly compare performance across these categories under budget constraints.

3. The set of compared TTA methods seems too limited. Given that TTA has been studied for many years, I would be interested to see whether newer methods—along with the evolution of the field—actually yield more consistent advantages under stricter and more realistic deployment conditions.

4. The experiments rely on a single model (ResNet-50), a single dataset (ImageNet-C), and a single task type (classification). This substantially limits the generality of the conclusions.

---

> ### Author Rebuttal · Authors · 2026-03-31
>
> **W1: Novelty relative to prior work**
>
> Please see our response to reviewer d95w-W2 for full details.
>
> **W2+Q1: Category-wise analysis**
>
> We thank the reviewer for this suggestion, as it surfaced a subtle framing error in our background section. We categorise methods by the component of the inference pipeline they modify to provide an accessible overview of how methods adapt. The taxonomy naming and phrase *"informs their computational overhead"* imply that adaptation location determines cost, which is not the case. We will remove this phrase and adjust the framing accordingly.
>
> Each method's overhead on either side of the prediction boundary determines its trade-offs. We do observe qualitatively distinct failure modes between gradient-based and gradient-free methods; for discrete, the former faces availability collapse at tight thresholds, while the latter faces accuracy stagnation at relaxed thresholds. However, this observation is specific to our method set and does not generalise; diffusion-driven adaptation, a gradient-free method, would suffer availability collapse. We will include this distinction in our results discussion.
>
> **W3+Q2: Adding recent methods**
>
> We extended our evaluation to ViT-B/16 on ImageNet-{C,R,V2} with four recent methods: SPA (ICML'25), CMF (ICLR'24), DeYO (ICLR'24) and ZeroSIAM (ICLR'26). Crucially, rank instability persists across model, dataset, and method choices. On ViT (Tab. 1), NEO dominates (33.8% win rate) despite never winning offline. On ResNet-50 (Tab. 2), introducing CMF and DeYO displaces ETA as the offline winner and further increases instability relative to our original results. These findings confirm that recent methods suffer similar failure modes and do not meet our three properties for deployable adaptation. We will include comprehensive results (omitted here for brevity) in our revised manuscript to demonstrate the generality of our conclusions.
>
> Table 1: Winner summary for ViT-B on ImageNet-C across 240 evaluations (15 offline, 75 discrete, 60 continuous and 90 amortised).
> |Method|Total|Offline|Discrete|Continuous|Amortised|
> |---|---|---|---|---|---|
> |NEO|81|0|34|40|7|
> |ETA|67|0|27|18|22|
> |SPA|56|13|0|0|43|
> |CMF|17|2|12|0|3|
> |SHOT-IM|14|0|0|0|14|
> |Tent|4|0|2|2|0|
> |ZeroSIAM|1|0|0|0|1|
>
> Table 2: Winner delta for ResNet-50 on ImageNet-C with DeYO and CMF.
> |Method|Offline (/15)|Discrete (/75)|Amortised (/90)|Total (/255)|
> |---|---|---|---|---|
> |ETA|15 → 0|51 → 36|46 → 23|156 → 103|
> |SHOT-IM|-|-|35 → 26|35 → 26|
> |CMF|10|7|22|39|
> |DeYO|5|8|10|23|
>
> Abridged results on ImageNet-{R,V2} are documented in our response to reviewer VMv7-W2.
>
> **W4+Q3: Generality of findings**
>
> While our extended results demonstrate empirical transfer across models and datasets, we emphasise that the generality of our three design properties is structural. They are necessary conditions derived from the failure modes our decompositions expose:
>
> 1. Corruption-conditioned compute allocation. If methods had it, overhead would scale with difficulty; the overhead decomposition would show this.
> 2. Time-aware scaling. If methods had it, availability and responsiveness would not collapse under tight deadlines; methods with negligible overhead satisfy it vacuously.
> 3. Anytime performance. If methods had it, frozen accuracy after budget exhaustion would not collapse below the non-adaptive baseline.
>
> A different evaluation setup may alter the severity of these failure modes for specific methods, but the underlying trade-off remains. Therefore, these design requirements remain universally applicable wherever a model must adapt under time constraints. We will clarify this structural argument in our method design discussion.
>
> **Q4: Practical guidance**
>
> We will add a dedicated subsection in our post-results discussion to detail how practitioners can configure Tempora to their use-cases. Specifically, we will clarify the mapping of physical constraints:
>
> 1. Discrete: system utilisation $\rho=\lambda/\gamma$ is derived using their model's baseline latency $\lambda$ and their stream's inter-arrival time $\gamma$.
> 2. Continuous: threshold $T$ is anchored to their application's user responsiveness expectations.
> 3. Amortised: budget $B$ defines the cumulative wall-clock overhead that their deployment can absorb.
>
> Practitioners can then use decomposed metrics as diagnostic tools to identify specific failure modes or acceptable trade-offs.
>
>
> **Limitations**
>
> Our post-results discussion (section 5) implicitly addresses the evaluation scope. We will consolidate this material under an explicit limitations section and include the crucial point that *specific method rankings* should not be generalised beyond the considered setting.
>
> ---
>
> Our rebuttal provides extensive further coverage of new models (ViT-B, RN-18), datasets (IN-R, IN-V2, C-10), methods (SPA, CMF, DeYO, ZeroSIAM), hardware (RPI 5), and seeds. We believe this substantially strengthens our findings to be generalisable across diverse settings of TTA.

---

> > ### Author Rebuttal · Reviewer_QNYa · 2026-04-04
> >
> > Thanks to the authors for their response. The rebuttal strengthens the submission. My concerns about empirical coverage and generality are alleviated, while my remaining concern is mainly about how clearly the paper can discuss its novelty relative to the prior work in the revised version.
> >
> > I have decided to raise my score.

---

> > > ### Author Response · Authors · 2026-04-08
> > >
> > > We thank the reviewer for raising their score. We clarify the novelty concern in our rebuttal reply to reviewer d95w and will make the differences relative to Alfarra et al. clear in our revised manuscript.

---

### Official Review · Reviewer_d95w · 2026-03-11

**Soundness:** 3
**Presentation:** 4
**Significance:** 3
**Originality:** 2
**Overall Recommendation:** 4
**Confidence:** 3

**Summary:**

The idea behind this paper is to define utility metrics that jointly score a TTA method on both prediction accuracy and how quickly it delivers those predictions. Three metrics cover different deployment pressures (hard deadlines where missed batches are lost, interactive settings where late predictions lose value gradually, and fixed compute budgets where the model freezes once overhead runs out). What makes this more than just "add a time penalty" is that each metric splits into factors that isolate specific failure modes. For example, discrete utility separates availability from served accuracy, which reveals that ETA serves only 41% of batches at full pipeline utilization despite having the best per-batch accuracy. The amortised metric exposes a different failure: Tent, ETA, and SAR collapse to 0.1% accuracy after their budget runs out, while SHOT-IM holds steady at 32%. Across 240 temporal evaluations on ImageNet-C, the offline ranking breaks down and no method consistently wins, which is the central empirical message.

**Compliance With Llm Reviewing Policy:**

Affirmed.

**Final Justification:**

The rebuttal resolved the main empirical concern (single architecture/dataset) with ViT-B/16, ResNet-18, new datasets, new methods, and cross-hardware results. The SHOT-IM explanation is now mechanistic and satisfying. The novelty relative to Alfarra et al. (2024) is better articulated after the follow-up and the buffer correction, physical-unit grounding, and decomposition framework are reasonable contributions but the conceptual distance from prior work remains modest, and the continuous/amortised metrics use standard formulations. Originality is the limiting factor. Score unchanged at 4 (Weak Accept).

**Key Questions For Authors:**

1. **Generalization beyond ResNet-50:** Results on a different architecture (e.g., a vision transformer, which lacks batch norm in the backbone) would directly address the scope concern. If Tempora's findings hold, or if new patterns emerge, it would strengthen the case that this framework captures fundamental properties of TTA methods rather than ResNet-50-specific behavior. This is the main thing that would improve the assessment.

2. **SHOT-IM frozen accuracy:** This is the most counterintuitive result in the paper and could inform how the community designs TTA methods going forward. The speculation about the information maximisation objective warrants deeper investigation. Is it because SHOT-IM updates the full backbone vs only BN parameters? Does the IM loss preserve feature space structure under budget pressure? A more thorough analysis here would turn an interesting observation into a concrete contribution to TTA method design.

3. **Hardware sensitivity:** All overhead measurements are on an RTX 4080. On weaker hardware like edge GPUs or CPUs, absolute latencies change and relative overheads could shift. Demonstrating stability of conclusions across hardware would strengthen the practical relevance of the framework, particularly given the deployment scenarios motivating each metric.

4. **Non-episodic evaluation:** LAME underperforms in all 240 cases because the episodic i.i.d. setup mismatches its design. This means the evaluation protocol itself affects rankings before temporal pressure comes in. Addressing whether Tempora should be combined with distributional realism benchmarks (e.g., BoTTA, UniTTA) would help situate the framework relative to other ongoing evaluation efforts in TTA.

**Limitations:**

Yes.

**Strengths And Weaknesses:**

### Strengths

1. **The problem matters and the framing is effective.** Based on the cited literature, TTA methods are not typically evaluated with latency in the loop. Alfarra et al. (2024) started looking at this with their discrete temporal evaluation but Tempora extends it to continuous and budget-constrained scenarios. Grounding each metric in a concrete deployment (surveillance systems, photo organisers, drones) makes the framework feel less abstract, which helps.

2. **The utility decompositions are the most valuable contribution for the community.** What makes this paper more than just "measuring utility under time pressure" is the decomposition of each metric into interpretable factors. Discrete utility splits into availability and served accuracy, continuous separates accuracy and responsiveness and their covariance, amortised separates adapted and frozen phase accuracy. These decompositions tell practitioners *why* rankings change, not just that they do. The computational insolvency concept is useful too. Table 3 showing ETA at 41% availability despite highest served accuracy is a good example of why these breakdowns are needed.

3. **Thorough evaluation, good writing.** 240 temporal evaluations across 16 scenarios, 7 methods, 15 corruptions with detailed decomposition tables. Figure 3 and the sweep tables are informative. Writing quality is notably high. The notation and terminology tables in the appendix (Tables 6, 7) are helpful for navigating the formalism.

### Weaknesses

1. **Only ResNet-50 on ImageNet-C, which is thin for a framework paper.**  Everything is one architecture, one dataset, severity 5. ViTs have very different latency profiles (and lack batch normalisation layers entirely, which would fundamentally change the discrete availability and amortised frozen-accuracy dynamics), mobile architectures have different overhead ratios, and the specific findings here (ETA loses 41.2%, AdaBN dominates at high pressure, SHOT-IM retains frozen accuracy) could all change on a different backbone. For a methods paper this scope would be fine but a framework paper that aims to change how the community evaluates TTA needs broader validation. The discussion acknowledges this but doesnt do anything about it. Atleast one additional architecture would help significantly.

2. **The novelty relative to Alfarra et al. (2024) needs stronger articulation.** Alfarra et al. already showed that temporal constraints affect TTA rankings and introduced a formal evaluation protocol. The new elements here are continuous and amortised metrics (hyperbolic discounting from Mazur (1987) and a budget cutoff, both standard formulations in their respective fields), a buffered discrete protocol, and the utility decompositions. The decompositions are new and valuable, but the other additions are covering more temporal scenarios with known tools. The submission should more clearly articulate what conceptual advance is offered beyond Alfarra et al., because as it stands the line between extension and new contribution is blurry.

3. **The core experimental findings have limited novelty beyond basic latency analysis.** Gradient-based methods are 2.4-4.9x slower (Table 2). Under time pressure, slower methods lose. This follows directly from the overhead decomposition without running 240 experiments. The more novel and interesting findings, like SHOT-IM keeping its frozen accuracy while Tent/ETA/SAR collapse to 0.1%, or the corruption-specific Spearman correlations, dont get enough space. In particular, the corruption-specific correlations (rs = -0.74 for brightness vs. rs = 0.21 for Gaussian noise under the same temporal pressure) reveal something about the interaction between corruption structure and adaptation value that cannot be predicted from overhead alone, and this deserves more attention. The SHOT-IM result could be a real contribution to TTA method design: the paper speculates its related to the information maximisation objective but a deeper investigation (e.g., is it because SHOT-IM updates the full backbone vs only BN parameters? does the IM loss preserve feature space structure under budget pressure?) would make this finding much more impactful.

### Minor Weaknesses

- **(Originality)** The buffered discrete protocol is a sensible practical improvement over Alfarra et al. (2024) but not a big conceptual step.
- **(Significance)** There is a real need for this kind of evaluation in TTA, but Alfarra et al. (2024) already established the basic point that latency matters. The incremental nature of the advance limits how much new ground this covers.

---

> ### Author Rebuttal · Authors · 2026-03-31
>
> **W1+Q1: Generalisation beyond RN-50**
>
> Please see our responses to reviewer QNYa-W3+W4 for full details.
>
> **W2+W4+W5: Novelty relative to prior work**
>
> By listing *"Rank Instability"* first in our contributions section, we inadvertently implied that its *existence* is our primary claim. We will revise this and clarify that Tempora advances prior work in two ways:
>
> 1. *Deployment realism and extended scope beyond discrete evaluation*. Prior works express the speed of a data stream relative to a model-specific proxy; Alfarra et al. do so relative to the source model's inference latency, and Ghunaim et al. (from CgEt-W1) do so relative to the baseline's FLOPs. The ratios make the hidden temporal constraint elastic, stretching to accommodate the algorithm (e.g., tuning a stream's speed to match a method's FLOPs) rather than reflecting a fixed deployment reality (e.g., a camera's frame rate). This proxy makes it difficult to model diverse operational conditions: if the "unit" of evaluation is the model's own speed, one cannot easily simulate exogenous factors (e.g., a fixed 500 ms total overhead or user's 2s patience thresholds) in a way that is consistent across models and methods. Tempora grounds evaluation in physical units (ms). In doing so, we revise and recontextualise Alfarra et al.'s discrete protocol while also extending evaluation to the continuous and amortised archetypes. We show that rank instability *persists* across a wider and more physically interpretable taxonomy of deployment constraints.
>
> 2. *Diagnosis*. Prior works observe *whether* rankings change. Tempora introduces a decomposition framework to diagnose *how* and *why* they change. This allows us to isolate specific failure modes and identify properties logically necessary for deployable adaptation. We expand on this in our response to reviewer QNYa-W4+Q3.
>
> Our discrete protocol differs from Alfarra et al. across three dimensions.
>
> 1. *Conceptual alignment*: we fix inter-arrival time as a static deployment parameter and introduce a single-batch buffer to keep the pipeline saturated.
> 2. *Evaluative purpose*: we sweep $\rho \in$ {25\%, 35\%, 50\%, 70\%, 100\%\} rather than a single operating point, and we provide an interpretable decomposition to identify failure modes.
> 3. *Methodological corrections* (identified via codebase audit): GPU warmup, `time.perf_counter` for timing precision, and corrected SHOT-IM parameterisation. Moreover, we assign null predictions to skipped batches to preserve a clean availability-accuracy separation.
>
> We have conducted a comprehensive evaluation of the buffering impact but omit it here for brevity; we are happy to provide it during the discussion phase and will include it in our appendix section B.
>
> **W3+Q2: SHOT-IM frozen accuracy**
>
> We verified that SHOT-IM, unlike the other adaptation methods, does not discard the source running statistics, meaning it has a more robust estimate of the distribution statistics by when it freezes. Other methods degenerate as their accumulated running statistics are not sufficiently mature; these methods do not collapse under ViT evaluation as layer normalisation does not track running statistics from the target domain. We will include this distinction in our results discussion.
>
> **Q3: Hardware sensitivity**
>
> We ran ResNet-18 on CIFAR-10-C across all three evaluation scenarios on both a Raspberry Pi ($\lambda$ = 486.6 ms) and an RTX 4080 ($\lambda$ = 1.5 ms). The ordinal rankings of methods by adaptation cost is stable across both platforms, with only Tent and ETA swapping within their tier due to measurement noise. Where the winner distribution shifts, the cause is more structural as the underlying kernels for gradient-based methods are less optimised on CPUs. This is acceptable as rankings need only hold in the target evaluation context.
>
> Table 1: Adaptation overhead and win counts across 165 evaluations (CPU and GPU).
> |Method|CPU adapt (ms)|CPU/lambda|GPU adapt (ms)|GPU/lambda|Rank CPU|Rank GPU|CPU wins|GPU wins|
> |---|---|---|---|---|---|---|---|---|
> |Standard|0|<0.01|0.01|0.01|1|2|0|0|
> |AdaBN|0|<0.01|0.01|0.01|2|1|103|60|
> |LAME|0.1|<0.01|0.03|0.02|3|3|0|0|
> |NEO|0.1|<0.01|0.03|0.02|4|4|21|22|
> |Tent|1330.9|3.75|1.48|1.15|6|5|13|45|
> |ETA|1328.6|3.74|1.89|1.46|5|6|13|33|
> |SHOT-IM|2916.1|8.21|3.25|2.52|7|7|15|5|
> |SAR|2975.6|8.37|5.06|3.92|8|8|0|0|
>
> **Q4: Non-episodic evaluation**
>
> Please see our response to reviewer VMv7-W4 for full details.

---

> > ### Author Rebuttal · Reviewer_d95w · 2026-04-01
> >
> > Thank you for the thorough response. The extended experiments (ViT-B/16, ResNet-18, new datasets, new methods, Raspberry Pi hardware) and the mechanistic SHOT-IM explanation resolve the empirical concerns raised in the review.
> >
> > The novelty concern relative to Alfarra et al. (2024) remains. The decompositions are a real contribution, but the core advance is still extending an existing evaluation paradigm using standard formulations. The broader experiments strengthen the empirical case without changing the conceptual distance from prior work.

---

> > > ### Author Response · Authors · 2026-04-08
> > >
> > > We thank the reviewer for their engagement and are glad the empirical concern was resolved. Below, we further clarify the novelty relative to Alfarra et al., focusing on discrete evaluation, as the continuous and amortised ones are unique to our work.
> > >
> > > Our discrete *scenario* is conceptually similar to that proposed by Alfarra et al. Both evaluate adaptation methods under time constraints imposed by an asynchronous data stream. Alfarra et al. focus almost exclusively on the dual-model scenario, which guarantees availability (α) by construction as a fallback model serves “skipped” samples. We focus on the single-model scenario briefly noted in their appendix. The substantive differences lie in our evaluation *protocol* (measurement process) and *metrics* (utility decomposition). We detail these across (1) protocol design, (2) evaluation purpose, and (3) methodological correctness.
> > >
> > > **Protocol design**
> > >
> > > 1. *Fixed inter-arrival time*. Alfarra et al. impose time constraints in the relative sense. They recompute the arrival interval (γ) per batch by running standard inference (reference) immediately before each adaptation step. The reference (arrival interval) and adaptation wall times then determine how many subsequent batches to skip. Arrivals are thus endogenously set and vary with per-batch measurement noise. We instead fix the interval using an offline sweep (mean + 6σ), making our timing model **stable and independent**. This enables practitioners to easily use their constraints and reduces the total evaluation time, as only the adaptive model runs on served batches.
> > > 2. *Buffer-based scheduling*. They use a ceiling operation for batch-skipping. It forces the model pipeline to idle when a method finishes at t+ε; it idles till the next arrival at t+γ, an unintended side effect rather than a motivated constraint. We introduce a **single-batch buffer to keep the pipeline saturated, reflecting standard system design practice**. In ours, skipping is not predetermined and emerges as a natural consequence of the wall-time simulation. Table 1, new for this rebuttal, highlights the impact of this change, particularly in its fair treatment of efficient methods.
> > >
> > > Table 1. Unbuffered (Ub, theirs) vs Buffered-100 (Buf, ours). Metrics averaged over 15 IN-C corruptions on RN-50. Ub is our fixed-interval variant of Alfarra et al.’s evaluation. It uses our wall-time simulation and includes our methodological corrections but is otherwise faithful to their intent. Idle-Ub is the forced idle time under Ub.
> > >
> > > |Method|Rank|α-Ub%|α-Buf%|Δα|Util.-Ub%|Util.-Buf%|ΔUtil.|Idle-Ub(s)|
> > > |---|---|---|---|---|---|---|---|---|
> > > |NEO|1→2|100.0|100.0|0.0|22.1|22.1|0.0|0.0|
> > > |Basic|2→4|100.0|100.0|0.0|18.2|18.2|0.0|0.0|
> > > |AdaBN|3→1|50.1|97.2|47.1|15.9|30.8|14.9|15.1|
> > > |ETA|4→3|33.4|41.0|7.6|15.0|18.7|3.7|5.7|
> > > |Tent|5→6|33.4|41.2|7.8|13.1|16.6|3.6|5.9|
> > > |LAME|6→5|66.8|98.8|32.0|13.0|17.3|4.3|10.1|
> > > |DeYO|7→7|26.3|31.8|5.5|11.7|14.3|2.6|5.4|
> > > |CMF|8→9|22.2|25.1|2.9|10.1|11.5|1.4|3.5|
> > > |SHOT|9→8|25.1|33.2|8.1|9.9|13.5|3.6|7.6|
> > > |SAR|10→10|20.1|20.6|0.5|7.5|7.8|0.3|0.7|
> > >
> > > Under Ub, AdaBN and ETA rank below the non-adaptive baseline mainly due to forced idling; AdaBN spends 15s idling (nearly half the simulation time). Ub systematically underestimates utility. The buffer corrects this. It does prolong the mean response time as batches wait in the buffer, but this is acceptable as the discrete protocol imposes no timeliness constraint, which concerns the continuous protocol.
> > >
> > > **Evaluative purpose**
> > >
> > > 1. *Interpretable decomposition.* Alfarra et al. focus on whether methods yield tangible returns when starved of batches under a shared, formalised penalty; ours focus on why some do and some don’t, revealed through the accuracy-availability decomposition absent from their framing. This surfaces availability as a failure mode, which is necessary for identifying what properties a deployable adaptation method must satisfy.
> > > 2. *Systematic utilisation sweep.* We sweep across a range of constraints to reveal whether a method degrades gracefully or collapses abruptly. They evaluate at a single operating point, which cannot reveal this distinction.
> > >
> > > **Methodological correctness**
> > >
> > > 1. *Null predictions.* As noted in CgEt-Q1.
> > > 2. *Arrival interval.* Their per-batch reference pass introduces potential priming effects from back-to-back model execution. Also, for methods that remove source normalisation statistics, this pass performs AdaBN implicitly, making the arrival interval method-dependent rather than shared.
> > >
> > > ---
> > >
> > > **Final remark.** We hope our response clarifies that the differences of our work relative to Alfarra et al. are significant along both design and empirical contributions. Together with the novel continuous and amortised protocols and the extensive additional experiments provided during rebuttal, we believe the revised manuscript represents a meaningful advance and would be grateful if the reviewer could consider this when finalising their score.

---

### Official Review · Reviewer_CgEt · 2026-03-13

**Soundness:** 3
**Presentation:** 4
**Significance:** 3
**Originality:** 3
**Overall Recommendation:** 4
**Confidence:** 5

**Summary:**

The paper studies test-time-adaptation (TTA) and extends previous work that embeds realistic deployment constraints to TTA. In TTA, a model is adapting online to an input stream of unlabeled image batches that experience some distribution shift over time. Alfarra et al. introduced the idea of penalizing TTA methods based on the time it takes them to process an incoming batch (slower methods would not be able to keep up with a real-time batch stream, so we should provide them fewer samples for adaptation). This paper extends that idea into a framework called Tempora, which introduces three concrete metrics to evaluate models under several realistic scenarios: discrete utility with deadlines (asynchronous stream but with hard deadlines for processing batches), continuous utility with latency decay (interactive stream where late predictions are less valuable), and amortized utility with overhead budgets (for budget-constrained setups where a model freezes once compute is exhausted). The punchline is that typical ranking of TTA methods is not predictive of ranking under this realistic time-constrained setup.

**Compliance With Llm Reviewing Policy:**

Affirmed.

**Key Questions For Authors:**

These questions are mostly for me to understand the details of the paper. To address the weaknesses, you can read the above section which should contain plenty of details about what is affecting my rating. I am happy to update my rating if further clarification is provided.

1. How do you deal with skipped batches when evaluating the performance of a method? If I remember correctly, Alfarra et al. would just assign random predictions to them or something like that, which makes including them in the metric computation straightforward.
2. I think the continuous and amortized scenarios in this paper are quite novel and interesting. Am I correct to think that the discrete utility for asynchronous streams with hard deadlines is basically what Alfarra et al. proposed but with a few modified design choices? Can you please list them here and explain why they needed to be modified?
3. Do you have plans to benchmark ViT as a backbone and include that in the future?
4. Why did the authors choose hyperbolic discounting for the continuous utility metric, and have they considered other discounting choices?

**Limitations:**

I would encourage the authors to explicitly mention that results are based on a single run with a random seed, rather than aggregated metrics across several seeds. In cases where some methods are close to each other, the rankings could slightly change due to randomness.

**Strengths And Weaknesses:**

## Strengths

1. The main strength of this paper is the warning that researchers should be careful in selecting a single TTA method based on offline accuracy. I completely agree, and I think this extends beyond TTA to all distribution shift problems. There is a lot of documented variation in performance across setups and it is unjustified to call a method a winner without rigorously testing lots of methods in one's target setup. The result that rankings invert when time constraints are imposed is a good reminder for the community.
2. The analysis of corruption-specific tradeoffs is also useful to direct community efforts towards more flexible methods that allocate compute based on corruptions. Although my guess is that such methods are still hard to develop without first developing a dedicated benchmark that helps estimating corruption difficulty and evaluating adaptable approaches on that benchmark itself. I personally have learned a lot from paragraph "Method design" in section 4.3.
3. I believe the application of the continuous and amortized utility metrics to TTA is quite novel and can inform the development of more real-world TTA methods.

## Weaknesses
1. The online time-constrained issue studied here (the system must predict on the current input before observing the next input) has been extensively addressed in the online CL community. In fact, Ghunaim et al. [A] addresses the exact same problem in online CL and arrives at the same central argument: when we move from conventional offline evaluations to modeling of realistic time constraints, model ranking invert and a simple inexpensive baseline completely outranks popular methods. Researchers in the community have already proposed some budget-aware approaches to online CL [B]. The stream-model relative complexity metric defined in [A] serves a similar purpose to the discrete utility defined in Tempora. And it seems that the amortized utility proposed here is also quite similar to the "slow stream" setup from Ghunaim et al. The current submission could benefit from a more extensive discussion about this (which could actually be framed in a way that brings more attention to Tempora. The observation that compute-aware evaluation break offline rankings is not specific to TTA but is a general phenomenon across online learning paradigms. So developing methods for these problems can help in many settings.)
2. There is an alternative explanation to why model ranking invert in this more realistic TTA setup: hyperparameter sensitivity. The TTAB paper cited by the authors extensively showed how variable TTA method performance could be with untuned hyperparameters. It is not ruled out, empirically at least, that the optimal offline methods discussed in the paper could perform really well under time-constraints if their hyperparameters are tuned again. This is even more difficult to rule out given that the results are reported for a single seed (unless I misunderstood the appendix details?)
3. The timing measurements in the paper are hardware specific. I could be mistaken, but I think ordering of methods by timing may vary significantly based on what their operations are optimized for. A discussion of why that is still acceptable would be useful.

- [A]: Real-Time Evaluation in Online Continual Learning: A New Hope, CVPR 2023
- [B]: Budgeted Online Continual Learning by Adaptive Layer Freezing and Frequency-based Sampling, ICLR 2025

---

> ### Author Rebuttal · Authors · 2026-03-31
>
> **W1: Relationship to Ghunaim et al.**
>
> We thank the reviewer for this framing. Tempora's archetypes and decompositions are not TTA specific; they apply wherever a model must adapt under time constraints, and we see their application to other online learning paradigms as a natural extension. We will note this in our revision and also include a progression of rank instability from Ghunaim et al. (online CL) to Alfarra et al. (online TTA) in our discussion of related works. Below, we clarify the specific parallels raised:
>
> 1. Both Ghunaim et al.'s relative complexity $C_s$ and Alfarra et al.'s relative adaptation speed are integer ratios used as *inputs* to predetermine how many batches to skip at each update step. Neither work tracks skipped batches as an aggregate *outcome* (availability) over the stream. Tempora's discrete utility captures this to reveal two failure modes: system errors (skipped batches) and model errors (incorrect predictions). This diagnostic capability is absent from both prior works, which only report accuracy.
>
> 2. The slow stream setup evaluates each method *in isolation* against a baseline scaled up to match its complexity; this FLOPs budget is per-batch and endogenous. The stream speed is tuned to match this, so updates occur on every batch. Here, time plays no meaningful role in shaping model behaviour. In contrast, Tempora's amortised evaluation applies a stream-wide, exogenous budget $B$ (in ms), chosen by a practitioner, uniformly *across all* methods. Once a method exhausts this budget, it switches to frozen inference. We sweep $B$ across a range of values. Our paradigm is grounded in deployment realism, theirs in comparative fairness.
>
> 3. Prior works observe the *existence* of rank instability. Tempora advances this in two ways: (1) by grounding evaluation in physical units (ms) rather than relative proxies, contraints become interpretable and consistent across architectures, and (2) by decomposing utility, we provide a framework to diagnose *why* rankings change. We will revise our contributions section to clarify this distinction. To respect character limits, we detail both advances in our response to reviewer d95w-W2.
>
> **W2: Sensitivity to randomness and hyperparameters**
>
> Results are reported on a single seed (2025); we will make this and the surrounding limitation explicit in our revision. As a sanity check, we ran two additional seeds on gaussian noise for ResNet-50 (Tab. 1) and ViT-B/16 (omitted for brevity). The strictest discrete and amortised settings are where randomness is most likely to influence results. Standard deviations are negligible, consistent with Alfarra et al.'s observation. Gradient-based methods show slightly higher variance than gradient-free ones, as expected.
>
> Table 1. ResNet-50, gaussian noise (std. dev. across three seeds)
> |Protocol|basic|adabn|lame|neo|tent|eta|shot|sar|cmf|deyo|
> |---|---|---|---|---|---|---|---|---|---|---|
> |Discrete (ρ=100%)|0.002|0.049|0.015|0.007|0.068|0.071|0.063|0.197|0.131|0.054|
> |Amortised (B=1000)|0.002|0.043|0.006|0.007|0.011|0.014|0.092|0.011|0.018|0.016|
>
> We agree that hyperparameters influence rankings and that it is a broader epistemic concern with both TTA method design and evaluation. We intentionally evaluate all methods at their defaults and will acknowledge this as a limitation of scope. That said, hyperparameter tuning cannot change a method's adaptation overhead. For methods approaching computational insolvency, the temporal component determines utility; this is a structural ceiling that no hyperparameter can change. As such, there may be method pairs that tuning cannot simply invert.
>
> **W3: Sensitivity to hardware**
>
> Please see our response to reviewer d95w-Q3 for full details.
>
> **Q1: Treatment of skipped batches**
>
> We assign invalid labels to skipped samples, making them de facto incorrect. Alfarra et al. assign random labels, but this contributes false positives. We note both points in appendix B (in a footnote) and will make it more prominent in our revision. Random labels would also obscure the true accuracy contributed by served batches, which Alfarra et al. do not track.
>
> **Q2: Discrete protocol modifications**
>
> Please see our response to reviewer d95w-W2 for full details.
>
> **Q3: ViT results**
>
> We evaluated ViT-B/16 on ImageNet-{C,R,V2} with SPA, CMF, DeYO and ZeroSIAM. Please see our responses to reviewer QNYa-W3+W4 for full details.
>
> **Q4: Rationale for hyperbolic discounting**
>
> Hyperbolic discounting is monotone decreasing, bounded in $[0,1]$, equals 1 at zero overhead, and continuous. Its parameter $T$ is anchored to HCI-motivated latency thresholds, giving it an interpretable meaning. We evaluated exponential and step discounting as alternatives: exponential collapses all gradient-based methods to near-zero utility at tight thresholds; step degenerates at relaxed thresholds, reducing utility to offline accuracy. Hyperbolic preserves meaningful ranking gradataion across thresholds.

---

> > ### Author Rebuttal · Reviewer_CgEt · 2026-04-03
> >
> > I thank the authors for the thoughtful and detailed rebuttal / answers to my questions. My concerns are addressed, and I think a framework to diagnose why rankings change can be very useful.
> >
> > I encourage the authors to implement the changes they are proposing to the related work section, as it provides very useful context for researchers who will build on this work. I also encourage them to caveat the results with the note on Sensitivity to randomness and hyperparameters.
> >
> > As for the point re the "structural ceiling that no hyperparameter can change", researchers can use this as a guideline to know when it's worth it to focus on efficiency rather than improving performance by adding more components or operations.
> >
> > I am happy to update my score.

---

> > > ### Author Response · Authors · 2026-04-08
> > >
> > > We thank the reviewer for raising their score and appreciate their acknowledgement of the significance of our work and our effort in addressing their concerns. We will reflect the proposed changes in the revised manuscript.

---

### Decision · Program_Chairs · 2026-04-30

**Decision:**

Accept (regular)

**Comment:**

TODO borderline but likely accept. main remaining concern is novelty w.r.t. to one work but there are contributions.. 3/4 reviewers lean toward accept and one increased score (but to 3 and not 4).

CgEt and d95w both confirm issues are addressed but do not increase their scores. AC still sees these as resolved and gives the work a boost.